# Percentile Criterion Optimization in Offline Reinforcement Learning

**Elita A. Lobo**
Department of Computer Science
University of Massachusetts Amherst
elobo@umass.edu

**Cyrus Cousins**
Department of Computer Science
University of Massachusetts Amherst
cbcousins@umass.edu

**Yair Zick**
Department of Computer Science
University of Massachusetts Amherst
yzick@umass.edu

**Marek Petrik**
Department of Computer Science
University of New Hampshire
mpetrik@cs.unh.edu

## Abstract

In reinforcement learning, robust policies for high-stakes decision-making problems with limited data are usually computed by optimizing the *percentile criterion*. The percentile criterion is approximately solved by constructing an *ambiguity set* that contains the true model with high probability and optimizing the policy for the worst model in the set. Since the percentile criterion is non-convex, constructing ambiguity sets is often challenging. Existing work uses *Bayesian credible regions* as ambiguity sets, but they are often unnecessarily large and result in learning overly conservative policies. To overcome these shortcomings, we propose a novel Value-at-Risk based dynamic programming algorithm to optimize the percentile criterion without explicitly constructing any ambiguity sets. Our theoretical and empirical results show that our algorithm implicitly constructs much smaller ambiguity sets and learns less conservative robust policies.

## 1 Introduction

Batch Reinforcement Learning (Batch RL) [26] is popularly used for solving sequential decision-making problems using limited data. These algorithms are crucial in high-stakes domains where exploration is either infeasible or expensive, and policies must be learned from limited data. In model-based Batch RL algorithms, transition probabilities are learned from the data as well. Due to insufficient data, these transition probabilities are often imprecise. Errors in transition probabilities can accumulate, resulting in low-performing policies that fail when deployed.

To account for the uncertainty in transition probabilities, prior work uses Bayesian models [10, 13, 27, 40, 45, 47] to model uncertainty and optimize the policy to maximize the returns corresponding to the worst $\alpha$-percentile transition probability model. These policies guarantee that the true expected returns will be at least as large as the optimal returns with high confidence. This technique is commonly referred to as the *percentile-criterion* optimization. Unfortunately, the percentile criterion is NP-hard to optimize. Thus, current work uses *Robust Markov Decision Processes* (RMDPs) to optimize a lower bound on the percentile criterion. An RMDP takes as input an ambiguity set (uncertainty set) that contains the true transition probability model with high confidence and finds a policy that maximizes the returns of the worst model in the ambiguity set.

Since the percentile criterion is non-convex, constructing ambiguity sets itself is a challenging problem. Existing work uses Bayesian credible regions (BCR) [40] as ambiguity sets. However, these ambiguity sets are often unnecessarily large [15, 40] and result in learning conservative robust policies.

37th Conference on Neural Information Processing Systems (NeurIPS 2023).

Some recent works approximate ambiguity sets using various heuristics [3, 40], but we show that they remain too conservative. Thus, the question of the *optimal ambiguity set, i.e., ambiguity sets that result in optimizing the tightest possible lower bound on the percentile criterion and less-conservative policies* remains unanswered.

**Our Contributions**  In this paper, we answer two important questions: a) *Are Bayesian credible regions the optimal ambiguity sets for optimizing the percentile criterion?* b) *Can we obtain a less conservative solution to the percentile criterion than RMDPs with BCR ambiguity sets while retaining its percentile guarantees?* Our theoretical findings show that Bayesian credible regions can grow significantly with the number of states and therefore, tend to be unnecessarily large, resulting in highly conservative policies. As our main contribution, we provide a dynamic programming framework (Section 3), which we name the $\mathrm{VaR}$ framework, for optimizing a lower bound on the percentile criterion without explicitly constructing ambiguity sets. Specifically, we propose a new robust Bellman operator, the Value at Risk (VaR) Bellman operator, for optimizing the percentile criterion. We show that it is a valid contraction mapping that optimizes a tighter lower bound on the percentile criterion, compared to RMDPs with BCR ambiguity sets (Section 3). We theoretically analyze and bound the performance loss of our framework (Section 3.1). We provide a Generalized $\mathrm{VaR}$ Value iteration algorithm and analyze its error bounds. We also show that there exist directions in which the Bayesian credible regions can grow unnecessarily large with the number of states in the MDP and possibly result in a conservative solution. On the other hand, the ambiguity sets implicitly optimized by the $\mathrm{VaR}$ Bellman operator tend to be smaller, i.e., they have a smaller asymptotic radius and are independent of the number of states (Section 4). Finally, we empirically demonstrate the efficacy of our framework in three domains (Section 5).

## 1.1  Related Work

Several work[13, 34, 40] propose different methods for solving the percentile criterion, as well as other robust measures for handling uncertainty in the transition probabilities estimates. Russel and Petrik [40] and Behzadian et al. [3] propose various heuristics for minimizing the size of the ambiguity sets constructed for the percentile-criterion. Russel and Petrik [40] propose a method that interleaves robust value iteration with ambiguity set size optimization. Behzadian et al. [3] propose an iterative algorithm that optimizes the weights of $\ell_1$ and $\ell_\infty$ ambiguity sets while optimizing the robust policy. However, these methods still construct Bayesian credible sets which can be unnecessarily large and result in conservative policies, as we show in Section 5.

Other works consider partial correlations between uncertain transition probabilities to mitigate the conservativeness of learned policies [4, 15, 19, 29, 30]. These approaches mitigate the conservativeness of S- and SA-rectangular ambiguity sets by capturing correlations between the uncertainty and by limiting the number of times the uncertain parameters deviate from the mean parameters. Despite these heuristics, most of these works [2, 20, 40, 53] either rely on weak statistical concentration bounds to construct frequentist ambiguity sets, or use Bayesian credible regions as ambiguity sets. These sets still tend to be unnecessarily large [15, 40], resulting in conservative policies.

Finally, a large number of works [12, 16, 36, 44, 49, 50, 52] have proposed RL algorithms that use measures like Conditional Value at Risk, Entropic risk measure amongst other risk measures. However, we note that these works use risk measures to obtain robustness guarantees against aleatoric uncertainty (system uncertainty) and not epistemic uncertainty (model uncertainty), which is the focus of our work. Since these works optimize a completely different objective, we do not compare our framework against theirs. Robust RL work [2, 14, 20, 28, 32, 55] proposes other robust measures for handling uncertainty in transition probabilities; however, these approaches do not provide probabilistic guarantees on the expected returns, and compute overly conservative policies.

## 2  Preliminaries

In the standard reinforcement learning setting, a sequential decision task is modeled as a Markov Decision Process (MDP) [37, 48]. An MDP is a tuple $\langle \mathcal{S}, \mathcal{A}, P, R, \boldsymbol{p}_0, \gamma \rangle$ that consists of (a) a set of states $\mathcal{S} = \{1, 2, \ldots, S\}$, (b) a set of actions $\mathcal{A} = \{1, 2, \ldots, A\}$, (c) a deterministic reward function $R \colon \mathcal{S} \times \mathcal{A} \times \mathcal{S} \to \mathbb{R}$, (d) a transition probability function $P \colon \mathcal{S} \times \mathcal{A} \to \Delta^S$, (e) an initial state distribution $\boldsymbol{p}_0 \in \Delta^S$, where $\Delta^S$ represents the $S$-dimensional probability simplex, and (f) a

discount factor $\gamma \in [0, 1]$. We use $\boldsymbol{p}_{s,a}$ to denote the vector of transition probabilities $P(s, a, \cdot)$ corresponding to the state $s$ and action $a$. Likewise, we use $\boldsymbol{r}_{s,a}$ to denote the vector of rewards $R(s, a, \cdot)$ corresponding to state $s$ and action $a$. A Markovian policy $\pi \colon \mathcal{S} \to \Delta^A$ maps each state $s$ to a distribution over actions $\mathcal{A}$. In a general RL setting, the goal is to compute a policy $\pi$ that maximizes the expected discounted return $\rho(\pi, P)$ over an infinite horizon,

$$\max_{\pi \in \Pi} \rho(\pi, P) = \max_{\pi \in \Pi} \mathbb{E}\left[\sum_{t=0}^{\infty} \gamma^t r(s_t, a_t, s_{t+1}) \mid s_0 \sim \boldsymbol{p}_0, a_t \sim \pi(s_t), s_{t+1} \sim P(s_t, a_t, \cdot)\right] \ .$$

The value of a policy $\pi$ at any state $s$ is the discounted sum of rewards received by an RL agent, if it starts from state $s$, i.e., $v^\pi(s) = \mathbb{E}\left[\sum_{t=0}^{\infty} \gamma^t r(s_t, a_t, s_{t+1}) | s_0 = s, a_t \sim \pi, s_{t+1} \sim P(s_t, a_t, \cdot)\right]$.

We assume a *batch reinforcement learning* setting [26] where the reward function is known, but the true transition probabilities $P^*$ are unknown. Following prior work on robust Bayesian RL [8, 13, 40, 54], we use parametric Bayesian models to represent uncertainty over the true transition probabilities $P^*$. We will use $\tilde{\boldsymbol{P}} = \{\tilde{\boldsymbol{p}}_{s,a}\}_{s \in \mathcal{S}, a \in \mathcal{A}}$ to denote the random transition probabilities. We assume we have a batch of data $\mathcal{D} = \{s_i, a_i, s_i'\}_{i=1}^N$, which in conjunction with some prior distribution defines the *posterior distribution* of transition probabilities $\tilde{\boldsymbol{P}}$.

The Fisher information measures the amount of information about the unknown true parameters $\boldsymbol{\theta}^* \in \mathbb{R}^d$ carried by an observable random variable $\tilde{\boldsymbol{X}} \in \mathbb{R}^m$. If $f(\boldsymbol{x}; \boldsymbol{\theta}^*)$ is the probability density of $\tilde{\boldsymbol{X}}$ conditioned on $\boldsymbol{\theta}^*$, then the Fisher information is given by $I(\boldsymbol{\theta}^*) = \mathbb{E}\left[(\nabla \log f(\tilde{\boldsymbol{X}}; \boldsymbol{\theta}^*))(\nabla \log f(\tilde{\boldsymbol{X}}; \boldsymbol{\theta}^*))^\top\right]$.

Then, in the Bayesian setting, the Bernstein von Mises theorem [51] states that under mild regularity conditions the posterior distribution of the parameters $\tilde{\boldsymbol{\theta}}$ converges in the limit to the distribution of the MLE of $\boldsymbol{\theta}^*$ which is asymptotically Gaussian, in particular, $\lim_{N \to \infty} \sqrt{N}(\tilde{\boldsymbol{\theta}} - \boldsymbol{\theta}^*) \rightsquigarrow \mathcal{N}(\boldsymbol{0}, I(\boldsymbol{\theta}^*)^{-1})$ where $\rightsquigarrow$ indicates convergence in distribution. In many cases, the above holds even though the conditions of the Bernstein-von Mises theorem are not met [17, 18]. Of particular interest in this setting is the Dirichlet distribution where the asymptotic MLE is a multivariate Gaussian under certain conditions on the prior distribution [18], although in this case, it is degenerate (i.e., the covariance matrix is not full-rank).

For asymptotic results, we assume that the data $\mathcal{D}$ is sampled such that each state $s$ and action $a$ is observed infinitely many times as $N \to \infty$. Furthermore, we assume henceforth that the prior is asymptotically negligible, and the MLE is asymptotically Gaussian with covariance matrix $I(\boldsymbol{P}^*)^{-1}/N$. Therefore, the posterior distribution of $\tilde{\boldsymbol{P}}$ is also asymptotically Gaussian with covariance matrix $I(\boldsymbol{P}^*)^{-1}/N$. Under these conditions, we will estimate the Fisher information corresponding to $\boldsymbol{\theta}^* = \boldsymbol{P}^*$ using the asymptotic covariance of the posterior distribution of $\tilde{\boldsymbol{P}}$.

To avoid unnecessary computational technicalities, we will assume that $\tilde{\boldsymbol{P}}$ is a discrete random variable taking on values $\tilde{\boldsymbol{P}}(\omega), \omega \in \Omega$ for some $\Omega = \{1, \ldots, M\}$ with a distribution $f$. That is, the random variable $\tilde{\boldsymbol{P}}$ represents a discrete approximation of the true, possibly continuous posterior, as is common in methods like Sample Average Approximation (SAA) [43].

**Percentile Criterion**  The $\alpha$-percentile criterion is popularly used to derive robust policies under model uncertainty [13]. It aims to compute a policy $\pi$ that maximizes the returns corresponding to the worst $\alpha$-percentile model:

$$\arg\max_{\pi \in \Pi} \left\{ y \in \mathbb{R} \,\Big|\, \Pr_{\tilde{\boldsymbol{P}} \sim f}\left[\rho(\pi, \tilde{\boldsymbol{P}}) \geq y\right] \geq 1 - \alpha \right\}. \tag{1}$$

The value $y$ lower-bounds the true expected discounted returns with confidence $1 - \alpha$ where $\alpha \in (0, 1/2)$. Optimizing the percentile criterion is equivalent to optimizing the Value at Risk ($\mathrm{VaR}_\alpha$) of expected discounted returns when there exists uncertainty in transition probabilities $\tilde{\boldsymbol{P}}$ and the expected returns function $\rho$ is lower-semicontinuous. The optimization in (1) is equivalent to

$$\max_{\pi \in \Pi} \mathrm{VaR}_\alpha\left[\rho(\pi, \tilde{\boldsymbol{P}})\right] \ , \tag{2}$$

where $\mathrm{VaR}_\alpha$ of a bounded random variable $\tilde{X}$ with a CDF function $F \colon \mathbb{R} \to [0, 1]$ is defined as [38]

$$\mathrm{VaR}_\alpha[\tilde{X}] = \sup\{t \in \mathbb{R} : \Pr[\tilde{X} \geq t] \geq 1 - \alpha\} \ . \tag{3}$$

A lower value of $\alpha$ in (1) indicates a higher confidence in the returns achieved in expectation. For example, $\mathrm{VaR}_{0.05}[\rho(\pi, \hat{\boldsymbol{P}})] = x$ indicates that the true returns will be at least equal to the robust returns $x$ for 95% of the transition probability models. When clear from context, we use VaR to denote the Value at Risk at confidence level $\alpha$. Unfortunately, the optimization problem in (1) is NP-hard to optimize and is usually approximately solved using Robust MDPs.

**Robust MDPs**  Robust MDPs (RMDPs) generalize MDPs to account for uncertainty, or ambiguity, in the transition probabilities. An *ambiguity set* for an RMDP is constructed such that it contains the true model with high confidence. The optimal policy of a Robust MDP $\pi^*$ maximizes the returns of the worst model in the ambiguity set: $\pi^* = \arg\max_{\pi \in \Pi} \min_{\boldsymbol{P} \in \mathcal{P}} \rho(\pi, \boldsymbol{P})$. General RMDPs are NP-hard to solve [53], but they are tractable for broad classes of ambiguity sets. The simplest such type is the SA-rectangular ambiguity set [33, 53], defined as

$$\mathcal{P} = \left\{ \boldsymbol{P} \in (\Delta^S)^{S \times A} \mid \boldsymbol{p}_{s,a} \in \mathcal{P}_{s,a}, \, \forall s \in \mathcal{S}, \, \forall a \in \mathcal{A} \right\} \ ,$$

for a given $\mathcal{P}_{s,a} \subseteq \Delta^{\mathcal{S}}, s \in \mathcal{S}, a \in \mathcal{A}$. SA-rectangular ambiguity sets [3, 40] assume that the transition probabilities corresponding to each state-action pair are independent. Similarly to MDPs, the optimal robust value function $\boldsymbol{v}^* \in \mathbb{R}^S$ for an SA-rectangular RMDP is the unique fixed point of the robust Bellman optimality operator $\mathcal{T} \colon \mathbb{R}^S \to \mathbb{R}^S$ defined as $(\mathcal{T}\boldsymbol{v})_s = \max_{a \in \mathcal{A}} \min_{\boldsymbol{p}_{s,a} \in \mathcal{P}_{s,a}} \boldsymbol{p}_{s,a}^{\mathsf{T}} (\boldsymbol{r}_{s,a} + \gamma \cdot \boldsymbol{v})$.

To optimize the percentile criterion, an SA-rectangular ambiguity set $\mathcal{P}$ is constructed such that it contains the true model with high probability, and thus, the following equation holds.

$$\Pr\left[ \rho(\pi, \tilde{\boldsymbol{P}}) \geq \min_{P \in \mathcal{P}} \rho(\pi, \boldsymbol{P}) \right] \geq 1 - \alpha \ .$$

Although RMDPs have been used to solve the percentile criterion [3], the quality of the robust policies it computes depends mainly on the size of the ambiguity sets. The larger the ambiguity sets, the more conservative the robust policy [30]. SA-rectangular ambiguity sets are most commonly studied; thus we focus our attention on SA-rectangular Robust MDPs. We investigate whether Bayesian credible regions are optimal ambiguity sets for optimizing the percentile criterion. We refer to SA-rectangular RMDPs and SA-rectangular ambiguity sets as Robust MDPs and ambiguity sets respectively.

Our work focuses on Bayesian (rather than frequentist) ambiguity sets. Bayesian ambiguity sets are usually constructed from Bayesian credible regions (BCR) [3, 40]. Given a state $s$ and an action $a$, let $\psi_{s,a}$ represent the size of the BCR ambiguity sets; $\mathcal{P}_{s,a}^{\mathrm{BCR}_\alpha}$ and $\bar{\boldsymbol{p}}_{s,a}$ represent the mean transition probabilities. The set $\mathcal{P}_{s,a}^{\mathrm{BCR}_\alpha}$ is constructed as

$$\mathcal{P}_{s,a}^{\mathrm{BCR}_\alpha} = \mathcal{P}_{s,a}(\boldsymbol{b}, \psi, q) = \left\{ \boldsymbol{p}_{s,a} \in \Delta^S \big| \|\boldsymbol{p}_{s,a} - \bar{\boldsymbol{p}}_{s,a}\|_{q,\boldsymbol{b}} \leq \psi_{s,a} \right\} \ , \tag{4}$$

where $\boldsymbol{q} \in \{1, \infty\}$ represents the norm of the weighted ball in (4) and $\boldsymbol{b} \in \mathbb{R}_+^S$ is a weight vector. Here, $\boldsymbol{b}$ is jointly optimized with $\psi \in \mathbb{R}$ to minimize the span of the ambiguity sets such that the true model is contained in the ambiguity set with high confidence, i.e., $\Pr(\tilde{\boldsymbol{p}}_{s,a} \in \mathcal{P}_{s,a}(\boldsymbol{b}, \psi, q)) \geq 1 - \alpha$. We refer to BCR ambiguity sets with non-uniform weights as weighted BCR ambiguity sets. We refer to the Robust Bellman optimality operator with BCR ambiguity sets ($\mathcal{T}_{\mathrm{BCR}_\alpha}$) as the $\mathrm{BCR}_\alpha$ Bellman optimality operator, and to RMDPs with BCR ambiguity sets as BCR RMDPs. For any $\delta \in (0, 1/2)$, setting the confidence level $\alpha$ in $\mathcal{T}_{\mathrm{BCR}_\alpha}$ to $\delta/SA$ for all state-action pairs yields $1 - \delta$ confidence on the returns of the optimal robust policy [3]. However, we show that even span-optimized BCR RMDPs can be sub-optimal for optimizing the percentile criterion.

We use the shorthand $\boldsymbol{w}_{s,a}$ for any $s \in \mathcal{S}, a \in \mathcal{A}$ to denote the vector of values associated with value $\boldsymbol{v} \in \mathbb{R}^S$ and the one-step return from state $s$ and action $a$, i.e., $\boldsymbol{w}_{s,a} = \boldsymbol{r}_{s,a} + \gamma \boldsymbol{v}$. We use $\bar{\boldsymbol{p}}_{s,a} \in \mathbb{R}^s$ and $\boldsymbol{\Sigma}_{s,a} \in \mathbb{R}^{S \times S}$ for any $s \in \mathcal{S}, a \in \mathcal{A}$ to represent the empirical mean and covariance of transition probabilities $\tilde{\boldsymbol{p}}_{s,a}$ estimated from $\mathcal{D}$. We use *tilde* to indicate that it is a random variable. We use $\phi(\cdot)$ and $\Phi(\cdot)$ to represent the probability distribution function (PDF) and cumulative distribution function (CDF) respectively of the normal distribution with mean 0 and variance 1. The $\boldsymbol{Z}$-Minkowski norm $\|\boldsymbol{x}\|_Z$ for a vector $\boldsymbol{x}$ given some positive-definite matrix $\boldsymbol{Z}$ is defined as $\|\boldsymbol{x}\|_{\boldsymbol{Z}} = \sqrt{\boldsymbol{x}^{\mathsf{T}} \boldsymbol{Z}^{-1} \boldsymbol{x}}$.

We illustrate the conservativeness of $\mathrm{BCR}_\alpha$ ambiguity sets with the following example.

*Example* 2.1.  Consider an MDP with four states $\{s_0, s_1, s_2, s_3\}$ and a single action $\{a_0\}$. The state $s_0$ is the initial state and the states $s_1, s_2, s_3$ are terminal states with zero rewards. For the

sake of simplicity, we assume that it is only possible to transition to state $s_1, s_2$ and $s_3$ from state $s_0$. The transition probability of $\boldsymbol{p}_{s_0,a_0}$ is uncertain and distributed as a Dirichlet distribution $\tilde{\boldsymbol{p}}_{s_0,a_0} \sim \text{Dir}(10,10,1)$ with mean $[0.48, 0.48, 0.04]$. The rewards for transitions from state $s_0$ are given by $\boldsymbol{r}_{s_0,a_0} = [0.25, 0.25, -1]$.

We wish to optimize the percentile criterion with confidence level $\delta = 0.2$. Following the sampling procedure proposed by Russel and Petrik [40] to construct a uniformly weighted $\text{BCR}_\alpha$ ambiguity set for $\tilde{\boldsymbol{p}}_{s_0,a}$ with 100 posterior samples, yields an ambiguity set $\mathcal{P}^{\text{BCR}_\alpha}_{s_0,a_0} = \{\boldsymbol{p} \in \Delta^S \,\big|\, \|\boldsymbol{p} - \tilde{\boldsymbol{p}}_{s_0,a_0}\|_1 \le 0.277\}$. In this case, the reward estimate against the worst model in the ambiguity set $\boldsymbol{p} = [0.50, 0.32, 0.18]$ is $\rho^{\text{BCR}_\alpha} = 0.025$. Since we have a single non-terminating state in the MDP, the percentile returns are given by $\rho^{\text{VaR}_\alpha} = \text{VaR}_{0.2}[\tilde{\boldsymbol{p}}^{\mathsf{T}}_{s_0,a_0} \boldsymbol{r}_{s_0,a_0}]$. Computing $\rho^{\text{VaR}_\alpha}$ for Dirichlet distribution $Dir(10, 10, 1)$, we get $\rho^{\text{VaR}_\alpha} = 0.17 > \rho^{\text{BCR}_\alpha}$. Thus, this example shows that BCR ambiguity sets can be unnecessarily large, and thereby result in conservative policies.

## 3 VaR Framework

We introduce the $\text{VaR}$ Bellman optimality operator $\mathcal{T}_{\text{VaR}_\alpha}$ for approximately solving the percentile criterion. We show that $\mathcal{T}_{\text{VaR}_\alpha}$ is a valid Bellman operator: it is a contraction mapping and lower bounds the percentile criterion. For any value function $\boldsymbol{v} \in \mathbb{R}^{\mathcal{S}}$, state $s \in \mathcal{S}$ and action $a \in \mathcal{A}$, we define the $\text{VaR}$ Bellman optimality operator $\mathcal{T}_{\text{VaR}_\alpha}$ as

$$(\mathcal{T}_{\text{VaR}_\alpha} \boldsymbol{v})_s = \max_{a \in \mathcal{A}} \text{VaR}_\alpha \left[ \tilde{\boldsymbol{p}}^{\mathsf{T}}_{s,a}(\boldsymbol{r}_{s,a} + \gamma \boldsymbol{v}) \right] \quad . \tag{5}$$

For each state $s$, $\mathcal{T}_{\text{VaR}_\alpha}$ maximizes the value corresponding to the worst $\alpha$-percentile model. In contrast to the $\text{BCR}_\alpha$ Bellman optimality operator $\mathcal{T}_{\text{BCR}_\alpha}$, computing $\mathcal{T}_{\text{VaR}_\alpha}$ does not require constructing ambiguity sets from confidence regions; it can simply be estimated from samples of the model posterior distribution, as we later show.

**Proposition 3.1** (Validity). *The following properties of $\mathcal{T}_{\text{VaR}_\alpha}$ hold for all value functions $\boldsymbol{u}, \boldsymbol{v} \in \mathbb{R}^{\mathcal{S}}$.*

1. *The operator $\mathcal{T}_{\text{VaR}_\alpha}$ is contraction mapping on $\mathbb{R}^{\mathcal{S}}$: $\|\mathcal{T}_{\text{VaR}_\alpha} \boldsymbol{u} - \mathcal{T}_{\text{VaR}_\alpha} \boldsymbol{v}\|_\infty \le \gamma \|\boldsymbol{u} - \boldsymbol{v}\|_\infty$ .*
2. *The operator $\mathcal{T}_{\text{VaR}_\alpha}$ is monotone: $\boldsymbol{u} \succeq \boldsymbol{v} \Rightarrow \mathcal{T}_{\text{VaR}_\alpha} \boldsymbol{u} \succeq \mathcal{T}_{\text{VaR}_\alpha} \boldsymbol{v}$.*
3. *The equality $\mathcal{T}_{\text{VaR}_\alpha} \hat{\boldsymbol{v}} = \hat{\boldsymbol{v}}$ has a unique solution.*

Proposition 3.1 formally proves that $\mathcal{T}_{\text{VaR}_\alpha}$ is a valid Bellman operator, i.e., it is a contraction mapping, monotone, and has a unique fixed point $\hat{\boldsymbol{v}} = (\mathcal{T}_{\text{VaR}_\alpha} \hat{\boldsymbol{v}})$. Here on, we will refer to the policy ($\hat{\pi}$) corresponding to the fixed point value $\hat{\boldsymbol{v}}$ as the $\text{VaR}_\alpha$ policy.

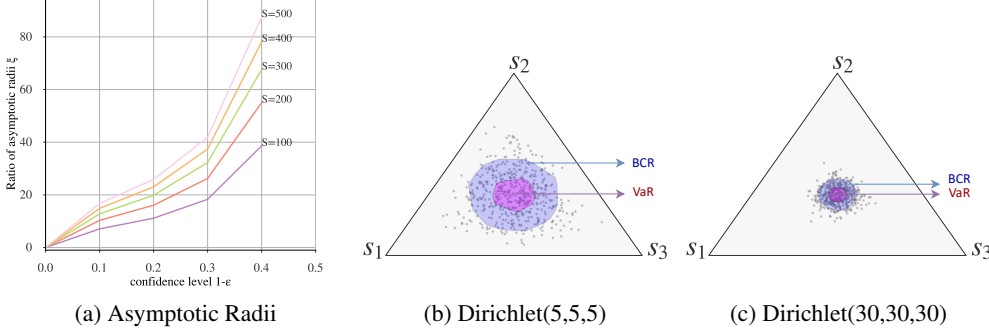

(a) Asymptotic Radii      (b) Dirichlet(5,5,5)      (c) Dirichlet(30,30,30)

Figure 1: Figure 1a (left) compares the asymptotic radius of $\Sigma^{-1}$-Minkowski norm BCR ambiguity sets to $\text{VaR}_\alpha$ ambiguity sets, where $\Sigma$ is the covariance matrix. The size of the BCR ambiguity sets significantly grows with the number of states. Figure 1b and Figure 1c (right) compare the asymptotic forms of $\text{BCR}_\alpha$ and $\text{VaR}_\alpha$ ambiguity sets under high and low uncertainty in $\tilde{\boldsymbol{P}}$ at confidence level $\alpha = 0.2$ respectively. The grey dots represent the transition probabilities samples from a Dirichlet distribution. The sizes of $\text{BCR}_\alpha$ and $\text{VaR}_\alpha$ ambiguity sets increase with an increase in the uncertainty in $\tilde{\boldsymbol{P}}$, however, the $\text{VaR}_\alpha$ ambiguity sets are smaller than the $\text{BCR}_\alpha$ ambiguity sets.

We now show that the VaR Bellman optimality operator $\mathcal{T}_{\mathrm{VaR}_\alpha}$ optimizes a lower bound on the percentile criterion. Given a policy $\pi$, a state $s \in \mathcal{S}$, and transition probabilities $\boldsymbol{P}$, let

$$(\mathcal{T}_{\boldsymbol{P}}^\pi \boldsymbol{v})_s = \sum_{a \in \mathcal{A}} \pi(s,a) \boldsymbol{p}_{s,a}^\mathsf{T}(\boldsymbol{r}_{s,a} + \gamma \boldsymbol{v}) \;\;,$$

represent the Bellman evaluation operator for transition probabilities $\boldsymbol{P}$. Furthermore, let

$$(\mathcal{T}_{\mathrm{VaR}_\alpha}^\pi \boldsymbol{v})_s = \sum_{a \in \mathcal{A}} \pi(s,a) \, \mathrm{VaR}_\alpha \left[ \tilde{\boldsymbol{p}}_{s,a}^\mathsf{T}(\boldsymbol{r}_{s,a} + \gamma \boldsymbol{v}) \right] \;\;,$$

represent the VaR Bellman evaluation operator for random transition probabilities $\tilde{\boldsymbol{P}}$. We use $\hat{\boldsymbol{v}}^\pi$ to denote the fixed point of $\mathcal{T}_{\mathrm{VaR}_\alpha}^\pi$. Furthermore, we use $\tilde{\boldsymbol{v}}^\pi$ to represent the random fixed point of $\mathcal{T}_{\tilde{\boldsymbol{P}}}^\pi$, which is computed using a random realization of the transition probabilities $\tilde{\boldsymbol{P}}$ sampled from the posterior distribution conditioned on observed transitions $\mathcal{D}$.

**Proposition 3.2** (Lower Bound Percentile Criterion). *For any $\delta \in (0, 1/2)$, if we set the confidence level $\alpha$ in the operator $\mathcal{T}_{\mathrm{VaR}_\alpha}^\pi$ to $\delta/s$, then for every policy $\pi \in \Pi : \Pr_{\tilde{\boldsymbol{P}}} \left[ \hat{\boldsymbol{v}}^\pi \preceq \tilde{\boldsymbol{v}}^\pi | \mathcal{D} \right] \geq 1 - \delta$.*

Proposition 3.2 shows that for any policy $\pi$ and state $s$, the $\mathrm{VaR}_\alpha$ value at state $s$, $\hat{\boldsymbol{v}}^\pi(s)$ lower bounds the true value $\tilde{\boldsymbol{v}}^\pi(s)$ with high confidence. Comparing Proposition 3.2 with the definition of the percentile-criterion in (1), we see that the percentile-criterion requires confidence guarantees only on the returns computed from the initial states, whereas the equation in Proposition 3.2 provides confidence guarantees on the value of every state. Therefore, for any policy $\pi$, the value $\boldsymbol{p}_0^\mathsf{T} \hat{\boldsymbol{v}}^\pi$ is a lower bound on the percentile-criterion objective $\mathrm{VaR}_\delta[\rho(\pi, \tilde{\boldsymbol{P}})]$. Since $\mathcal{T}_{\mathrm{VaR}_\alpha}$ finds a policy $\pi$ that maximizes the value $\boldsymbol{p}_0^\mathsf{T} \hat{\boldsymbol{v}}^\pi$, it follows [37] that $\mathcal{T}_{\mathrm{VaR}_\alpha}$ optimizes a lower bound on the percentile criterion in (1).

**Proposition 3.3.** *Suppose that $\tilde{\boldsymbol{p}}_{s,a}$ for any state $s$ and action $a$, is a multivariate sub-Gaussian with mean $\bar{\boldsymbol{p}}_{s,a}$ and covariance factor $\boldsymbol{\Sigma}_{s,a}$, i.e., $\mathbb{E}\left[ \exp\left( \lambda(\tilde{\boldsymbol{p}}_{s,a} - \bar{\boldsymbol{p}}_{s,a})^\mathsf{T} \boldsymbol{w} \right) \right] \leq \exp\left( \lambda^2 \boldsymbol{w}^\mathsf{T} \boldsymbol{\Sigma}_{s,a} \boldsymbol{w}/2 \right), \forall \lambda \in \mathbb{R}, \forall \boldsymbol{w} \in \mathbb{R}^S$. Then, for any state $s \in \mathcal{S}$, $\mathcal{T}_{\mathrm{VaR}_\alpha}$ satisfies*

$$(\mathcal{T}_{\mathrm{VaR}_\alpha} \boldsymbol{v})_s \geq \max_{a \in \mathcal{A}} \left( \bar{\boldsymbol{p}}_{s,a}^\mathsf{T} \boldsymbol{w}_{s,a} - \sqrt{2 \ln(1/\alpha)} \sqrt{\boldsymbol{w}_{s,a} \boldsymbol{\Sigma}_{s,a} \boldsymbol{w}_{s,a}} \right) \;\;.$$

*As a special case, when $\tilde{\boldsymbol{p}}_{s,a}$ is normally distributed $\tilde{\boldsymbol{p}}_{s,a} \sim \mathcal{N}(\bar{\boldsymbol{p}}_{s,a}, \boldsymbol{\Sigma}_{s,a})$, then $\mathcal{T}_{\mathrm{VaR}_\alpha}$ for any state $s \in \mathcal{S}$ can be expressed as*

$$(\mathcal{T}_{\mathrm{VaR}_\alpha} \boldsymbol{v})_s = \max_{a \in \mathcal{A}} \left( \bar{\boldsymbol{p}}_{s,a}^\mathsf{T} \boldsymbol{w}_{s,a} - \Phi^{-1}(1 - \alpha) \sqrt{\boldsymbol{w}_{s,a} \boldsymbol{\Sigma}_{s,a} \boldsymbol{w}_{s,a}} \right) \;\;.$$

Proposition 3.3 shows that by assuming that the transition probabilities are sub-Gaussian, we can easily compute a lower bound of the VaR Bellman update $(\mathcal{T}_{\mathrm{VaR}_\alpha} \boldsymbol{v})$ for a given value function using only the mean and the covariance matrix of $\tilde{\boldsymbol{P}}$. In the special case where $\tilde{\boldsymbol{P}}$ is normally distributed, we can compute the VaR Bellman optimality operator $\mathcal{T}_{\mathrm{VaR}_\alpha}(\boldsymbol{v})$ exactly.

## 3.1 Performance Guarantees

We now derive finite-sample and asymptotic bounds on the loss of the VaR framework.

**Theorem 3.4** (Performance). *Let $\hat{\boldsymbol{v}}$ be the fixed point of the $\mathrm{VaR}$ Bellman optimality operator $\mathcal{T}_{\mathrm{VaR}_\alpha}$ and $\pi^*$ be the optimal policy in (1). Let $\rho^* = \mathrm{VaR}_\delta\left[ \rho(\pi^*, \tilde{\boldsymbol{P}}) \right]$ denote the optimal percentile returns and $\hat{\rho} = \boldsymbol{p}_0^\mathsf{T} \hat{\boldsymbol{v}}$ denote the lower bound on the percentile returns computed using the Bellman operator $\mathcal{T}_{\mathrm{VaR}_\alpha}$ with $\alpha = \delta/s$. Then for each $\delta \in (0, 1/2)$:*

$$\rho^* - \hat{\rho} \leq \frac{1}{1 - \gamma} \max_{s \in \mathcal{S}} \max_{a \in \mathcal{A}} \left( \mathrm{VaR}_{1 - \frac{1-\delta}{S}} \left[ \tilde{\boldsymbol{p}}_{s,a}^\mathsf{T} \boldsymbol{w}_{s,a} \right] - \mathrm{VaR}_\alpha \left[ \tilde{\boldsymbol{p}}_{s,a}^\mathsf{T} \boldsymbol{w}_{s,a} \right] \right) \;\;. \tag{6}$$

Theorem 3.4 bounds the finite sample performance loss of the VaR framework. The loss varies proportionally to the maximum difference between the $\delta/s$ and $1 - (1-\delta/s)$ percentile of the one-step Bellman update for the optimal robust value function $\hat{\boldsymbol{v}}$. Furthermore, the VaR framework performs better when the uncertainty in the transition probabilities is small.

**Theorem 3.5** (Asymptotic Performance). *Suppose that the normality assumptions on the posterior distribution $\tilde{P}$ in Section 2 are satisfied. For any $\delta \in (0, 1/2)$, set $\alpha = \delta/S$ in $\mathcal{T}_{\mathrm{VaR}_\alpha}$. For any state $s$ and action $a$, let $I(\boldsymbol{p}^*_{s,a})$ be the Fisher information matrix corresponding to the true transition probabilities $\boldsymbol{p}^*_{s,a}$. Furthermore, let $\sigma^2_{\max} = \max_{s\in\mathcal{S},a\in\mathcal{A}} \hat{\boldsymbol{w}}^\mathsf{T}_{s,a} I(\boldsymbol{p}^*_{s,a})^{-1} \hat{\boldsymbol{w}}_{s,a}$ be the maximum over state-action pairs of the asymptotic variance of the returns estimate $\tilde{\boldsymbol{p}}^\mathsf{T}_{s,a}\hat{\boldsymbol{w}}_{s,a}$. Then the asymptotic performance of the $\mathrm{VaR}$ framework $\hat{\rho}$ w.r.t. the optimal percentile returns $\rho^*$ satisfies*

$$\lim_{N\to\infty} \sqrt{N}(\rho^* - \hat{\rho}) \leq \frac{1}{1-\gamma}\left(2\Phi^{-1}(1-\delta/S)\sigma_{\max}\right) \leq \frac{1}{1-\gamma}\sqrt{8\ln(S/\delta)}\sigma_{\max} \ .$$

Theorem 3.5 shows that the asymptotic loss in performance of the VaR framework convergence to 0, i.e., $\lim_{N\to\infty}(\rho^* - \hat{\rho}) = 0$.

### 3.2 Dynamic Programming Algorithm

We provide a detailed description of the VaR value iteration algorithm (Algorithm 3.1) below. We also bound the number of samples required to estimate a single VaR Bellman update $(\mathcal{T}^\pi_{\mathrm{VaR}_\alpha}\boldsymbol{v})_s$ for any given policy $\pi$ and state $s$ with high confidence $1 - \zeta$.

---

**Algorithm 3.1:** Generalized VaR Value Iteration Algorithm

---

**Input:** Confidence $\alpha \in (0, 1/2)$, Value approximation error $\varepsilon \geq 0$, Transition models
$\quad\quad \tilde{\boldsymbol{P}}(\omega_1), \tilde{\boldsymbol{P}}(\omega_2), \ldots, \tilde{\boldsymbol{P}}(\omega_M)$ sampled from posterior distribution $f$
**Output:** Robust policy $\pi_k$, Lower bound $\boldsymbol{u}_k$

1  Initialize robust value-function $\boldsymbol{u}_0 = \boldsymbol{0}, k = 0$
2  **repeat**
3  $\quad$ **for** $s \leftarrow 1$ **to** $S$ **do**
4  $\quad\quad$ **for** $a \leftarrow 1$ **to** $A$ **do**
5  $\quad\quad\quad \boldsymbol{w}_{s,a} \leftarrow \boldsymbol{r}_{s,a} + \gamma\boldsymbol{u}_k$ $\quad\quad\quad\quad\quad\quad\quad\quad\quad\quad\quad\quad\quad$ // 1-step return from $(s,a)$
6  $\quad\quad\quad \boldsymbol{q}_a \leftarrow \widehat{\mathrm{VaR}}_\alpha[\tilde{\boldsymbol{p}}^\mathsf{T}_{s,a}\boldsymbol{w}_{s,a}]$ $\quad\quad\quad\quad\quad\quad\quad\quad\quad\quad\quad$ // empirical $\mathrm{VaR}_\alpha$
7  $\quad\quad$ **end**
8  $\quad$ **end**
9  $\quad k \leftarrow k + 1$
10 $\quad \boldsymbol{u}_k(s) \leftarrow \max_{a\in\mathcal{A}}(\boldsymbol{q}_a), \quad \pi_k(s) \leftarrow \arg\max_{a\in\mathcal{A}}(\boldsymbol{q}_a)$ $\quad\quad$ // $\mathrm{VaR}_\alpha$ Bellman optimality update
11 **until** $\|\boldsymbol{u}_k - \boldsymbol{u}_{k-1}\|_\infty \leq \varepsilon(1-\gamma)/\gamma$
12 **return** $\pi_k, \boldsymbol{u}_k$

---

In each iteration of Algorithm 3.1, we compute the one-step VaR Bellman update $\mathcal{T}_{\mathrm{VaR}_\alpha}(\boldsymbol{v})$ using the current value function $\boldsymbol{v}$. When $\tilde{\boldsymbol{P}}$ is not normally distributed, we use the Quick Select algorithm [23] to efficiently compute the empirical estimate of the $\alpha$-percentile of 1-step returns for any state $s$, action $a$, and value function $\boldsymbol{v}$, i.e., $\widehat{\mathrm{VaR}}_\alpha[\tilde{\boldsymbol{p}}^\mathsf{T}_{s,a}(\boldsymbol{r}_{s,a} + \gamma\boldsymbol{v})]$. On the other hand, when $\tilde{\boldsymbol{P}}$ is normally distributed, we compute the VaR Bellman optimality update $\mathcal{T}_{\mathrm{VaR}_\alpha}(\boldsymbol{v})$ using the empirical estimate of mean $(\bar{\boldsymbol{p}}_{s,a})_{s\in\mathcal{S},a\in\mathcal{A}}$ and covariance $(\boldsymbol{\Sigma}_{s,a})_{s\in\mathcal{S},a\in\mathcal{A}}$ of the transition probabilities derived from the data $\mathcal{D}$ (Proposition 3.3). We repeat these steps until convergence.

**Proposition 3.6** (Empirical Error Bound). *For any state $s$, action $a$ and value function $\boldsymbol{v}$, let $\widehat{\mathrm{VaR}}_\alpha[\tilde{\boldsymbol{p}}^\mathsf{T}_{s,a}\boldsymbol{w}_{s,a}]$ represent the empirical estimate of $\alpha$-percentile of returns $\mathrm{VaR}_\alpha[\tilde{\boldsymbol{p}}^\mathsf{T}_{s,a}\boldsymbol{w}_{s,a}]$ and $\Phi_f$ represent the cumulative density function (CDF) of the random estimate of returns $\tilde{\boldsymbol{p}}^\mathsf{T}_{s,a}\boldsymbol{w}_{s,a}$. Suppose that $\Phi_f$ is differentiable at the point $\mathrm{VaR}_\alpha[\tilde{\boldsymbol{p}}^\mathsf{T}_{s,a}\boldsymbol{w}_{s,a}]$ and let $\eta = \Phi'_f\left(\mathrm{VaR}_\alpha[\tilde{\boldsymbol{p}}^\mathsf{T}_{s,a}\boldsymbol{w}_{s,a}]\right)$ represents the density of estimate of returns at point $\mathrm{VaR}_\alpha[\tilde{\boldsymbol{p}}^\mathsf{T}_{s,a}\boldsymbol{w}_{s,a}]$. Let $M^*$ be the number of posterior samples required to obtain empirical error $\varepsilon \in \mathbb{R}$, with confidence $1 - \zeta$, where $0 < \zeta < 1$, i.e., $\Pr\left[|\widehat{\mathrm{VaR}}_\alpha[\tilde{\boldsymbol{p}}^\mathsf{T}_{s,a}\boldsymbol{w}_{s,a}] - \mathrm{VaR}_\alpha[\tilde{\boldsymbol{p}}^\mathsf{T}_{s,a}\boldsymbol{w}_{s,a}]| > \varepsilon\right] \leq \zeta$. Then, $\lim_{\varepsilon\to 0} M^*\varepsilon^2 = \ln(2/\varsigma)/2\eta^2$.*

We now show that Algorithm 3.1 produces a policy $\pi_k$ and value function $\boldsymbol{u}_k$ that approximates the optimal value function $\hat{\boldsymbol{v}}$.

**Proposition 3.7** (Value Iteration Error). *Define the empirical $\mathrm{VaR}$ Bellman optimality operator $\mathcal{T}_{\widehat{\mathrm{VaR}}_\alpha}$ for any value $\boldsymbol{v} \in \mathbb{R}^S$ and state $s \in \mathcal{S}$ as $(\mathcal{T}_{\widehat{\mathrm{VaR}}_\alpha}\boldsymbol{v})_s = \max_{a\in\mathcal{A}} \widehat{\mathrm{VaR}}_\alpha\left[\tilde{\boldsymbol{p}}^\mathsf{T}_{s,a}\boldsymbol{w}_{s,a}\right]$. Let*

$\hat{\boldsymbol{v}} \in \mathbb{R}^S$ and $\hat{\boldsymbol{u}} \in \mathbb{R}^S$ *represent the fixed points of* $\mathcal{T}_{\mathrm{VaR}_\alpha}$ *and* $\mathcal{T}_{\widehat{\mathrm{VaR}}_\alpha}$ *respectively. Suppose that Algorithm 3.1 returns policy* $\pi_k$ *and value function* $\boldsymbol{u}_k$ *and* $\|\hat{\boldsymbol{u}} - \boldsymbol{u}_0\|_\infty \leq {}^{r_{\max}}\!/_{1-\gamma}$. *Then,*

$$\|\hat{\boldsymbol{u}} - \boldsymbol{u}_k\|_\infty \leq \varepsilon, \text{ and } \|\hat{\boldsymbol{u}} - \boldsymbol{u}_{\pi_k}\|_\infty \leq \frac{2\varepsilon\gamma}{1-\gamma} \ .$$

*Furthermore, the gap between the empirical and true value function is bounded by*

$$\|\hat{\boldsymbol{v}} - \hat{\boldsymbol{u}}\|_\infty \leq \frac{1}{1-\gamma} \min\left( \|\mathcal{T}_{\widehat{\mathrm{VaR}}_\alpha} \hat{\boldsymbol{u}} - \mathcal{T}_{\mathrm{VaR}_\alpha} \hat{\boldsymbol{u}}\|_\infty, \|\mathcal{T}_{\widehat{\mathrm{VaR}}_\alpha} \hat{\boldsymbol{v}} - \mathcal{T}_{\mathrm{VaR}_\alpha} \hat{\boldsymbol{v}}\|_\infty \right) \ .$$

**Proposition 3.8** (Time Complexity)**.** *Let* $r_{\max} = \max_{s,s' \in \mathcal{S}, a \in \mathcal{A}} |R(s,a,s')|$. *Then, Algorithm 3.1 terminates in* $k = \lceil \log_{1/\gamma}\left(\frac{r_{max}}{\varepsilon(1-\gamma)}\right) \rceil$ *iterations with time complexity* $\mathcal{O}\left( S^2 AM \log_{1/\gamma}\left(\frac{r_{max}}{\varepsilon(1-\gamma)}\right) \right)$.

*Furthermore, for any failure probability* $\zeta \in (0,1)$, *suppose that the CDF* ($\Phi_f$) *of the random estimate of 1-step returns* $\tilde{\boldsymbol{p}}_{s,a}^\mathsf{T} \hat{\boldsymbol{w}}_{s,a}$ *is differentiable at the point* $\mathrm{VaR}_\alpha[\tilde{\boldsymbol{p}}_{s,a}^\mathsf{T} \hat{\boldsymbol{w}}_{s,a}]$, *and set* $\eta = \Phi'(\mathrm{VaR}_\alpha[\tilde{\boldsymbol{p}}_{s,a}^\mathsf{T} \hat{\boldsymbol{w}}_{s,a}])$ *and* $M = \mathcal{O}\left( \frac{\log(S/\zeta)}{\eta^2 \varepsilon^2 (1-\gamma)^2} \right)$. *Then with probability at least* $1 - \zeta$, *it holds that* $\|\hat{\boldsymbol{v}} - \boldsymbol{u}_k\|_\infty \leq \mathcal{O}(\varepsilon)$, *and Algorithm 3.1 runs in* $\mathcal{O}\left( \frac{S^2 A \log_{1/\gamma}\left(\frac{r_{max}}{\varepsilon(1-\gamma)}\right) \log\left(\frac{S}{\zeta}\right)}{\eta^2 \varepsilon^2 (1-\gamma)^2} \right)$ *time.*

## 4 Comparison with Bayesian Credible Regions

We are now ready to answer the question: *Are Bayesian credible regions the optimal ambiguity sets for optimizing the percentile criterion?* For this, we compare the VaR framework with BCR Robust MDPs. First, we derive the robust form of the VaR framework and show that in contrast to the BCR Bellman operator, the VaR Bellman optimality operator implicitly constructs value function dependent ambiguity sets, and thus, these sets tend to be smaller (Proposition 4.1). Then, we compare the asymptotic radii of the BCR ambiguity sets and the VaR ambiguity sets implicitly constructed by $\mathcal{T}_{\mathrm{VaR}_\alpha}$. For any given confidence level $\alpha$, the radius of the VaR ambiguity sets are asymptotically smaller than that of BCR ambiguity sets (Theorem 4.3). Precisely, the ratio of the radii of VaR ambiguity sets to BCR ambiguity sets is at least $\sqrt{\chi^2_{S-1,1-\alpha}}/\Phi^{-1}(1-\alpha)$, where $\chi^2_{S-1,1-\alpha}$ is the CDF inverse of $1-\alpha$ percentile of Chi-squared distribution with degree of freedom $S-1$ and $\Phi^{-1}(1-\alpha)$ is the $1-\alpha$ percentile of $\mathcal{N}(0,1)$. This implies that there exist directions in which the BCR ambiguity sets are at least $\Omega(\sqrt{S})$ larger than VaR ambiguity sets. Thus, we prove that VaR framework is better suited for optimizing the percentile criterion than RMDPs with BCR ambiguity sets.

For any value function $\boldsymbol{v}$, define the VaR ambiguity set $\mathcal{P}^{\mathrm{VaR},\boldsymbol{v}}$ as

$$\mathcal{P}^{\mathrm{VaR},\boldsymbol{v}} = \bigtimes_{s \in \mathcal{S}, a \in \mathcal{A}} \mathcal{P}^{\mathrm{VaR},\boldsymbol{v}}_{s,a} \quad \text{where} \quad \mathcal{P}^{\mathrm{VaR},\boldsymbol{v}}_{s,a} = \left\{ \boldsymbol{p}_{s,a} \in \Delta^{\mathcal{S}} \mid \boldsymbol{p}_{s,a}^\mathsf{T} \boldsymbol{v} \geq \mathrm{VaR}_\alpha\left[ \tilde{\boldsymbol{p}}_{s,a}^\mathsf{T} \boldsymbol{v} \right] \right\}. \quad (7)$$

**Proposition 4.1** (Equivalence)**.** *The* VaR *Bellman optimality operator* $\mathcal{T}_{\mathrm{VaR}_\alpha}$ *can be expressed as*

$$\forall s \in \mathcal{S} : (\mathcal{T}_{\mathrm{VaR}_\alpha} \boldsymbol{v})_s = \max_{a \in \mathcal{A}} \min_{\boldsymbol{p} \in \mathcal{P}^{\mathrm{VaR},\boldsymbol{v}}_{s,a}} \boldsymbol{p}^\mathsf{T}(\boldsymbol{r}_{s,a} + \gamma \boldsymbol{v}) \ .$$

*Furthermore, the optimal VaR policy* $\hat{\pi} \in \Pi^D$ *solves* $\max_{\pi \in \Pi^D} \min_{\boldsymbol{P} \in \mathcal{P}^{\mathrm{VaR},\hat{\boldsymbol{v}}}} \rho(\pi, \boldsymbol{P})$ , *where* $\hat{\boldsymbol{v}} \in \mathbb{R}^S$ *is the fixed point of the* VaR *Bellman operator* $\mathcal{T}_{\mathrm{VaR}_\alpha}$, *i.e.,* $\hat{\boldsymbol{v}} = (\mathcal{T}_{\mathrm{VaR}_\alpha} \hat{\boldsymbol{v}})$.

Proposition 4.1 shows that the VaR Bellman optimality operator optimizes a unique robust MDP whose ambiguity sets are SA-rectangular and value function dependent. Notice that for any state $s$, action $a$ and value function $\boldsymbol{v}$, the VaR ambiguity set is a half-space $\left\{ \boldsymbol{p}_{s,a} \in \Delta^S : \boldsymbol{p}_{s,a}^\mathsf{T} \boldsymbol{v} \geq \mathrm{VaR}_\alpha\left[ \tilde{\boldsymbol{p}}_{s,a}^\mathsf{T} \boldsymbol{v} \right] \right\}$ dependent on the value function $\boldsymbol{v}$. In contrast, BCR ambiguity sets are independent of any policy or value function and are constructed such that they provide high-confidence guarantees on returns of all policies simultaneously. As a result, BCR ambiguity sets tend to be unnecessarily large.

We now compute the ratio of the asymptotic radii of BCR ambiguity sets and VaR ambiguity sets.

**Theorem 4.2** (Asymptotic Radii of VaR Ambiguity Sets)**.** *For any state* $s$ *and action* $a$, *let* $I(\{(\boldsymbol{p}^*_{s,a})_i\}_{i=1}^{S-1})$ *be the Fisher information of the first* $S-1$ *transition probabilities, i.e.,* $\{(\boldsymbol{p}^*_{s,a})_i\}_{i=1}^{S-1}$.

*Define* $I'(\boldsymbol{p}_{s,a}^*) = \begin{bmatrix} I(\{(\boldsymbol{p}_{s,a}^*)_i\}_{i=1}^{S-1}) & \mathbf{0} \\ \mathbf{0}^\mathsf{T} & 0 \end{bmatrix}$. *Suppose that the normality assumptions on the posterior distribution* $\tilde{\boldsymbol{P}}$ *in Section 2 are satisfied. Then,*

$$\lim_{N \to \infty} \sqrt{N}(\mathcal{P}_{s,a}^{\mathrm{VaR}_\alpha} - \boldsymbol{p}_{s,a}^*) = \left\{ \boldsymbol{p}_{s,a} \in \Delta^\mathcal{S} \,\middle|\, \|\boldsymbol{p}_{s,a} - \boldsymbol{p}_{s,a}^*\|_{I'(\boldsymbol{p}_{s,a}^*)^{-1}} \leq \Phi^{-1}(1-\alpha) \right\} - \boldsymbol{p}_{s,a}^* \ . \quad (8)$$

Theorem 4.2 shows that the asymptotic form of the VaR ambiguity set is an ellipsoid. Note that, in contrast to the finite-sample VaR ambiguity set $\mathcal{P}_{s,a}^{\mathrm{VaR},\boldsymbol{v}}$ in problem (7), the asymptotic VaR ambiguity set $\mathcal{P}_{s,a}^{\mathrm{VaR}}$ is independent of the value function $\boldsymbol{v}$. This is because $\mathrm{VaR}_\alpha\left[\tilde{\boldsymbol{p}}_{s,a}^\mathsf{T}\boldsymbol{v}\right]$ is convex in the value function $\boldsymbol{v}$ [39]. Therefore, the asymptotic ambiguity set $\mathcal{P}_{s,a}^{\mathrm{VaR}}$ is simply the intersection of closed half-spaces in $\mathcal{P}_{s,a}^{\mathrm{VaR},\boldsymbol{v}}$, computed over all value functions $\boldsymbol{v} \in \mathbb{R}^\mathcal{S}$ [9]. It is also worth noting that the radius of the asymptotic VaR ambiguity set $\mathcal{P}_{s,a}^{\mathrm{VaR}}$ is constant. In contrast, the asymptotic radius of the BCR ambiguity sets grows with the number of states in at least one direction, as we show in the following proposition.

**Theorem 4.3** (Asymptotic Radius of Bayesian Credible Regions)**.** *For any state $s$ and action $a$, let $\mathcal{P}_{s,a}^{BCR_\alpha}$ represent any Bayesian credible region. Let $\xi < \sqrt{\chi_{S-1,1-\alpha}^2}/\Phi^{-1}(1-\alpha)$. Suppose that the normality assumptions on the posterior distribution of $\tilde{\boldsymbol{P}}$ in Section 2 are satisfied. Then,*

$$\forall s \in \mathcal{S}, a \in \mathcal{A}: \lim_{N \to \infty} \sqrt{N}(\mathcal{P}_{s,a}^{BCR_\alpha} - \boldsymbol{p}_{s,a}^*) \not\subseteq \lim_{N \to \infty} \sqrt{N}\xi(\mathcal{P}_{s,a}^{\mathrm{VaR}_\alpha} - \boldsymbol{p}_{s,a}^*) \ . \quad (9)$$

We note that Theorem 4.3 is an adaptation of Theorem 10 in [21] in RL which proves that there exist directions in which the Bayesian credible regions is at least $\xi = \sqrt{\chi_{S-1,1-\alpha}^2}/\Phi^{-1}(1-\alpha)$ larger than VaR ambiguity sets. Since the value of $\xi$ only grows with the number of states, we conclude that BCR ambiguity sets are sub-optimal for optimizing the percentile criterion. Figure 1a shows the growth in the ratio of radius of BCR to VaR ambiguity sets with an increasing number of states.

## 5 Experiments

We now empirically analyze the robustness of the VaR framework in three different domains.

*Riverswim:* The Riverswim MDP [46] consists of five states and two actions. The state represents the coordinates of the swimmer in the river and action represents the direction of the swim. The task of the agent is to learn a policy that would take the swimmer to the other end of the river.

*Population Growth Model:* The Population Growth MDP [25] models the population growth of pests and consists of 50 states and 5 actions. The states represent the pest population and actions represent the pest control measures. In our experiments, we use two different instantiations of the Population Growth Model: Population-Small and Population, which vary in the number of posterior samples.

*Inventory Management:* The Inventory Management MDP [56] models the classical inventory management problem and consists of 30 states and 30 actions. States represent the inventory level and actions represent the inventory to be purchased. The sale price $s$, holding cost $c$ and purchase costs $\boldsymbol{P}$ are 3.99, 0.03, and 2.219. The demand is normally distributed with mean=$s/4$ and standard deviation $s/6$.

**Implementation details:** For each domain in our experiments, we sample a dataset $\mathcal{D}$ consisting of $n$ tuples of the form $\{s, a, r, s'\}$, corresponding to the state $s$, the action taken $a$, the reward $r$ and the next state $s'$. We construct a posterior distribution over the models using $\mathcal{D}$, assuming Dirichlet priors over the model parameters. Using MCMC sampling, we construct two datasets $\mathcal{D}_1$ and $\mathcal{D}_2$ containing $M$ and $K$ transition probability models, respectively.

We construct $L$ train datasets by randomly sampling $80\%$ of the models from $\mathcal{D}_1$ each time. We use $\mathcal{D}_2$ as our test dataset. For any confidence level $\delta$, we train one RL agent per train dataset and method.

For evaluation, we consider two instances of the VaR framework: one (denoted by *VaRN*) that assumes that $\tilde{\boldsymbol{P}}$ is a multivariate normal, and another (denoted by *VaR*) that does not assume any structure over $\tilde{\boldsymbol{P}}$. We use seven baseline methods for evaluating the robustness of our framework. They are: Naive Hoeffding [35], Optimized Hoeffding (Opt Hoeffding) [3], Soft-Robust [5], and BCR Robust MDPs with weighted $\ell_1$ ambiguity sets (*WBCR $\ell_1$*) [3], weighted $\ell_\infty$ ambiguity sets

(*WBCR* $\ell_\infty$) [3], unweighted $\ell_1$ ambiguity sets (*BCR* $\ell_1$) [40] and unweighted $\ell_\infty$ ambiguity sets (*BCR* $\ell_1$) ambiguity [40] sets. See Appendix C in the appendix for more details.

We report the 95% confidence interval of the robust performance ($\delta$-percentile of expected returns) of the VaR framework on the test dataset with that of other baselines for different values of $\delta$.

| Methods | Riverswim | Inventory | Population-Small | Population |
|---|---|---|---|---|
| VaR | **68.54 ± 5.08** | 457.95 ± 0.74 | **-3102.48 ± 429.7** | -4576.87 ± 147.3 |
| VaRN | 67.27 ± 0.0 | 452.78 ± 0.02 | -4005.53 ± 8.76 | **-4570.17 ± 38.84** |
| BCR $l_1$ | 67.27 ± 0.0 | 369.67 ± 0.0 | -5614.95 ± 80.28 | -6013.21 ± 1177.94 |
| BCR $l_\infty$ | 67.27 ± 0.0 | 199.41 ± 39.02 | -7908.92 ± 41.6 | -9033.7 ± 84.28 |
| WBCR $l_1$ | 67.9 ± 3.82 | 454.1 ± 4.16 | -5290.38 ± 1084.26 | -5408.01 ± 225.2 |
| WBCR $l_\infty$ | 67.27 ± 0.0 | 199.4 ± 39.02 | -7712.43 ± 55.96 | -8377.64 ± 126.24 |
| Soft-Robust | 61.79 ± 1.46 | **460.6 ± 0.0** | -3647.18 ± 94.62 | -6932.86 ± 154.16 |
| Naive Hoeffding | 51.52 ± 6.06 | -0.0 ± 0.0 | -8647.7 ± 59.5 | -9127.14 ± 140.98 |
| Opt Hoeffding | 50.76 ± 4.56 | -0.0 ± 0.0 | -8640.48 ± 2.34 | -9163.64 ± 13.62 |

Table 1: shows the 95% confidence interval of the robust (percentile) returns achieved by *VaR*, *VaRN*, *BCR* $\ell_1$, *BCR* $\ell_\infty$, *WBCR* $\ell_1$, *WBCR*, *Soft Robust*, *Worst RMDP*, *Naive Hoeffding* and *Opt Hoeffding* agents at $\delta = 0.05$ in Riverswim, Inventory, Population-Small, and Population domain.

**Experimental Results**    Table 1 summarizes the performance of the VaR framework and the baselines for confidence level $\delta = 0.05$ (Table 2 and Table 3 in the appendix summarizes the results for $\delta = 0.15$ and $\delta = 0.3$ respectively.). We observe that for confidence level $\delta = 0.05$, the VaR framework outperforms the baseline methods in terms of mean robust performance in most domains. On the other hand, for $\delta = 0.15$, the VaR framework outperforms baselines in the Population and Population-Small domains and has comparable performance to the Soft-Robust method in the Inventory domain. However, at $\delta = 0.3$ we observe that the VaR framework has lower robust performance relative to the the Soft-Robust method in Riverswim and Population domains. We conjecture that this is because the Soft-Robust method optimizes the policy to maximize the mean of expected returns and is therefore able to perform better in cases where a lower level of robustness is required. However, we note that in contrast to our method, the Soft-Robust method does not provide probabilistic guarantees against worst-case scenarios.

Furthermore, as expected, we find that in many cases, the robust performance of BCR Robust MDPs with span-optimized (weighted) ambiguity sets (*WBCR* $\ell_1$, WBCR $\ell_\infty$) is relatively higher than the robust performance of Robust MDPs with unweighted BCR ambiguity sets (*BCR* $\ell_1$, BCR $\ell_\infty$). However, we find that even Robust MDPs with span-optimized BCR ambiguity sets are generally unable to outperform the robust performance of our $\text{VaR}_\alpha$ framework.

Figure 2 compares the robust performance of the VaR framework and the baselines on both train and test models. The trends in the robust performance of the VaR framework and the baselines are similar on both train and test models.

## 6   Conclusion and Future Work

The main limitation of the VaR framework is that it does not consider the correlations in the uncertainty of transition probabilities across states and actions [19, 29, 30]. However, due to the non-convex nature of the percentile-criterion [3, 40], constructing a tractable $\text{VaR}_\alpha$ Bellman operator that considers these correlations is not feasible. One plausible solution is to use a Conditional Value at Risk Bellman operator [11, 27] which is convex and lower bounds the Value at Risk measure. We leave the analysis of this approach for future work. Empirical analysis of the $\text{VaR}_\alpha$ framework in domains with continuous state-action spaces is also an interesting avenue for future work.

In conclusion, we propose a novel dynamic programming algorithm that optimizes a tight lower-bound approximation on the percentile criterion without explicitly constructing ambiguity sets. We theoretically show that our algorithm implicitly constructs tight ambiguity sets whose asymptotic radius is smaller than that of any Bayesian credible region, and therefore, computes less conservative policies with the same confidence guarantees on returns. We also derive finite-sample and asymptotic bounds on the performance loss due to our approximation. Finally, our experimental results demonstrate the efficacy of our method in several domains.

## Acknowledgments

The work in the paper was supported, in part, by NSF grants 2144601 and 1815275.

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

# A  Additional theoretical results

**Definition A.1** (Translation Subvariance)**.** The operator $\mathfrak{T} : \mathbb{R}^S \to \mathbb{R}^S$ satisfies the *translation subvariance* property if for all vector $\boldsymbol{v} \in \mathbb{R}^S$ and scalar $c$, there exists $\gamma \in [0, 1)$ that satisfies

$$\mathfrak{T}(\boldsymbol{v} + c\boldsymbol{1}) = (\mathfrak{T}\boldsymbol{v}) + \gamma c\boldsymbol{1} \quad .$$

.

**Definition A.2** (Monotonicity)**.** The operator $\mathfrak{T} : \mathbb{R}^S \to \mathbb{R}^S$ satisfies the *monotonicity* property if for all $\boldsymbol{v} \in \mathbb{R}^S$ and $\boldsymbol{u} \in \mathbb{R}^S$ such that $\boldsymbol{v} \preceq \boldsymbol{u}$, $\mathfrak{T}$ satisfies

$$\mathfrak{T}\boldsymbol{v} \preceq \mathfrak{T}\boldsymbol{u} \quad .$$

**Lemma A.3** (Contraction Mapping [6])**.** *The operator* $\mathfrak{T} : \mathbb{R}^S \to \mathbb{R}^S$ *is a contraction mapping if it satisfies* monotonicity *and* translation subvariance *properties, i.e., there exists* $\gamma \in [0, 1)$ *such that for all* $\boldsymbol{u} \in \mathbb{R}^S$, $\boldsymbol{v} \in \mathbb{R}^S$, $\mathfrak{T}$ *satisfies*

$$\|\mathfrak{T}\boldsymbol{u} - \mathfrak{T}\boldsymbol{v}\|_\infty \le \gamma \|\boldsymbol{u} - \boldsymbol{v}\|_\infty \quad .$$

The proof of Lemma A.3 follows directly from Proposition 2.1.3 in [6]. We re-derive the proof for the sake of completeness.

*Proof.* Denote

$$c = \max_{s \in S} |\boldsymbol{u}_s - \boldsymbol{v}_s| \quad .$$

Therefore for all $s \in \mathcal{S}$,

$$\boldsymbol{u}_s - c \le \boldsymbol{v}_s \le \boldsymbol{u}_s + c \quad .$$

Applying $\mathfrak{T}$ to these inequalities and using the *translation subvariance* (Definition A.1) and *monotonicity* (Definition A.2) properties, we obtain that for all $s \in \mathcal{S}$,

$$(\mathfrak{T}\boldsymbol{u})_s - \gamma c \le (\mathfrak{T}\boldsymbol{v})_s \le (\mathfrak{T}\boldsymbol{u})_s + \gamma c \quad .$$

It follows that for all $s \in \mathcal{S}$,

$$|(\mathfrak{T}\boldsymbol{v})_s - (\mathfrak{T}\boldsymbol{u})_s| \le \gamma c \quad ,$$

and therefore $\|\mathfrak{T}\boldsymbol{u} - \mathfrak{T}\boldsymbol{v}\|_\infty \le \gamma c$ , proving the stated result.

$\square$

**Lemma A.4.** *For any policy* $\pi$, *let* $\hat{\boldsymbol{v}}^\pi$ *and* $\boldsymbol{v}^\pi$ *be the fixed point of the* VaR *policy evaluation operator* $\mathcal{T}^\pi_{\mathrm{VaR}_\alpha}$ *and Bellman policy evaluation operator* $\mathcal{T}^\pi_P$. *If the VaR Bellman policy evaluation operator* $\mathcal{T}^\pi_{\mathrm{VaR}_\alpha}$ *dominates the Bellman policy evaluation operator* $\mathcal{T}^\pi_P$ *at* $\hat{\boldsymbol{v}}^\pi$, *i.e.,* $\mathcal{T}^\pi_{\mathrm{VaR}_\alpha} \hat{\boldsymbol{v}}^\pi \preceq \mathcal{T}^\pi_P \hat{\boldsymbol{v}}^\pi$, *then, the fixed point of the* VaR *Bellman evaluation operator* $\mathcal{T}^\pi_{\mathrm{VaR}_\alpha}$ *dominates the fixed point of the Bellman evaluation operator* $\mathcal{T}^\pi_P$, *i.e.,* $\hat{\boldsymbol{v}}^\pi \preceq \boldsymbol{v}^\pi$.

We note that in contrast to the Bellman policy evaluation operator $\mathcal{T}^\pi_P$, the VaR Bellman policy evaluation operator $\mathcal{T}^\pi_{\mathrm{VaR}_\alpha}$ is a function of the random variable $\tilde{\boldsymbol{P}}$ and is not dependent on the transition probabilities $\boldsymbol{P}$ assumed in this setting.

Using the assumption $\mathcal{T}_{\mathrm{VaR}_\alpha} \hat{\boldsymbol{v}}^\pi \preceq \mathcal{T}^\pi_P \hat{\boldsymbol{v}}^\pi$, and from $\hat{\boldsymbol{v}}^\pi = \mathcal{T}^\pi_{\mathrm{VaR}_\alpha} \hat{\boldsymbol{v}}^\pi$ and $\boldsymbol{v}^\pi = \mathcal{T}^\pi_P \boldsymbol{v}^\pi$, we get by algebraic manipulations:

*Proof.*

$$\hat{\boldsymbol{v}}^\pi - \boldsymbol{v}^\pi = \mathcal{T}^\pi_{\mathrm{VaR}_\alpha} \hat{\boldsymbol{v}}^\pi - \mathcal{T}^\pi_P \boldsymbol{v}^\pi \preceq \mathcal{T}^\pi_P \hat{\boldsymbol{v}}^\pi - \mathcal{T}^\pi_P \boldsymbol{v}^\pi \preceq \gamma \boldsymbol{P}_\pi (\hat{\boldsymbol{v}}^\pi - \boldsymbol{v}^\pi) \quad .$$

Here $\boldsymbol{P}_\pi$ is the transition probability function corresponding to policy $\pi$. Subtracting $\gamma \boldsymbol{P}_\pi (\hat{\boldsymbol{v}}^\pi - \boldsymbol{v}^\pi)$ from the above inequality gives,

$$(\boldsymbol{I} - \gamma \boldsymbol{P}_\pi)(\hat{\boldsymbol{v}}^\pi - \boldsymbol{v}^\pi) \preceq \boldsymbol{0} \quad .$$

where $\boldsymbol{I}$ is the identity matrix. $(\boldsymbol{I} - \gamma \boldsymbol{P}_\pi)^{-1}$ is monotone as can be seen from its Neumann series.

$$\hat{\boldsymbol{v}}^\pi - \boldsymbol{v}^\pi \preceq (\boldsymbol{I} - \gamma \boldsymbol{P}_\pi)^{-1} \boldsymbol{0} = \boldsymbol{0} \quad .$$

which proves the result.

$\square$

# B  Proofs

## B.1  Proof of Proposition 3.1

**Proposition 3.1** (Validity). *The following properties of $\mathcal{T}_{\mathrm{VaR}_\alpha}$ hold for all value functions $\boldsymbol{u}, \boldsymbol{v} \in \mathbb{R}^{\mathcal{S}}$.*

1. *The operator $\mathcal{T}_{\mathrm{VaR}_\alpha}$ is contraction mapping on $\mathbb{R}^{\mathcal{S}}$: $\|\mathcal{T}_{\mathrm{VaR}_\alpha}\boldsymbol{u} - \mathcal{T}_{\mathrm{VaR}_\alpha}\boldsymbol{v}\|_\infty \leq \gamma \|\boldsymbol{u} - \boldsymbol{v}\|_\infty$ .*
2. *The operator $\mathcal{T}_{\mathrm{VaR}_\alpha}$ is monotone: $\boldsymbol{u} \succeq \boldsymbol{v} \Rightarrow \mathcal{T}_{\mathrm{VaR}_\alpha}\boldsymbol{u} \succeq \mathcal{T}_{\mathrm{VaR}_\alpha}\boldsymbol{v}$.*
3. *The equality $\mathcal{T}_{\mathrm{VaR}_\alpha}\hat{\boldsymbol{v}} = \hat{\boldsymbol{v}}$ has a unique solution.*

*Proof.* From Lemma A.3, we know that an operator is a contraction mapping if it satisfies *monotonicity* and *subvariance* property.

In this proof, we will show that the VaR Bellman operator $\mathcal{T}_{\mathrm{VaR}_\alpha}^\pi$ satisfies *monotonicity* and *subvariance* property, and therefore, is a contraction mapping.

We will use shorthand $\boldsymbol{r}_{s,a}$ to denote the reward vector corresponding to state $s$ and action $a$, i.e., $\boldsymbol{r}_{s,a} = R(s, a, \cdot)$.

First, we show that $\mathcal{T}_{\mathrm{VaR}_\alpha}$ satisfies *translation subvariance* property. Consider any $c \in \mathbb{R}$ and state $s$. Then,

$$
\begin{aligned}
(\mathcal{T}_{\mathrm{VaR}_\alpha}(\boldsymbol{v} + c\mathbf{1}))_s &= \max_{a \in \mathcal{A}} \mathrm{VaR}_\alpha[\tilde{\boldsymbol{p}}_{s,a}^\mathsf{T}(\boldsymbol{r}_{s,a} + \gamma(\boldsymbol{v} + c\mathbf{1}))] \\
&\stackrel{(a)}{=} \max_{a \in \mathcal{A}} \mathrm{VaR}_\alpha[\tilde{\boldsymbol{p}}_{s,a}^\mathsf{T}(\boldsymbol{r}_{s,a} + \gamma\boldsymbol{v} + \gamma c\mathbf{1})] \\
&\stackrel{(b)}{=} \max_{a \in \mathcal{A}} \mathrm{VaR}_\alpha[\tilde{\boldsymbol{p}}_{s,a}^\mathsf{T}(\boldsymbol{r}_{s,a} + \gamma\boldsymbol{v}) + \gamma c] \\
&\stackrel{(c)}{=} \max_{a \in \mathcal{A}} \mathrm{VaR}_\alpha[\tilde{\boldsymbol{p}}_{s,a}^\mathsf{T}(\boldsymbol{r}_{s,a} + \gamma\boldsymbol{v})] + \gamma c \\
&\stackrel{(d)}{=} (\mathcal{T}_{\mathrm{VaR}_\alpha}\boldsymbol{v})_s + \gamma c \ .
\end{aligned}
$$

$(a)$ follows from simple algebraic manipulations, $(b)$ follows from $\gamma c\bar{\boldsymbol{p}}_{s,a}^\mathsf{T}\mathbf{1} = \gamma c$, $(c)$ follows from the translational invariance property of $\mathrm{VaR}_\alpha$ measure [41], and $(d)$ follows the definition of the VaR Bellman operator $\mathcal{T}_{\mathrm{VaR}_\alpha}^\pi$.

Next, we show that $\mathcal{T}_{\mathrm{VaR}_\alpha}$ satisfies the *monotonicity* property.

Let $\boldsymbol{u}$ and $\boldsymbol{v}$ be any two value functions such that $\boldsymbol{v} \preceq \boldsymbol{u}$. Consider any state $s \in \mathcal{S}$. Then,

$$
\begin{aligned}
(\mathcal{T}_{\mathrm{VaR}_\alpha}\boldsymbol{v})_s - (\mathcal{T}_{\mathrm{VaR}_\alpha}\boldsymbol{u})_s &\stackrel{(a)}{=} \max_{a \in \mathcal{A}} \mathrm{VaR}_\alpha[\tilde{\boldsymbol{p}}_{s,a}^\mathsf{T}(\boldsymbol{r}_{s,a} + \gamma\boldsymbol{v})] - \max_{a \in \mathcal{A}} \mathrm{VaR}_\alpha[\tilde{\boldsymbol{p}}_{s,a}^\mathsf{T}(\boldsymbol{r}_{s,a} + \gamma\boldsymbol{u})] \\
&\stackrel{(b)}{\leq} \max_{a \in \mathcal{A}} \left(\mathrm{VaR}_\alpha[\tilde{\boldsymbol{p}}_{s,a}^\mathsf{T}(\boldsymbol{r}_{s,a} + \gamma\boldsymbol{v})] - \mathrm{VaR}_\alpha[\tilde{\boldsymbol{p}}_{s,a}^\mathsf{T}(\boldsymbol{r}_{s,a} + \gamma\boldsymbol{u})]\right) \\
&\stackrel{(c)}{\leq} 0 \\
(\mathcal{T}_{\mathrm{VaR}_\alpha}\boldsymbol{v})_s &\leq (\mathcal{T}_{\mathrm{VaR}_\alpha}\boldsymbol{u})_s \ .
\end{aligned}
$$

$(a)$ follows from the definition of the $\mathrm{VaR}_\alpha$ Bellman operator $\mathcal{T}_{\mathrm{VaR}_\alpha}$, $(b)$ follows from the fact that $\forall a' \in \mathcal{A}, -\max_{a \in \mathcal{A}} \mathrm{VaR}_\alpha[\tilde{\boldsymbol{p}}_{s,a}^\mathsf{T}(\boldsymbol{r}_{s,a} + \gamma\boldsymbol{v})] \leq -\mathrm{VaR}_\alpha[\boldsymbol{p}_{s,a'}^\mathsf{T}(\boldsymbol{r}_{s,a'} + \gamma\boldsymbol{v})]$, $(c)$ follows from the monotonicity property [41] of the VaR operator and $\boldsymbol{u} \succeq \boldsymbol{v}$.

Thus, we prove that $\mathcal{T}_{\mathrm{VaR}_\alpha}$ is a $\gamma$-contraction mapping and a monotone operator. Since $\mathcal{T}_{\mathrm{VaR}_\alpha}$ is a contraction operator on a Banach space and from the Banach fixed point theorem [1], it follows that the operator $\mathcal{T}_{\mathrm{VaR}_\alpha}$ has a unique solution $\hat{\boldsymbol{v}}$, i.e., $\mathcal{T}_{\mathrm{VaR}_\alpha}\hat{\boldsymbol{v}} = \hat{\boldsymbol{v}}$. □

Given a policy $\pi$, a state $s \in \mathcal{S}$, and transition probabilities $\boldsymbol{P}$, let

$$
(\mathcal{T}_{\boldsymbol{P}}^\pi\boldsymbol{v})_s = \sum_{a \in \mathcal{A}} \pi(s,a)\boldsymbol{p}_{s,a}^\mathsf{T}\boldsymbol{w}_{s,a} \text{ and } (\mathcal{T}_{\mathrm{VaR}_\alpha}^\pi\boldsymbol{v})_s = \sum_{a \in \mathcal{A}} \pi(s,a) \mathrm{VaR}_\alpha\left[\tilde{\boldsymbol{p}}_{s,a}^\mathsf{T}\boldsymbol{w}_{s,a}\right] \ ,
$$

represent the Bellman evaluation operator corresponding to transition probabilities $\boldsymbol{P}$ and the $\mathrm{VaR}_\alpha$ Bellman evaluation operator, respectively.

## B.2 Proof of Proposition 3.2

**Proposition 3.2** (Lower Bound Percentile Criterion). *For any $\delta \in (0, 1/2)$, if we set the confidence level $\alpha$ in the operator $\mathcal{T}_{\mathrm{VaR}_\alpha}^\pi$ to $\delta/S$, then for every policy $\pi \in \Pi : \Pr_{\tilde{P}} [\hat{v}^\pi \preceq \tilde{v}^\pi | \mathcal{D}] \geq 1 - \delta$.*

*Proof.* Let $\alpha = \delta/S$. Recall that for any policy $\pi$, state $s \in \mathcal{S}$, transition probabilities $P$, the Bellman evaluation operator $\mathcal{T}_P^\pi$ and the VaR Bellman evaluation operator $\mathcal{T}_{\mathrm{VaR}_\alpha}^\pi$ are defined as

$$(\mathcal{T}_P^\pi v)_s = \sum_{a \in \mathcal{A}} \pi(s,a) p_{s,a}^\mathsf{T} w_{s,a} \text{ and } (\mathcal{T}_{\mathrm{VaR}_\alpha}^\pi v)_s = \sum_{a \in \mathcal{A}} \pi(s,a) \mathrm{VaR}_\alpha \left[ \tilde{p}_{s,a}^\mathsf{T} w_{s,a} \right] \quad,$$

respectively.

Let $\hat{v}^\pi$ be the fixed point of $\mathcal{T}_{\mathrm{VaR}_\alpha}^\pi$ conditioned on observed transitions $\mathcal{D}$, and let $\tilde{v}^\pi$ be a random variable that represents the fixed point of $\mathcal{T}_{\tilde{P}}^\pi$ for a given realization of the transition probabilities $\tilde{P}$. Then, applying Lemma A.4 to $\mathcal{T}_{\mathrm{VaR}_\alpha}^\pi$ and $\mathcal{T}_{\tilde{P}}^\pi$, we have, $\hat{v}^\pi \preceq v^\pi$ implies

$$\mathcal{T}_{\mathrm{VaR}_\alpha}^\pi \hat{v}^\pi \preceq \mathcal{T}_{\tilde{P}}^\pi \hat{v}^\pi \quad.$$

That is for each state $s$,

$$\mathrm{VaR}_\alpha [\tilde{p}_{s,\pi(s)}^\mathsf{T} \hat{v}^\pi] \leq \tilde{p}_{s,\pi(s)}^\mathsf{T} \hat{v}^\pi \quad. \tag{10}$$

Using the equation (10), we can bound the probability that the VaR value function lower bounds the true value. $\qquad\square$

$$\Pr_{\tilde{P}} [\hat{v}^\pi \preceq \tilde{v}^\pi | \mathcal{D}] = \Pr_{\tilde{P}} \left[ \forall s \in \mathcal{S} : \mathrm{VaR}_\alpha \left[ \tilde{p}_{s,\pi(s)}^\mathsf{T} \hat{v}^\pi \right] \leq \tilde{p}_{s,\pi(s)}^\mathsf{T} \hat{v}^\pi \middle| \mathcal{D} \right] \quad. \tag{11}$$

From the definition of VaR, we know that for any state $s$ and action $a$,

$$\Pr_{\tilde{P}} \left[ \mathrm{VaR}_\alpha \left[ \tilde{p}_{s,a}^\mathsf{T} \hat{v}^\pi \right] \leq \tilde{p}_{s,a}^\mathsf{T} \hat{v}^\pi | \mathcal{D} \right] \geq 1 - \alpha \quad, \tag{12}$$

Therefore, using union bound and (11) in (12), we can write

$$\Pr_{\tilde{P}} [\hat{v}^\pi \succ \tilde{v}^\pi | \mathcal{D}] \leq \sum_{s \in \mathcal{S}} \Pr_{\tilde{P}} \left[ \mathrm{VaR}_\alpha [\tilde{p}_{s,\pi(s)}^\mathsf{T} \hat{v}^\pi] > \tilde{p}_{s,\pi(s)}^\mathsf{T} \hat{v}^\pi \middle| \mathcal{D} \right] \quad.$$

Thus,

$$\Pr [\hat{v}^\pi \succ \tilde{v}^\pi | \mathcal{D}] = \sum_{s \in \mathcal{S}} \frac{\delta}{S} = S \frac{\delta}{S} = \delta \quad.$$

## B.3 Proof of Proposition 3.3

**Proposition 3.3.** *Suppose that $\tilde{\boldsymbol{p}}_{s,a}$ for any state $s$ and action $a$, is a multivariate sub-Gaussian with mean $\bar{\boldsymbol{p}}_{s,a}$ and covariance factor $\boldsymbol{\Sigma}_{s,a}$, i.e., $\mathbb{E}\left[\exp\left(\lambda(\tilde{\boldsymbol{p}}_{\boldsymbol{s,a}} - \bar{\boldsymbol{p}}_{s,a})^{\mathsf{T}}\boldsymbol{w}\right)\right] \leq \exp\left(\lambda^2 \boldsymbol{w}^{\mathsf{T}}\boldsymbol{\Sigma}_{s,a}\boldsymbol{w}/2\right), \forall \lambda \in \mathbb{R}, \forall \boldsymbol{w} \in \mathbb{R}^S$. Then, for any state $s \in \mathcal{S}$, $\mathcal{T}_{\mathrm{VaR}_\alpha}$ satisfies*

$$(\mathcal{T}_{\mathrm{VaR}_\alpha}\boldsymbol{v})_s \geq \max_{a \in \mathcal{A}}\left(\bar{\boldsymbol{p}}_{s,a}^{\mathsf{T}}\boldsymbol{w}_{s,a} - \sqrt{2\ln(1/\alpha)}\sqrt{\boldsymbol{w}_{s,a}\boldsymbol{\Sigma}_{s,a}\boldsymbol{w}_{s,a}}\right) \ .$$

*As a special case, when $\tilde{\boldsymbol{p}}_{s,a}$ is normally distributed $\tilde{\boldsymbol{p}}_{s,a} \sim \mathcal{N}(\bar{\boldsymbol{p}}_{s,a}, \boldsymbol{\Sigma}_{s,a})$, then $\mathcal{T}_{\mathrm{VaR}_\alpha}$ for any state $s \in \mathcal{S}$ can be expressed as*

$$(\mathcal{T}_{\mathrm{VaR}_\alpha}\boldsymbol{v})_s = \max_{a \in \mathcal{A}}\left(\bar{\boldsymbol{p}}_{s,a}^{\mathsf{T}}\boldsymbol{w}_{s,a} - \Phi^{-1}(1-\alpha)\sqrt{\boldsymbol{w}_{s,a}\boldsymbol{\Sigma}_{s,a}\boldsymbol{w}_{s,a}}\right) \ .$$

*Proof.*

$$
\begin{aligned}
(\mathcal{T}_{\mathrm{VaR}_\alpha}^{\pi}\boldsymbol{v})_s &= \max_{a \in \mathcal{A}}\mathrm{VaR}_\alpha[\tilde{\boldsymbol{p}}_{s,a}^{\mathsf{T}}\boldsymbol{w}_{s,a}] \\
&\overset{(a)}{=} \max_{a \in \mathcal{A}}\sup\left\{t\,\middle|\,\Pr\left(\tilde{\boldsymbol{p}}_{s,a}^{\mathsf{T}}\boldsymbol{w}_{s,a} \geq t\right) \geq 1-\alpha\right\} \\
&\overset{(b)}{=} \max_{a \in \mathcal{A}}\inf\left\{t\,\middle|\,\Pr\left((\tilde{\boldsymbol{p}}_{s,a} - \bar{\boldsymbol{p}}_{s,a})^{\mathsf{T}}\boldsymbol{w}_{s,a} > (t - \bar{\boldsymbol{p}}_{s,a}^{\mathsf{T}}\boldsymbol{w}_{s,a})\right) < 1-\alpha\right\} \\
&\overset{(c)}{=} \max_{a \in \mathcal{A}}\inf\left\{t\,\middle|\,\Pr\left(\exp((\tilde{\boldsymbol{p}}_{s,a} - \bar{\boldsymbol{p}}_{s,a})^{\mathsf{T}}\boldsymbol{w}_{s,a}) > \exp(t - \bar{\boldsymbol{p}}_{s,a}^{\mathsf{T}}\boldsymbol{w}_{s,a})\right) < 1-\alpha\right\} \\
&\overset{(d)}{\geq} \max_{a \in \mathcal{A}}\inf\left\{t\,\middle|\,1 - \exp\left(\frac{-(t - \bar{\boldsymbol{p}}_{s,a}^{\mathsf{T}}\boldsymbol{w}_{s,a})^2}{2\boldsymbol{w}_{s,a}^{\mathsf{T}}\boldsymbol{\Sigma}_{s,a}\boldsymbol{w}_{s,a}}\right) < 1-\alpha\right\} \\
&\overset{(e)}{=} \max_{a \in \mathcal{A}}\inf\left\{t\,\middle|\,(t - \bar{\boldsymbol{p}}_{s,a}^{\mathsf{T}}\boldsymbol{w}_{s,a})^2 < -2\ln(\alpha)\boldsymbol{w}_{s,a}^{\mathsf{T}}\boldsymbol{\Sigma}_{s,a}\boldsymbol{w}_{s,a}\right\} \\
&\overset{(f)}{=} \max_{a \in \mathcal{A}}\inf\left\{t\,\middle|\,(t - \bar{\boldsymbol{p}}_{s,a}^{\mathsf{T}}\boldsymbol{w}_{s,a}) \in \left(-\sqrt{2\ln(1/\alpha)}\sqrt{\boldsymbol{w}_{s,a}^{\mathsf{T}}\boldsymbol{\Sigma}_{s,a}\boldsymbol{w}_{s,a}}, \sqrt{2\ln(1/\alpha)}\sqrt{\boldsymbol{w}_{s,a}^{\mathsf{T}}\boldsymbol{\Sigma}_{s,a}\boldsymbol{w}_{s,a}}\right)\right\} \\
&\overset{(g)}{=} \max_{a \in \mathcal{A}}\left(\bar{\boldsymbol{p}}_{s,a}^{\mathsf{T}}\boldsymbol{w}_{s,a} - \sqrt{2\ln(1/\alpha)}\sqrt{\boldsymbol{w}_{s,a}\boldsymbol{\Sigma}_{s,a}\boldsymbol{w}_{s,a}}\right) \ .
\end{aligned}
$$

Equality $(a)$ follows from the definition of VaR, $(b)$ follows from using the fact that $\mathrm{VaR}_\alpha[\tilde{X}] = \inf\{t \in \mathbb{R}: \Pr[\tilde{X} > t] < 1-\alpha\}$ and subtracting $\bar{\boldsymbol{p}}_{s,a}^{\mathsf{T}}\boldsymbol{w}_{s,a}$ on both sides, $(c)$ follows from taking exponential on both sides, $(d)$ follows from applying the upper-bound given by Chernoff bound for a sub-Gaussian distribution [7] i.e., $\Pr\left(\exp((\tilde{\boldsymbol{p}}_{s,a} - \bar{\boldsymbol{p}}_{s,a})^{\mathsf{T}}\boldsymbol{w}_{s,a}) \leq \exp(t - \bar{\boldsymbol{p}}_{s,a}^{\mathsf{T}}\boldsymbol{w}_{s,a})\right) \leq \exp\left(\frac{-(t - \bar{\boldsymbol{p}}_{s,a}^{\mathsf{T}}\boldsymbol{w}_{s,a})^2}{2\boldsymbol{w}_{s,a}^{\mathsf{T}}\boldsymbol{\Sigma}_{s,a}\boldsymbol{w}_{s,a}}\right)$, $(e)$ follows from subtracting 1 on both sides and then, taking $\ln$ on both sides, $(f)$ follows from simple algebraic manipulations and $(g)$ follows from taking the infimum of the solution interval of t.

Solving for $t$ in step $(g)$, we get $t = \mathrm{VaR}_\alpha[\tilde{\boldsymbol{p}}_{s,a}^{\mathsf{T}}\boldsymbol{w}_{s,a}] = \bar{\boldsymbol{p}}_{s,a}^{\mathsf{T}}\boldsymbol{w}_{s,a} - \sqrt{2\ln(1/\alpha)}\sqrt{\boldsymbol{w}_{s,a}\boldsymbol{\Sigma}_{s,a}\boldsymbol{w}_{s,a}}$ which proves the stated result.

Next, we derive the VaR Bellman optimality update $\mathcal{T}_{\mathrm{VaR}_\alpha}(\boldsymbol{v})$ when $\tilde{\boldsymbol{p}}_{s,a}\forall s \in \mathcal{S}, a \in \mathcal{A}$ is normally distributed.

Consider the VaR Bellman optimality operator defined for any state $s$ and value function $\boldsymbol{v}$ as

$$(\mathcal{T}_{\mathrm{VaR}_\alpha}\boldsymbol{v})_s = \max_{a \in \mathcal{A}}\mathrm{VaR}_\alpha\left[\tilde{\boldsymbol{p}}_{s,a}^{\mathsf{T}}\boldsymbol{w}_{s,a}\right] \ . \tag{13}$$

From the theory of multivariate Gaussian distributions [7], we know that, for any state $s$ and action $a$, if $\bar{\boldsymbol{p}}_{s,a}$ is Gaussian distributed $\mathcal{N}(\bar{\boldsymbol{p}}_{s,a}, \boldsymbol{\Sigma}_{s,a})$, then, $\tilde{\boldsymbol{p}}_{s,a}^{\mathsf{T}}\boldsymbol{w}_{s,a}$ is also Gaussian distributed $\mathcal{N}(\bar{\boldsymbol{p}}_{s,a}^{\mathsf{T}}\boldsymbol{w}_{s,a}, \boldsymbol{w}_{s,a}^{\mathsf{T}}\boldsymbol{\Sigma}_{s,a}\boldsymbol{w}_{s,a})$. To find the $\mathrm{VaR}_\alpha[\tilde{\boldsymbol{p}}_{s,a}^{\mathsf{T}}\boldsymbol{w}_{s,a}]$ for any state $s$ and action $a$, it is sufficient

to find $t$ such that $\Pr(\tilde{\boldsymbol{p}}_{s,a}^\mathsf{T}\boldsymbol{w}_{s,a} > t) = 1 - \alpha$.

$$\Pr\left(\frac{(\tilde{\boldsymbol{p}} - \bar{\boldsymbol{p}}_{s,a})^\mathsf{T}\boldsymbol{w}_{s,a}}{\sqrt{\boldsymbol{w}_{s,a}^\mathsf{T}\boldsymbol{\Sigma}_{s,a}\boldsymbol{w}_{s,a}}} > \frac{t - \bar{\boldsymbol{p}}_{s,a}^\mathsf{T}\boldsymbol{w}_{s,a}}{\sqrt{\boldsymbol{w}_{s,a}^\mathsf{T}\boldsymbol{\Sigma}_{s,a}\boldsymbol{w}_{s,a}}}\right) = 1 - \alpha$$

$$1 - \Phi\left(\frac{t - \bar{\boldsymbol{p}}_{s,a}^\mathsf{T}\boldsymbol{w}_{s,a}}{\sqrt{\boldsymbol{w}_{s,a}^\mathsf{T}\boldsymbol{\Sigma}_{s,a}\boldsymbol{w}_{s,a}}}\right) = 1 - \alpha$$

$$\left(\frac{t - \bar{\boldsymbol{p}}_{s,a}^\mathsf{T}\boldsymbol{w}_{s,a}}{\sqrt{\boldsymbol{w}_{s,a}^\mathsf{T}\boldsymbol{\Sigma}_{s,a}\boldsymbol{w}_{s,a}}}\right) = \Phi^{-1}(\alpha)$$

$$t = \bar{\boldsymbol{p}}_{s,a}^\mathsf{T}\boldsymbol{w}_{s,a} + \Phi^{-1}(\alpha)\sqrt{\boldsymbol{w}_{s,a}^\mathsf{T}\boldsymbol{\Sigma}_{s,a}\boldsymbol{w}_{s,a}}$$

$$t = \bar{\boldsymbol{p}}_{s,a}^\mathsf{T}\boldsymbol{w}_{s,a} - \Phi^{-1}(1 - \alpha)\sqrt{\boldsymbol{w}_{s,a}^\mathsf{T}\boldsymbol{\Sigma}_{s,a}\boldsymbol{w}_{s,a}} \ ,$$

The first equation follows from substracting $\bar{\boldsymbol{p}}_{s,a}^\mathsf{T}\boldsymbol{w}_{s,a}$ and dividing by $\sqrt{\boldsymbol{w}_{s,a}^\mathsf{T}\boldsymbol{\Sigma}_{s,a}\boldsymbol{w}_{s,a}}$ on both sides. The second equality follows from the definition of CDF of $\mathcal{N}(0,1)$ and the third equality follows from simple algebraic manipulations.

Substituting the value of $t = \mathrm{VaR}_\alpha[\tilde{\boldsymbol{p}}_{s,a}^\mathsf{T}\boldsymbol{w}_{s,a}] = \bar{\boldsymbol{p}}_{s,a}^\mathsf{T}\boldsymbol{w}_{s,a} - \Phi^{-1}(1 - \alpha)\sqrt{\boldsymbol{w}_{s,a}^\mathsf{T}\boldsymbol{\Sigma}_{s,a}\boldsymbol{w}_{s,a}}$ in (13), we obtain the stated results. $\qquad\square$

The normal form of the VaR Bellman optimality operator $\mathcal{T}_{\mathrm{VaR}_\alpha}$ is useful to analyze the asymptotic properties of the VaR Bellman operator.

## B.4 Proof of Theorem 3.4

**Theorem 3.4** (Performance). *Let $\hat{\boldsymbol{v}}$ be the fixed point of the* VaR *Bellman optimality operator $\mathcal{T}_{\mathrm{VaR}_\alpha}$ and $\pi^*$ be the optimal policy in* (1). *Let $\rho^* = \mathrm{VaR}_\delta[\rho(\pi^*, \tilde{\boldsymbol{P}})]$ denote the optimal percentile returns and $\hat{\rho} = \boldsymbol{p}_0^\mathsf{T}\hat{\boldsymbol{v}}$ denote the lower bound on the percentile returns computed using the Bellman operator $\mathcal{T}_{\mathrm{VaR}_\alpha}$ with $\alpha = \delta/s$. Then for each $\delta \in (0, 1/2)$:*

$$\rho^* - \hat{\rho} \le \frac{1}{1 - \gamma} \max_{s \in \mathcal{S}} \max_{a \in \mathcal{A}} \left(\mathrm{VaR}_{1 - \frac{1-\delta}{S}}\left[\tilde{\boldsymbol{p}}_{s,a}^\mathsf{T}\boldsymbol{w}_{s,a}\right] - \mathrm{VaR}_\alpha\left[\tilde{\boldsymbol{p}}_{s,a}^\mathsf{T}\boldsymbol{w}_{s,a}\right]\right) \ . \tag{6}$$

*Proof.* We denote the optimal policy that optimizes the $\delta$-percentile criterion by $\pi^*$, i.e., $\pi^* \in \arg\max_{\pi \in \Pi} \mathrm{VaR}_\delta[\rho(\pi, \tilde{\boldsymbol{P}})]$.

Recall that $\hat{\boldsymbol{v}} \in \mathbb{R}^S$ is the fixed point of the $\mathrm{VaR}_\alpha$ Bellman optimality operator $\mathcal{T}_{\mathrm{VaR}_\alpha}$, i.e., $\hat{\boldsymbol{v}} = \mathcal{T}_{\mathrm{VaR}_\alpha}\hat{\boldsymbol{v}}$ and $\hat{\rho} = \boldsymbol{p}_0^\mathsf{T}\hat{\boldsymbol{v}}$ is the returns of the corresponding policy. The operator $\mathcal{T}_{\boldsymbol{P}}^\pi$ represents the Bellman evaluation operator for a given policy $\pi \in \Pi$ and a transition probability model $\boldsymbol{P}$. The Bellman evaluation operator $\mathcal{T}_{\boldsymbol{P}}^\pi$ for any $s \in \mathcal{S}$ and value $\boldsymbol{v} \in \mathbb{R}^S$ is defined as

$$(\mathcal{T}_{\boldsymbol{P}}^\pi \boldsymbol{v})_s = \boldsymbol{p}_{s,\pi(s)}^\mathsf{T}\boldsymbol{w}_{s,\pi(s)} \ ,$$

where $\boldsymbol{w}_{s,\pi(s)} = \boldsymbol{r}_{s,\pi(s)} + \gamma \cdot \boldsymbol{v}$.

It is known that the Bellman operator $\mathcal{T}_{\boldsymbol{P}}^\pi$ is a $\gamma$-contraction mapping, monotone, and has a unique fixed point. We will use $\tilde{\boldsymbol{v}}^{\pi^*} = \mathcal{T}_{\tilde{\boldsymbol{P}}}^{\pi^*}\tilde{\boldsymbol{v}}^{\pi^*}$ to represent the random unique fixed point of the Bellman operator $\mathcal{T}_{\tilde{\boldsymbol{P}}}^{\pi^*}$ defined for a random realization of transition probabilities $\tilde{\boldsymbol{P}}$. Furthermore, we will use $\boldsymbol{p}_0^\mathsf{T}\tilde{\boldsymbol{v}}^{\pi^*} = \rho(\pi^*, \tilde{\boldsymbol{P}})$ to denote the random expected returns corresponding to policy $\pi^*$ and random realization of the transition probabilities $\tilde{\boldsymbol{P}}$.

Suppose that $c \in \mathbb{R}_+$ upper-bounds the difference in performance of the $\mathrm{VaR}_\alpha$ policy $\hat{\pi}$ and optimal percentile-criterion policy $\pi^*$, i.e., $\rho^* - \hat{\rho} = c \iff \rho^* = \hat{\rho} - c$.

From the definition of VaR and $\pi^*$, we know that
$$\rho^* = \text{VaR}_\delta[\rho(\pi^*, \tilde{P})] = \sup\{t : \Pr[\rho(\pi^*, \tilde{P}) \geq t] \geq 1 - \delta\} \ .$$

Since $\hat{\rho} + c$ upper bounds $\rho^*$, we can write
$$\Pr[\rho(\pi^*, \tilde{P}) \leq \hat{\rho} + c] \geq \delta \iff \Pr[\rho(\pi^*, \tilde{P}) - \hat{\rho} \leq c] \geq \delta \ . \tag{14}$$

The above equation suggests that the error in the performance of the $\text{VaR}_\alpha$ policy is upper-bounded by $c$ if $\Pr[\rho(\pi^*, \tilde{P}) - \hat{\rho} \leq c]$ holds with at least $\delta$ probability.

To derive a lower bound on $c$, we proceed as follows.

We begin by showing that $\rho(\pi^*, \tilde{P}) - \hat{\rho} \leq \|\tilde{v}^{\pi^*} - \hat{v}\|_\infty$.
$$
\begin{aligned}
\rho(\pi^*, \tilde{P}) - \hat{\rho} &= \rho(\pi^*, \tilde{P}) - p_0^\top \hat{v} && \text{from the definition of } \hat{\rho} \text{ and } \rho(\pi^*, \tilde{P}) \\
&\leq |p_0^\top \tilde{v}^{\pi^*} - p_0^\top \hat{v}| && |x| \geq x \\
&\leq \|p_0\|_1 \|\tilde{v}^{\pi^*} - \hat{v}\|_\infty && \text{from Hölder's Inequality} \\
&\leq \|\tilde{v}^{\pi^*} - \hat{v}\|_\infty && \|p_0\|_1 = 1 \ .
\end{aligned}
\tag{15}
$$

Next, we bound $\|\tilde{v}^{\pi^*} - \hat{v}\|_\infty$ to obtain a lower-bound on $c$.
$$
\begin{aligned}
\|\tilde{v}^{\pi^*} - \hat{v}\|_\infty &\overset{(a)}{=} \|\tilde{v}^{\pi^*} - \mathcal{T}_{\tilde{P}}^{\pi^*} \hat{v} + \mathcal{T}_{\tilde{P}}^{\pi^*} \hat{v} - \hat{v}\|_\infty \\
&\overset{(b)}{=} \|\mathcal{T}_{\tilde{P}}^{\pi^*} \tilde{v}^{\pi^*} - \mathcal{T}_{\tilde{P}}^{\pi^*} \hat{v} + \mathcal{T}_{\tilde{P}}^{\pi^*} \hat{v} - \mathcal{T}_{\text{VaR}_\alpha} \hat{v}\|_\infty \\
&\overset{(c)}{\leq} \|\mathcal{T}_{\tilde{P}}^{\pi^*} \tilde{v}^{\pi^*} - \mathcal{T}_{\tilde{P}}^{\pi^*} \hat{v}\|_\infty + \|\mathcal{T}_{\tilde{P}}^{\pi^*} \hat{v} - \mathcal{T}_{\text{VaR}_\alpha} \hat{v}\|_\infty \\
&\overset{(d)}{\leq} \gamma \|\tilde{v}^{\pi^*} - \hat{v}\|_\infty + \|\mathcal{T}_{\tilde{P}}^{\pi^*} \hat{v} - \mathcal{T}_{\text{VaR}_\alpha} \hat{v}\|_\infty \ .
\end{aligned}
$$

$(a)$ follows by simply adding and subtracting $\mathcal{T}_{\tilde{P}}^{\pi^*} \hat{v}$, $(b)$ follows from the fact that $\tilde{v}^{\pi^*}$ and $\hat{v}$ are the fixed points of $\mathcal{T}_{\tilde{P}}^{\pi^*}$ and $\mathcal{T}_{\text{VaR}_\alpha}$ respectively, $(c)$ follows from applying the triangle inequality to the R.H.S., and $(d)$ follows from the contraction property of a Bellman operator.

Rearranging and re-normalizing the above terms, we get
$$\|\tilde{v}^{\pi^*} - \hat{v}\|_\infty \leq \frac{1}{(1 - \gamma)} \|\mathcal{T}_{\tilde{P}}^{\pi^*} \hat{v} - \mathcal{T}_{\text{VaR}_\alpha} \hat{v}\|_\infty \ . \tag{16}$$

Then, combining (15) and (16), we get
$$
\begin{aligned}
\rho(\pi^*, \tilde{P}) - \hat{\rho} &\leq \frac{1}{1 - \gamma} \|\mathcal{T}_{\tilde{P}}^{\pi^*} \hat{v} - \mathcal{T}_{\text{VaR}_\alpha} \hat{v}\|_\infty \\
&\overset{(a)}{=} \frac{1}{1 - \gamma} \max_{s \in \mathcal{S}} \left( (\mathcal{T}_{\tilde{P}}^{\pi^*} \hat{v})_s - (\mathcal{T}_{\text{VaR}_\alpha} \hat{v})_s \right) \\
&\overset{(b)}{=} \frac{1}{1 - \gamma} \max_{s \in \mathcal{S}} \left( \tilde{p}_{s,\pi^*(s)}^\top \hat{w}_{s,\pi^*(s)} - \text{VaR}_\alpha \left[ \tilde{p}_{s,\hat{\pi}(s)}^\top \hat{w}_{s,\hat{\pi}(s)} \right] \right) \\
&\overset{(c)}{\leq} \frac{1}{1 - \gamma} \max_{s \in \mathcal{S}} \left( \tilde{p}_{s,\pi^*(s)}^\top \hat{w}_{s,\pi^*(s)} - \text{VaR}_\alpha \left[ \tilde{p}_{s,\pi^*(s)}^\top \hat{w}_{s,\pi^*(s)} \right] \right) \ .
\end{aligned}
\tag{17}
$$

$(a)$ follows from the definition of $l_\infty$ norm, $(b)$ follows from the definitions of $\mathcal{T}_{\text{VaR}_\alpha}$ and $\mathcal{T}_{\tilde{P}}^{\pi^*}$ Bellman operators, and $(c)$ follows from the fact that $\hat{\pi}(s) = \arg\max_{a \in \mathcal{A}} \text{VaR}_\alpha \left[ \tilde{p}_{s,a}^\top \hat{w}_{s,a} \right]$ and thus, $\text{VaR}_\alpha \left[ \tilde{p}_{s,\hat{\pi}(s)}^\top \hat{w}_{s,\hat{\pi}(s)} \right] \geq \text{VaR}_\alpha \left[ \tilde{p}_{s,\pi^*(s)}^\top \hat{w}_{s,\pi^*(s)} \right]$.

Therefore, for any $\varepsilon > 0$, we can write
$$\Pr\left[ \rho(\pi^*, \tilde{P}) - \hat{\rho} \leq \frac{1}{1 - \gamma} \max_{s \in \mathcal{S}} \left( \text{VaR}_{1 - \frac{1 - \delta - \varepsilon}{S}} \left[ \tilde{p}_{s,\pi^*(s)}^\top \hat{w}_{s,\pi^*(s)} \right] - \text{VaR}_\alpha \left[ \tilde{p}_{s,\pi^*(s)}^\top \hat{w}_{s,\pi^*(s)} \right] \right) \right] \geq \delta$$
$$\Pr\left[ \rho(\pi^*, \tilde{P}) - \hat{\rho} \leq \frac{1}{1 - \gamma} \max_{s \in \mathcal{S}, a \in \mathcal{A}} \left( \text{VaR}_{1 - \frac{1 - \delta - \varepsilon}{S}} \left[ \tilde{p}_{s,a}^\top \hat{w}_{s,a} \right] - \text{VaR}_\alpha \left[ \tilde{p}_{s,a}^\top \hat{w}_{s,a} \right] \right) \right] \geq \delta \ .$$
$$\tag{18}$$

The first equation in the above follows from $\Pr\left[\tilde{\boldsymbol{p}}_{s,a}\boldsymbol{w}_{s,a} > \mathrm{VaR}_{1-\frac{1-\delta-\varepsilon}{S}}\left[\tilde{\boldsymbol{p}}_{s,a}^{\mathsf{T}}\hat{\boldsymbol{w}}_{s,a}\right]\right] < \frac{1-\delta}{S}$ for any state $s$ and action $a$ and applying union bound over all states yields $\Pr\left[\tilde{\boldsymbol{p}}_{s,a}\boldsymbol{w}_{s,a} > \mathrm{VaR}_{1-\frac{1-\delta-\varepsilon}{S}}\left[\tilde{\boldsymbol{p}}_{s,a}^{\mathsf{T}}\hat{\boldsymbol{w}}_{s,a}\right]\forall s\in\mathcal{S}\right] < 1 - \delta$. Thus, we get $\Pr\left[\tilde{\boldsymbol{p}}_{s,a}\boldsymbol{w}_{s,a} \le \mathrm{VaR}_{1-\frac{1-\delta-\varepsilon}{S}}\left[\tilde{\boldsymbol{p}}_{s,a}^{\mathsf{T}}\hat{\boldsymbol{w}}_{s,a}\right],\forall s\in\mathcal{S}\right] \ge \delta$. The second equation follows by simply replacing $\pi^*(s)$ with the worst-case action, i.e., action that maximizes the upper bound.

Comparing (14) and (18), we get

$$\rho^* - \hat{\rho} \le \frac{1}{1-\gamma}\max_{s\in\mathcal{S}}\max_{a\in\mathcal{A}}\left(\inf_{\varepsilon>0}\mathrm{VaR}_{1-\frac{1-\delta-\varepsilon}{S}}\left[\tilde{\boldsymbol{p}}_{s,a}^{\mathsf{T}}\boldsymbol{w}_{s,a}\right] - \mathrm{VaR}_{\alpha}\left[\tilde{\boldsymbol{p}}_{s,a}^{\mathsf{T}}\boldsymbol{w}_{s,a}\right]\right) .$$

Note that VaR of returns is upper semicontinuous and thus, this implies that the infimum in the above equation is achieved at $\varepsilon = 0$.

Therefore, we can write

$$\rho^* - \hat{\rho} \le \frac{1}{1-\gamma}\max_{s\in\mathcal{S}}\max_{a\in\mathcal{A}}\left(\mathrm{VaR}_{1-\frac{1-\delta}{S}}\left[\tilde{\boldsymbol{p}}_{s,a}^{\mathsf{T}}\boldsymbol{w}_{s,a}\right] - \mathrm{VaR}_{\alpha}\left[\tilde{\boldsymbol{p}}_{s,a}^{\mathsf{T}}\boldsymbol{w}_{s,a}\right]\right) .$$

$\square$

## B.5 Proof of Theorem 3.5

**Theorem 3.5** (Asymptotic Performance). *Suppose that the normality assumptions on the posterior distribution $\tilde{\boldsymbol{P}}$ in Section 2 are satisfied. For any $\delta \in (0, 1/2)$, set $\alpha = \delta/S$ in $\mathcal{T}_{\mathrm{VaR}_\alpha}$. For any state $s$ and action $a$, let $I(\boldsymbol{p}_{s,a}^*)$ be the Fisher information matrix corresponding to the true transition probabilities $\boldsymbol{p}_{s,a}^*$. Furthermore, let $\sigma_{\max}^2 = \max_{s\in\mathcal{S},a\in\mathcal{A}}\hat{\boldsymbol{w}}_{s,a}^{\mathsf{T}}I(\boldsymbol{p}_{s,a}^*)^{-1}\hat{\boldsymbol{w}}_{s,a}$ be the maximum over state-action pairs of the asymptotic variance of the returns estimate $\tilde{\boldsymbol{p}}_{s,a}^{\mathsf{T}}\hat{\boldsymbol{w}}_{s,a}$. Then the asymptotic performance of the $\mathrm{VaR}$ framework $\hat{\rho}$ w.r.t. the optimal percentile returns $\rho^*$ satisfies*

$$\lim_{N\to\infty}\sqrt{N}(\rho^* - \hat{\rho}) \le \frac{1}{1-\gamma}\left(2\Phi^{-1}(1-\delta/S)\sigma_{\max}\right) \le \frac{1}{1-\gamma}\sqrt{8\ln(S/\delta)}\sigma_{\max} .$$

*Proof.* As noted in Section 2, we assume that as $N \to \infty$, the posterior distribution of transition probabilities at any state $s$ and action $a$ ($\tilde{\boldsymbol{p}}_{s,a}$) converges in the limit to a multivariate Gaussian distribution with mean $\boldsymbol{p}_{s,a}^*$ and covariance matrix $I(\boldsymbol{p}_{s,a}^*)^{-1}/N$. Hence, we can write

$$\lim_{N\to\infty}\sqrt{N}(\tilde{\boldsymbol{p}}_{s,a}^{\mathsf{T}}\boldsymbol{w}_{s,a} - \boldsymbol{p}_{s,a}^{*\mathsf{T}}\boldsymbol{w}_{s,a}) \rightsquigarrow \mathcal{N}(0, \boldsymbol{w}_{s,a}^{\mathsf{T}}I(\boldsymbol{p}_{s,a}^*)^{-1}\boldsymbol{w}_{s,a}) .$$

We know from Proposition 3.3, that $\mathrm{VaR}_\alpha$ of a univariate Gaussian random variable $X \sim \mathcal{N}(\mu, \sigma^2)$ can be written as $\mathrm{VaR}_\alpha[X] = \mu + \Phi^{-1}(\alpha)\sigma$. Therefore, applying this result to the R.H.S. of Equation (6), we get

$$
\begin{aligned}
\lim_{N\to\infty}\sqrt{N}(\rho^* - \hat{\rho}) &\overset{(a)}{\le} \frac{1}{1-\gamma}\max_{s\in\mathcal{S}}\max_{a\in\mathcal{A}}\left(\sqrt{N}\boldsymbol{p}_{s,a}^{*\mathsf{T}}\boldsymbol{w}_{s,a} + \Phi^{-1}\left(1 - \frac{1-\delta}{S}\right)\sqrt{\boldsymbol{w}_{s,a}^{\mathsf{T}}I(\boldsymbol{p}_{s,a}^*)^{-1}\boldsymbol{w}_{s,a}}\right. \\
&\qquad\qquad\left. - \left(\sqrt{N}\boldsymbol{p}_{s,a}^{*\mathsf{T}}\boldsymbol{w}_{s,a} + \Phi^{-1}\left(\frac{\delta}{S}\right)\sqrt{\boldsymbol{w}_{s,a}^{\mathsf{T}}I(\boldsymbol{p}_{s,a}^*)^{-1}\boldsymbol{w}_{s,a}}\right)\right) \\
&\overset{(b)}{=} \frac{1}{1-\gamma}\max_{s\in\mathcal{S}}\max_{a\in\mathcal{A}}\left(\Phi^{-1}\left(1 - \frac{1-\delta}{S}\right)\sqrt{\boldsymbol{w}_{s,a}^{\mathsf{T}}I(\boldsymbol{p}_{s,a}^*)^{-1}\boldsymbol{w}_{s,a}}\right. \\
&\qquad\qquad\left. - \Phi^{-1}\left(\frac{\delta}{S}\right)\sqrt{\boldsymbol{w}_{s,a}^{\mathsf{T}}I(\boldsymbol{p}_{s,a}^*)^{-1}\boldsymbol{w}_{s,a}}\right) \\
&\overset{(c)}{=} \frac{1}{1-\gamma}\max_{s\in\mathcal{S}}\max_{a\in\mathcal{A}}\left(\left(\Phi^{-1}\left(1 - \frac{1-\delta}{S}\right) - \Phi^{-1}\left(\frac{\delta}{S}\right)\right)\sqrt{\boldsymbol{w}_{s,a}^{\mathsf{T}}I(\boldsymbol{p}_{s,a}^*)^{-1}\boldsymbol{w}_{s,a}}\right) \\
&\overset{(d)}{\le} \frac{1}{1-\gamma}\max_{s\in\mathcal{S}}\max_{a\in\mathcal{A}}\left(2\Phi^{-1}\left(1 - \frac{\delta}{S}\right)\sqrt{\boldsymbol{w}_{s,a}^{\mathsf{T}}I(\boldsymbol{p}_{s,a}^*)^{-1}\boldsymbol{w}_{s,a}}\right) .
\end{aligned}
$$

$$(19)$$

$(a)$ follows from the definition of $\text{VaR}_\alpha$ under Gaussian assumptions, $(b)$ and $(c)$ follow from simple algebraic manipulations, and $(d)$ follows from the fact that $\forall \delta \in (0, 1/2), 0 \geq \Phi^{-1}\left(1 - \frac{1-\delta}{S}\right) \geq \Phi^{-1}(\frac{\delta}{S}), -\Phi^{-1}(\frac{\delta}{S}) = \Phi^{-1}(1 - \frac{\delta}{S})$, and assuming $S > 1$.

We prove the second inequality in Theorem 3.5, by leveraging the sub-Gaussian bounds for a standard normal distribution $\mathcal{N}(0, \sigma^2)$ to show that $\forall \alpha' \in (0, 1/2), \Phi^{-1}(1 - \alpha') \leq \sqrt{2\ln(1/\alpha')}$.

We know that for a standard normal distribution $X \sim \mathcal{N}(0, \sigma^2)$ where $\sigma \in \mathbb{R}$, the following sub-Gaussian bounds holds [7]:

$$\Pr\left(-\sqrt{2\ln(2/\alpha')}\sigma \leq X \leq \sqrt{2\ln(2/\alpha')}\sigma\right) \geq 1 - \alpha' \ . \tag{20}$$

By definition, for a standard normal distribution $\mathcal{N}(0, \sigma^2)$, it holds

$$\Pr\left(-\Phi^{-1}(1 - \alpha'/2)\sigma \leq X \leq \Phi^{-1}(1 - \alpha'/2)\sigma\right) = 1 - \alpha' \ . \tag{21}$$

Comparing equation (20) and (21), we get

$$\Phi^{-1}(1 - \alpha'/2) \leq \sqrt{2\ln(2/\alpha')} \implies \Phi^{-1}(1 - \alpha') \leq \sqrt{2\ln(1/\alpha')} \ . \tag{22}$$

Using the result in (22) in equation (19), proves the second inequality of the theorem. $\quad\square$

## B.6 Proof of Proposition 3.6

**Proposition 3.6** (Empirical Error Bound). *For any state $s$, action $a$ and value function $v$, let $\widehat{\text{VaR}}_\alpha[\tilde{\boldsymbol{p}}_{s,a}^\mathsf{T}\boldsymbol{w}_{s,a}]$ represent the empirical estimate of $\alpha$-percentile of returns $\text{VaR}_\alpha[\tilde{\boldsymbol{p}}_{s,a}^\mathsf{T}\boldsymbol{w}_{s,a}]$ and $\Phi_f$ represent the cumulative density function (CDF) of the random estimate of returns $\tilde{\boldsymbol{p}}_{s,a}^\mathsf{T}\boldsymbol{w}_{s,a}$. Suppose that $\Phi_f$ is differentiable at the point $\text{VaR}_\alpha[\tilde{\boldsymbol{p}}_{s,a}^\mathsf{T}\boldsymbol{w}_{s,a}]$ and let $\eta = \Phi_f'\left(\text{VaR}_\alpha[\tilde{\boldsymbol{p}}_{s,a}^\mathsf{T}\boldsymbol{w}_{s,a}]\right)$ represents the density of estimate of returns at point $\text{VaR}_\alpha[\tilde{\boldsymbol{p}}_{s,a}^\mathsf{T}\boldsymbol{w}_{s,a}]$. Let $M^*$ be the number of posterior samples required to obtain empirical error $\varepsilon \in \mathbb{R}$, with confidence $1 - \zeta$, where $0 < \zeta < 1$, i.e., $\Pr\left[|\widehat{\text{VaR}}_\alpha[\tilde{\boldsymbol{p}}_{s,a}^\mathsf{T}\boldsymbol{w}_{s,a}] - \text{VaR}_\alpha[\tilde{\boldsymbol{p}}_{s,a}^\mathsf{T}\boldsymbol{w}_{s,a}]| > \varepsilon\right] \leq \zeta$. Then, $\lim_{\varepsilon \to 0} M^*\varepsilon^2 = \ln(2/\zeta)/2\eta^2$.*

*Proof.* To prove this theorem, we first compute the derivative of the inverse of the CDF $\partial\Phi_f^{-1}(\alpha)/\partial\alpha$ as follows. From the definition of the cdf $\Phi_f$ and VaR, we know that, for any $\alpha \in (0, 1/2)$, $\Phi_f^{-1}(\alpha) = \text{VaR}_\alpha[\tilde{\boldsymbol{p}}_{s,a}^\mathsf{T}\boldsymbol{w}_{s,a}]$.

From the inverse-function theorem, we get,

$$
\begin{aligned}
(\Phi_f^{-1}(\alpha))' &= \frac{1}{\Phi_f'(\text{VaR}_\alpha[\tilde{\boldsymbol{p}}_{s,a}^\mathsf{T}\boldsymbol{w}_{s,a}])} \\
&= \frac{1}{\Phi_f'(\Phi_f^{-1}(\alpha))} \\
&= \frac{1}{\eta} \ .
\end{aligned}
$$

Equipped with the above result, we can now proceed to prove the main result.

To prove the result, we need to find $M^*$ such that

$$\Pr\left[\text{VaR}_\alpha[\tilde{\boldsymbol{p}}_{s,a}^\mathsf{T}\boldsymbol{w}_{s,a}] - \varepsilon \leq \widehat{\text{VaR}}_\alpha[\tilde{\boldsymbol{p}}_{s,a}^\mathsf{T}\boldsymbol{w}_{s,a}] \leq \text{VaR}_\alpha[\tilde{\boldsymbol{p}}_{s,a}^\mathsf{T}\boldsymbol{w}_{s,a}] + \varepsilon\right] \geq 1 - \zeta$$
$$\overset{(a)}{=} \Pr\left[\Phi_f^{-1}(\alpha - \varepsilon\eta) \leq \widehat{\text{VaR}}_\alpha[\tilde{\boldsymbol{p}}_{s,a}^\mathsf{T}\boldsymbol{w}_{s,a}] \leq \Phi_f^{-1}(\alpha + \varepsilon\eta)\right] \geq 1 - \zeta \ . \tag{23}$$

Equation $(a)$ follows from applying a first order Taylor expansion to $\Phi_f^{-1}$ around the point $\alpha$. We apply the following results to obtain a bound on $M^*$.

Let $\hat{F}$ and $F$ represent the empirical CDF and the true CDF of a random variable $\tilde{Z}$. Suppose that the empirical estimate of the CDF $\hat{F}$ is estimated using $M^*$ samples from the true distribution of $\tilde{Z}$ and

$0 < \zeta < 1$ represents the desired level of confidence guarantees, Then, from DWK inequality [31], we know that

$$\Pr\left(\|\hat{F} - F\|_\infty \geq \sqrt{\ln(2/\varsigma)/2M^*}\right) \leq \zeta \ .$$

The above equation implies that

$$\Pr\left(\exists p \in (0,1) : F^{-1}(p) < \hat{F}^{-1}(p - z_t) \text{ or } F^{-1}(p) > \hat{F}^{-1}(p + u_t)\right) \leq \zeta \ , \tag{24}$$

where $z_t = u_t = \sqrt{\ln(2/\varsigma)/2M^*}$.

Thus, applying equation (24) to (23), i.e., $z_t = \sqrt{\ln(2/\varsigma)/2M^*} = \varepsilon\eta$ gives $M^* = \ln(2/\varsigma)/2\varepsilon^2\eta^2$.

$\square$

### B.7 Proof of Proposition 3.7

**Proposition 3.7** (Value Iteration Error). *Define the empirical* VaR *Bellman optimality operator* $\mathcal{T}_{\widehat{\mathrm{VaR}}_\alpha}$ *for any value* $v \in \mathbb{R}^S$ *and state* $s \in \mathcal{S}$ *as* $(\mathcal{T}_{\widehat{\mathrm{VaR}}_\alpha} v)_s = \max_{a \in \mathcal{A}} \widehat{\mathrm{VaR}}_\alpha [\tilde{p}_{s,a}^\top w_{s,a}]$. *Let* $\hat{v} \in \mathbb{R}^S$ *and* $\hat{u} \in \mathbb{R}^S$ *represent the fixed points of* $\mathcal{T}_{\mathrm{VaR}_\alpha}$ *and* $\mathcal{T}_{\widehat{\mathrm{VaR}}_\alpha}$ *respectively. Suppose that Algorithm 3.1 returns policy* $\pi_k$ *and value function* $u_k$ *and* $\|\hat{u} - u_0\|_\infty \leq r_{\max}/1-\gamma$. *Then,*

$$\|\hat{u} - u_k\|_\infty \leq \varepsilon, \text{ and } \|\hat{u} - u_{\pi_k}\|_\infty \leq \frac{2\varepsilon\gamma}{1 - \gamma} \ .$$

*Furthermore, the gap between the empirical and true value function is bounded by*

$$\|\hat{v} - \hat{u}\|_\infty \leq \frac{1}{1 - \gamma} \min\left(\|\mathcal{T}_{\widehat{\mathrm{VaR}}_\alpha}\hat{u} - \mathcal{T}_{\mathrm{VaR}_\alpha}\hat{u}\|_\infty, \|\mathcal{T}_{\widehat{\mathrm{VaR}}_\alpha}\hat{v} - \mathcal{T}_{\mathrm{VaR}_\alpha}\hat{v}\|_\infty\right) \ .$$

*Proof.* We begin by noting that similar to the VaR Bellman operator $\mathcal{T}_{\mathrm{VaR}_\alpha}$, the empirical VaR Bellman operator $\mathcal{T}_{\widehat{\mathrm{VaR}}_\alpha}$ is also a contraction mapping because of the monotonicity of the $\mathrm{VaR}_\alpha$ operator and since the empirical $\mathrm{VaR}_\alpha$ in the algorithm is always computed from a fixed set of transition probabilities sampled from the posterior of $\tilde{P}$.

We can bound the error between the value function $u_k$ returned by Algorithm 3.1 and the fixed point $(\hat{u})$ of $\mathcal{T}_{\widehat{\mathrm{VaR}}_\alpha}$ as

$$\|u_{k-1} - \hat{u}\|_\infty = \|u_{k-1} + u_k - u_k - \hat{u}\|_\infty$$

$$\overset{(a)}{\leq} \|u_{k-1} - u_k\|_\infty + \|u_k - \hat{u}\|_\infty$$

$$\frac{1}{\gamma}\|u_k - \hat{u}\|_\infty \overset{(b)}{\leq} \frac{\varepsilon(\gamma - 1)}{\gamma} + \|u_k - \hat{u}\|_\infty$$

$$\left(\frac{1}{\gamma} - 1\right)\|u_k - \hat{u}\|_\infty \overset{(c)}{\leq} \frac{\varepsilon(1 - \gamma)}{\gamma}$$

$$\|u_k - \hat{u}\|_\infty \overset{(d)}{\leq} \varepsilon \ .$$

$(a)$ follows from applying the triangle inequality to the $l_\infty$ norm, $(b)$ follows from the fact that $\|u_{k-1} - u_k\|_\infty \leq \varepsilon(1-\gamma)/\gamma$ and $\|u_{k-1} - \hat{u}\|_\infty \leq \frac{1}{\gamma}\|u_k - \hat{u}\|_\infty$ due to the contraction property of $\mathcal{T}_{\widehat{\mathrm{VaR}}_\alpha}$ operator, and finally, $(c)$ and $(d)$ follow from simply arranging the terms on both sides.

Next, we bound the approximation error $\|u_k - u_{\pi_k}\|_\infty$. From the above result, we can write

$$-\varepsilon\mathbf{1} \leq \hat{u} - u_k \leq \varepsilon\mathbf{1} \ . \tag{25}$$

We will use the shorthand $\pi$, $\mathcal{T}$ and $\mathcal{T}^{\pi_k}$ to represent the policy $\pi_k$, empirical VaR Bellman optimality $\mathcal{T}_{\widehat{\mathrm{VaR}_\alpha}}$, and VaR Bellman policy evaluation operator $\mathcal{T}^\pi_{\widehat{\mathrm{VaR}_\alpha}}$ respectively. Suppose that $\delta = \hat{\boldsymbol{u}} - \boldsymbol{u}_\pi$.

$$
\begin{aligned}
\delta &= \hat{\boldsymbol{u}} - \boldsymbol{u}_\pi \\
&\stackrel{\text{(a)}}{=} \mathcal{T}\hat{\boldsymbol{u}} - \mathcal{T}^\pi \boldsymbol{u}_\pi \\
&\stackrel{\text{(b)}}{\leq} \mathcal{T}(\boldsymbol{u}_k + \varepsilon \mathbf{1}) - \mathcal{T}^\pi \boldsymbol{u}_\pi \\
&\stackrel{\text{(c)}}{=} \mathcal{T}\boldsymbol{u}_k - \mathcal{T}^\pi \boldsymbol{u}_\pi + \gamma\varepsilon\mathbf{1} \\
&\stackrel{\text{(d)}}{=} \mathcal{T}^\pi \boldsymbol{u}_k - \mathcal{T}^\pi \boldsymbol{u}_\pi + \gamma\varepsilon\mathbf{1} \\
&\stackrel{\text{(e)}}{\leq} \mathcal{T}^\pi(\hat{\boldsymbol{u}} + \varepsilon\mathbf{1}) - \mathcal{T}^\pi \boldsymbol{u}_\pi + \gamma\varepsilon\mathbf{1} \\
&\stackrel{\text{(f)}}{=} \mathcal{T}^\pi \hat{\boldsymbol{u}} - \mathcal{T}^\pi \boldsymbol{u}_\pi + 2\gamma\varepsilon\mathbf{1} \\
\|\delta\|_\infty &\stackrel{\text{(g)}}{=} \|\mathcal{T}^\pi \hat{\boldsymbol{u}} - \mathcal{T}^\pi \boldsymbol{u}_\pi + 2\gamma\varepsilon\mathbf{1}\|_\infty \\
&\stackrel{\text{(h)}}{=} \gamma\|\hat{\boldsymbol{u}} - \boldsymbol{u}_\pi\|_\infty + 2\gamma\varepsilon\mathbf{1} \\
(1-\gamma)\|\delta\|_\infty &\stackrel{\text{(i)}}{=} 2\gamma\varepsilon\mathbf{1} \\
\|\delta\|_\infty &\stackrel{\text{(j)}}{\leq} \frac{2\gamma\varepsilon}{(1-\gamma)} \quad .
\end{aligned}
\tag{26}
$$

$(a)$ follows from the fact that $\hat{\boldsymbol{u}}$ is the fixed point of $\mathcal{T}\hat{\boldsymbol{u}}$ and $\boldsymbol{u}_\pi$ is the fixed point of $\mathcal{T}^\pi$, $(b)$ follows from (25), $(c)$ follows from the $\gamma$-contraction property of $\mathcal{T}$ operator, $(d)$ follows from the definition of policy $\pi = \pi_k$, $(e)$ follows from (25), $(f)$ follows from simple algebraic manipulations, $(g)$ follows from taking $l_\infty$-norm on both sides, $(h)$ follows from applying triangle inequality, and $(i)$ and $(j)$ follow from simple algebraic manipulations.

Thus, $\|\hat{\boldsymbol{u}} - \boldsymbol{u}_\pi\|_\infty \leq \frac{2\gamma\varepsilon}{(1-\gamma)}$.

Next we bound the error between the optimal value function $\hat{\boldsymbol{v}}$ and the fixed point of the empirical VaR Bellman optimality operator $\hat{\boldsymbol{u}}$.

$$
\begin{aligned}
\|\hat{\boldsymbol{v}} - \hat{\boldsymbol{u}}\|_\infty &\stackrel{\text{(a)}}{=} \|\mathcal{T}_{\mathrm{VaR}_\alpha}\hat{\boldsymbol{v}} + \mathcal{T}_{\mathrm{VaR}_\alpha}\hat{\boldsymbol{u}} - \mathcal{T}_{\mathrm{VaR}_\alpha}\hat{\boldsymbol{u}} - \mathcal{T}_{\widehat{\mathrm{VaR}_\alpha}}\hat{\boldsymbol{u}}\|_\infty \\
&\stackrel{\text{(b)}}{=} \|\mathcal{T}_{\mathrm{VaR}_\alpha}\hat{\boldsymbol{v}} - \mathcal{T}_{\mathrm{VaR}_\alpha}\hat{\boldsymbol{u}} + \mathcal{T}_{\mathrm{VaR}_\alpha}\hat{\boldsymbol{u}} - \mathcal{T}_{\widehat{\mathrm{VaR}_\alpha}}\hat{\boldsymbol{u}}\|_\infty \\
&\stackrel{\text{(c)}}{\leq} \|\mathcal{T}_{\mathrm{VaR}_\alpha}\hat{\boldsymbol{v}} - \mathcal{T}_{\mathrm{VaR}_\alpha}\hat{\boldsymbol{u}}\|_\infty + \|\mathcal{T}_{\mathrm{VaR}_\alpha}\hat{\boldsymbol{u}} - \mathcal{T}_{\widehat{\mathrm{VaR}_\alpha}}\hat{\boldsymbol{u}}\|_\infty \\
&\stackrel{\text{(d)}}{\leq} \gamma\|\hat{\boldsymbol{v}} - \hat{\boldsymbol{u}}\|_\infty + \|\mathcal{T}_{\mathrm{VaR}_\alpha}\hat{\boldsymbol{u}} - \mathcal{T}_{\widehat{\mathrm{VaR}_\alpha}}\hat{\boldsymbol{u}}\|_\infty \quad .
\end{aligned}
$$

$(a)$ follows by simply adding and subtracting $\mathcal{T}_{\mathrm{VaR}_\alpha}\hat{\boldsymbol{u}}$, $(b)$ follows from simply arranging the terms, $(c)$ follows from applying triangle inequality, and $(d)$ follows from the contraction property of the Bellman operator $\mathcal{T}_{\mathrm{VaR}_\alpha}$.

Using simple algebraic manipulations and rearranging the terms in the above equation, we can write

$$
\|\hat{\boldsymbol{v}} - \hat{\boldsymbol{u}}\|_\infty \leq \frac{\|\mathcal{T}_{\mathrm{VaR}_\alpha}\hat{\boldsymbol{u}} - \mathcal{T}_{\widehat{\mathrm{VaR}_\alpha}}\hat{\boldsymbol{u}}\|_\infty}{1-\gamma} \quad .
\tag{27}
$$

Similarly, we can bound $\|\hat{\boldsymbol{v}} - \hat{\boldsymbol{u}}\|_\infty$ in the following manner.

$$
\begin{aligned}
\|\hat{\boldsymbol{v}} - \hat{\boldsymbol{u}}\|_\infty &\stackrel{\text{(a)}}{=} \|\mathcal{T}_{\mathrm{VaR}_\alpha}\hat{\boldsymbol{v}} + \mathcal{T}_{\widehat{\mathrm{VaR}_\alpha}}\hat{\boldsymbol{v}} - \mathcal{T}_{\widehat{\mathrm{VaR}_\alpha}}\hat{\boldsymbol{v}} - \mathcal{T}_{\widehat{\mathrm{VaR}_\alpha}}\hat{\boldsymbol{u}}\|_\infty \\
&\stackrel{\text{(b)}}{=} \|\mathcal{T}_{\mathrm{VaR}_\alpha}\hat{\boldsymbol{v}} - \mathcal{T}_{\widehat{\mathrm{VaR}_\alpha}}\hat{\boldsymbol{v}} + \mathcal{T}_{\widehat{\mathrm{VaR}_\alpha}}\hat{\boldsymbol{v}} - \mathcal{T}_{\widehat{\mathrm{VaR}_\alpha}}\hat{\boldsymbol{u}}\|_\infty \\
&\stackrel{\text{(c)}}{\leq} \|\mathcal{T}_{\widehat{\mathrm{VaR}_\alpha}}\hat{\boldsymbol{v}} - \mathcal{T}_{\widehat{\mathrm{VaR}_\alpha}}\hat{\boldsymbol{u}}\|_\infty + \|\mathcal{T}_{\mathrm{VaR}_\alpha}\hat{\boldsymbol{v}} - \mathcal{T}_{\widehat{\mathrm{VaR}_\alpha}}\hat{\boldsymbol{v}}\|_\infty \\
&\stackrel{\text{(d)}}{\leq} \gamma\|\hat{\boldsymbol{v}} - \hat{\boldsymbol{u}}\|_\infty + \|\mathcal{T}_{\mathrm{VaR}_\alpha}\hat{\boldsymbol{v}} - \mathcal{T}_{\widehat{\mathrm{VaR}_\alpha}}\hat{\boldsymbol{v}}\|_\infty \quad .
\end{aligned}
$$

$(a)$ follows by simply adding and subtracting $\mathcal{T}_{\widehat{\mathrm{VaR}}_\alpha}\hat{\boldsymbol{v}}$, $(b)$ follows from simply arranging the terms, $(c)$ follows from applying triangle inequality, and $(d)$ follows from the contraction property of the Bellman operator $\mathcal{T}_{\widehat{\mathrm{VaR}}_\alpha}$.

Using simple algebraic manipulations and rearranging the terms in the above equation, we can write

$$\|\hat{\boldsymbol{v}} - \hat{\boldsymbol{u}}\|_\infty \leq \frac{\|\mathcal{T}_{\mathrm{VaR}_\alpha}\hat{\boldsymbol{v}} - \mathcal{T}_{\widehat{\mathrm{VaR}}_\alpha}\hat{\boldsymbol{v}}\|_\infty}{1 - \gamma} \quad . \tag{28}$$

Thus $\|\hat{\boldsymbol{v}} - \hat{\boldsymbol{u}}\|_\infty \leq \frac{1}{1-\gamma} \min\left(\|\mathcal{T}_{\mathrm{VaR}_\alpha}\hat{\boldsymbol{v}} - \mathcal{T}_{\widehat{\mathrm{VaR}}_\alpha}\hat{\boldsymbol{v}}\|_\infty, \|\mathcal{T}_{\mathrm{VaR}_\alpha}\hat{\boldsymbol{u}} - \mathcal{T}_{\widehat{\mathrm{VaR}}_\alpha}\hat{\boldsymbol{u}}\|_\infty\right)$.

$\square$

## B.8   Proof of Proposition 3.8

**Proposition 3.8** (Time Complexity). *Let $r_{\max} = \max_{s,s' \in \mathcal{S}, a \in \mathcal{A}} |R(s, a, s')|$. Then, Algorithm 3.1 terminates in $k = \lceil \log_{1/\gamma}\left(\frac{r_{max}}{\varepsilon(1-\gamma)}\right) \rceil$ iterations with time complexity $\mathcal{O}\left(S^2 A M \log_{1/\gamma}\left(\frac{r_{max}}{\varepsilon(1-\gamma)}\right)\right)$.*

*Furthermore, for any failure probability $\zeta \in (0, 1)$, suppose that the CDF ($\Phi_f$) of the random estimate of 1-step returns $\tilde{\boldsymbol{p}}_{s,a}^\mathsf{T}\hat{\boldsymbol{w}}_{s,a}$ is differentiable at the point $\mathrm{VaR}_\alpha[\tilde{\boldsymbol{p}}_{s,a}^\mathsf{T}\hat{\boldsymbol{w}}_{s,a}]$, and set $\eta = \Phi'(\mathrm{VaR}_\alpha[\tilde{\boldsymbol{p}}_{s,a}^\mathsf{T}\hat{\boldsymbol{w}}_{s,a}])$ and $M = \mathcal{O}\left(\frac{\log(S/\zeta)}{\eta^2\varepsilon^2(1-\gamma)^2}\right)$. Then with probability at least $1 - \zeta$, it holds that $\|\hat{\boldsymbol{v}} - \boldsymbol{u}_k\|_\infty \leq \mathcal{O}(\varepsilon)$, and Algorithm 3.1 runs in $\mathcal{O}\left(\frac{S^2 A \log_{1/\gamma}\left(\frac{r_{max}}{\varepsilon(1-\gamma)}\right)\log\left(\frac{S}{\zeta}\right)}{\eta^2\varepsilon^2(1-\gamma)^2}\right)$ time.*

*Proof.* To bound the number of iterations ($k$) required to achieve value approximation error $\|\boldsymbol{u}_k - \hat{\boldsymbol{u}}\|_\infty \leq \varepsilon$, we first bound the value approximation error $\|\boldsymbol{u}_k - \hat{\boldsymbol{u}}\|_\infty$ as follows.

$$\begin{aligned}
\|\hat{\boldsymbol{u}} - \boldsymbol{u}_k\|_\infty &\overset{(a)}{\leq} \|\mathcal{T}_{\widehat{\mathrm{VaR}}_\alpha}\hat{\boldsymbol{u}} - \mathcal{T}_{\widehat{\mathrm{VaR}}_\alpha}\boldsymbol{u}_{k-1}\|_\infty \\
&\overset{(b)}{\leq} \gamma\|\hat{\boldsymbol{u}} - \boldsymbol{u}_{k-1}\|_\infty \\
&\overset{(c)}{\leq} \gamma^2\|\mathcal{T}_{\widehat{\mathrm{VaR}}_\alpha}\hat{\boldsymbol{u}} - \mathcal{T}_{\widehat{\mathrm{VaR}}_\alpha}\boldsymbol{u}_{k-2}\|_\infty \\
&\overset{(d)}{\leq} \gamma^k \frac{r_{\max}}{1 - \gamma} \quad .
\end{aligned}$$

$(a)$ follows from the definition of the VaR Bellman operator $\mathcal{T}_{\mathrm{VaR}_\alpha}$, $(b)$ follows from the contraction property of the empirical VaR Bellman operator $\mathcal{T}_{\widehat{\mathrm{VaR}}_\alpha}$, $(c)$ follows from applying the same procedure as in $(a)$ to step $(b)$, and $(d)$ follows from unrolling $(c)$ over $k - 2$ time steps and using the fact that $\|\hat{\boldsymbol{u}} - \boldsymbol{u}_0\|_\infty \leq r_{\max}/1-\gamma$ for $\boldsymbol{u}_0 = \boldsymbol{0}$.

We find $k$ such that $\|\hat{\boldsymbol{u}} - \boldsymbol{u}_k\|_\infty \leq \varepsilon$,

$$\begin{aligned}
\frac{\gamma^k}{1 - \gamma}(r_{\max}) &= \varepsilon \\
k &= \left\lceil \frac{\ln\left(\frac{r_{\max}}{\varepsilon(1-\gamma)}\right)}{\ln(1/\gamma)} \right\rceil \\
k &= \left\lceil \log_{\frac{1}{\gamma}}\left(\frac{r_{\max}}{\varepsilon(1-\gamma)}\right) \right\rceil \quad .
\end{aligned} \tag{29}$$

The time complexity follows from the fact that any quantile of an array of real values can be computed in linear time using the Quick Select algorithm [23], computing 1-step return requires $\mathcal{O}(S)$ operations, and a single iteration of the loop in lines 3-12 of Algorithm 3.1 computes quantile of returns $SA$ times.

Suppose that $\Phi_f$ is differentiable at the point $\mathrm{VaR}_\alpha[\tilde{\boldsymbol{p}}_{s,a}^\mathsf{T}\boldsymbol{w}_{s,a}]$ and let $\eta = \Phi'_f(\mathrm{VaR}_\alpha[\tilde{\boldsymbol{p}}_{s,a}^\mathsf{T}\boldsymbol{w}_{s,a}])$ represents the density of estimate of returns at point $\mathrm{VaR}_\alpha[\tilde{\boldsymbol{p}}_{s,a}^\mathsf{T}\boldsymbol{w}_{s,a}]$.

Let for any state $s$ and action $a$, $\hat{\boldsymbol{w}}_{s,a} = \boldsymbol{r}_{s,a} + \gamma\hat{\boldsymbol{v}}$. From Proposition 3.6, we know that to guarantee $\Pr\left[|\widehat{\mathrm{VaR}}_\alpha[\tilde{\boldsymbol{p}}_{s,a}^\mathsf{T}\hat{\boldsymbol{w}}_{s,a}] - \mathrm{VaR}_\alpha[\tilde{\boldsymbol{p}}_{s,a}^\mathsf{T}\hat{\boldsymbol{w}}_{s,a}]| > \varepsilon\right] \le \zeta$ , M should satisfy $\lim_{\varepsilon\to 0} M\varepsilon^2 = \ln(2/\varsigma)/2\eta^2$.

Furthermore, in Proposition 3.7, we showed that $\|\hat{\boldsymbol{v}} - \hat{\boldsymbol{u}}\|_\infty \le \frac{\|\mathcal{T}_{\mathrm{VaR}_\alpha}\hat{\boldsymbol{v}} - \mathcal{T}_{\widehat{\mathrm{VaR}}_\alpha}\hat{\boldsymbol{v}}\|_\infty}{1-\gamma}$.

Then,

$$
\begin{aligned}
\Pr\left[\|\hat{\boldsymbol{v}} - \hat{\boldsymbol{u}}\|_\infty \le \varepsilon\right] &\overset{(a)}{\ge} \Pr\left[\frac{\|\mathcal{T}_{\mathrm{VaR}_\alpha}\hat{\boldsymbol{v}} - \mathcal{T}_{\widehat{\mathrm{VaR}}_\alpha}\hat{\boldsymbol{v}}\|_\infty}{1-\gamma} \le \varepsilon\right] \\
&\overset{(b)}{=} \Pr\left[\|\mathcal{T}_{\mathrm{VaR}_\alpha}\hat{\boldsymbol{v}} - \mathcal{T}_{\widehat{\mathrm{VaR}}_\alpha}\hat{\boldsymbol{v}}\|_\infty \le \varepsilon(1-\gamma)\right] \\
&\overset{(c)}{=} 1 - \sum_{s\in\mathcal{S}} \Pr\left[|\max_{a\in\mathcal{A}}\widehat{\mathrm{VaR}}_\alpha[\tilde{\boldsymbol{p}}_{s,a}^\mathsf{T}\hat{\boldsymbol{w}}_{s,a}] - \max_{a\in\mathcal{A}}\mathrm{VaR}_\alpha[\tilde{\boldsymbol{p}}_{s,a}^\mathsf{T}\hat{\boldsymbol{w}}_{s,a}]| > \varepsilon(1-\gamma)\right] \\
&\overset{(d)}{=} 1 - \frac{\zeta}{S}S \\
&= 1 - \zeta .
\end{aligned}
$$

(30)

$(a)$ follows from the fact that $\|\hat{\boldsymbol{v}} - \hat{\boldsymbol{u}}\|_\infty \le \frac{\|\mathcal{T}_{\mathrm{VaR}_\alpha}\hat{\boldsymbol{v}} - \mathcal{T}_{\widehat{\mathrm{VaR}}_\alpha}\hat{\boldsymbol{v}}\|_\infty}{1-\gamma}$, $(b)$ follows from arranging terms, $(c)$ follows from applying the union bound, and $(d)$ follows from applying Proposition 3.6 and setting confidence to $\varsigma/s$.

The final time complexity results follow from the triangle inequality $\|\hat{\boldsymbol{v}} - \boldsymbol{u}_k\|_\infty \le \|\hat{\boldsymbol{v}} - \hat{\boldsymbol{u}}\|_\infty + \|\boldsymbol{u}_k - \hat{\boldsymbol{u}}\|_\infty$ and combining results in Equation (30), Proposition 3.6 and the first part of Proposition 3.8.

$\square$

## B.9 Proof of Proposition 4.1

**Proposition 4.1** (Equivalence). *The* $\mathrm{VaR}$ *Bellman optimality operator* $\mathcal{T}_{\mathrm{VaR}_\alpha}$ *can be expressed as*

$$\forall s\in\mathcal{S}: (\mathcal{T}_{\mathrm{VaR}_\alpha}\boldsymbol{v})_s = \max_{a\in\mathcal{A}} \min_{\boldsymbol{p}\in\mathcal{P}_{s,a}^{\mathrm{VaR},\boldsymbol{v}}} \boldsymbol{p}^\mathsf{T}(\boldsymbol{r}_{s,a} + \gamma\boldsymbol{v}) .$$

*Furthermore, the optimal* $\mathrm{VaR}$ *policy* $\hat{\pi} \in \Pi^D$ *solves* $\max_{\pi\in\Pi^D} \min_{\boldsymbol{P}\in\mathcal{P}^{\mathrm{VaR},\hat{\boldsymbol{v}}}} \rho(\pi,\boldsymbol{P})$ *, where* $\hat{\boldsymbol{v}} \in \mathbb{R}^S$ *is the fixed point of the* $\mathrm{VaR}$ *Bellman operator* $\mathcal{T}_{\mathrm{VaR}_\alpha}$, *i.e.,* $\hat{\boldsymbol{v}} = (\mathcal{T}_{\mathrm{VaR}_\alpha}\hat{\boldsymbol{v}})$.

*Proof.* The first part of the proposition simply follows from the definition of the $\mathrm{VaR}$ operator. For any state $s$, action $a$ and value function $\boldsymbol{v}$, we can write

$$\min_{\boldsymbol{p}\in\mathcal{P}_{s,a}^{\mathrm{VaR},\boldsymbol{v}}} \boldsymbol{p}^\mathsf{T}\boldsymbol{w}_{s,a} = \mathrm{VaR}_\alpha\left[\tilde{\boldsymbol{p}}_{s,a}^\mathsf{T}\boldsymbol{w}_{s,a}\right] . \tag{31}$$

The minimum above exists as it minimizes a linear function on a compact set. The direction "$\ge$" in (31) follows immediately from the constraint in the construction of the set $\mathcal{P}_{s,a}^{\mathrm{VaR},\boldsymbol{v}}$. The direction "$\le$" follows by linear program duality of the minimization over $\boldsymbol{p}$ and from the fact that $\mathrm{VaR}_\alpha\left[\tilde{\boldsymbol{p}}_{s,a}^\mathsf{T}\boldsymbol{w}_{s,a}\right] \ge \min_{s'\in\mathcal{S}}\boldsymbol{w}_{s,a,s'}$.

Recall that the $\mathrm{VaR}$ Bellman optimality operator defined for each $s\in\mathcal{S}$ and $\boldsymbol{v}\in\mathbb{R}^S$ as

$$(\mathcal{T}_{\mathrm{VaR}_\alpha}\boldsymbol{v})_s = \max_{a\in\mathcal{A}}\mathrm{VaR}_\alpha\left[\tilde{\boldsymbol{p}}_{s,a}^\mathsf{T}\boldsymbol{w}_{s,a}\right] . \tag{32}$$

Suppose that $\hat{\boldsymbol{v}}$ is the unique fixed point of $\mathcal{T}_{\mathrm{VaR}_\alpha}$ and $\hat{\boldsymbol{w}}_{s,a} = \boldsymbol{r}_{s,a} + \gamma\cdot\hat{\boldsymbol{v}}$ is the corresponding transition value for each $s\in\mathcal{S}$ and $a\in\mathcal{A}$. That is

$$\hat{\boldsymbol{v}} = \mathcal{T}_{\mathrm{VaR}_\alpha}\hat{\boldsymbol{v}} .$$

Then, we define the following robust Bellman operator $\hat{\mathcal{T}}: \mathbb{R}^S \to \mathbb{R}^S$ for each $s\in\mathcal{S}$ and $\boldsymbol{v}\in\mathbb{R}^S$ as

$$
\begin{aligned}
(\hat{\mathcal{T}}\boldsymbol{v})_s &= \max_{a\in\mathcal{A}} \min_{\boldsymbol{p}\in\mathcal{P}_{s,a}^{\mathrm{VaR},\hat{\boldsymbol{v}}}} \boldsymbol{p}^\mathsf{T}\boldsymbol{w}_{s,a} , \quad \text{where} \\
\mathcal{P}_{s,a}^{\mathrm{VaR},\hat{\boldsymbol{v}}} &= \left\{\boldsymbol{p}\in\Delta^S \mid \boldsymbol{p}^\mathsf{T}\hat{\boldsymbol{w}}_{s,a} \ge \mathrm{VaR}_\alpha\left[\tilde{\boldsymbol{p}}_{s,a}^\mathsf{T}\hat{\boldsymbol{w}}_{s,a}\right]\right\} .
\end{aligned}
\tag{33}
$$

We now show that $\hat{v}$ is also the fixed point of the robust Bellman operator $\hat{\mathcal{T}}$.

Using the equality in (31), we now get that

$$\begin{aligned}
\hat{v} &= \mathcal{T}_{\text{VaR}_\alpha} \hat{v} \\
&= \max_{a \in \mathcal{A}} \text{VaR}_\alpha \left[ \tilde{p}_{s,a}^\mathsf{T} \hat{w}_{s,a} \right] \\
&= \max_{a \in \mathcal{A}} \min_{p \in \mathcal{P}_{s,a}^{\text{VaR},\hat{v}}} p^\mathsf{T} \hat{w}_{s,a} \\
&= \hat{\mathcal{T}} \hat{v} \ .
\end{aligned}$$

Therefore, $\hat{v}$ is the unique fixed point of the SA-rectangular robust Bellman operator [24] and $\hat{\pi}$ is greedy with respect to the optimal robust value function of $\hat{v}$ and, therefore, is an optimal policy that solves

$$\max_{\pi \in \Pi^D} \min_{\boldsymbol{P} \in \mathcal{P}^{\text{VaR},\hat{v}}} \rho(\pi, \boldsymbol{P}) \ , \tag{34}$$

where $\hat{v} \in \mathbb{R}^S$ is the fixed point of the VaR Bellman operator $\mathcal{T}_{\text{VaR}_\alpha}$, i.e., $\hat{v} = (\mathcal{T}_{\text{VaR}_\alpha} \hat{v})$. $\qquad \square$

### B.10  Proof of Theorem 4.2

**Theorem 4.2** (Asymptotic Radii of VaR Ambiguity Sets)**.** *For any state $s$ and action $a$, let $I(\{(\boldsymbol{p}_{s,a}^*)_i\}_{i=1}^{S-1})$ be the Fisher information of the first $S-1$ transition probabilities, i.e., $\{(\boldsymbol{p}_{s,a}^*)_i\}_{i=1}^{S-1}$. Define $I'(\boldsymbol{p}_{s,a}^*) = \begin{bmatrix} I(\{(\boldsymbol{p}_{s,a}^*)_i\}_{i=1}^{S-1}) & \mathbf{0} \\ \mathbf{0}^\mathsf{T} & 0 \end{bmatrix}$. Suppose that the normality assumptions on the posterior distribution $\tilde{\boldsymbol{P}}$ in Section 2 are satisfied. Then,*

$$\lim_{N \to \infty} \sqrt{N}(\mathcal{P}_{s,a}^{\text{VaR}_\alpha} - \boldsymbol{p}_{s,a}^*) = \left\{ \boldsymbol{p}_{s,a} \in \Delta^S \middle| \|\boldsymbol{p}_{s,a} - \boldsymbol{p}_{s,a}^*\|_{I'(\boldsymbol{p}_{s,a}^*)^{-1}} \leq \Phi^{-1}(1-\alpha) \right\} - \boldsymbol{p}_{s,a}^* \ . \tag{8}$$

For any state $s$ and action $a$, we define the 1-step Bellman update function $f_{s,a} : \mathbb{R}^S \to \mathbb{R}$ such that

$$f_{s,a}(\boldsymbol{v}) = \text{VaR}_\alpha \left[ \tilde{\boldsymbol{p}}_{s,a}^\mathsf{T}(\boldsymbol{r}_{s,a} + \gamma \boldsymbol{v}) \right],$$

As noted in (2), we assume that as $N \to \infty$, the posterior distribution of transition probabilities for any state $s$ and action $a$ $(\tilde{\boldsymbol{p}}_{s,a})$ converges in the limit to a multivariate Gaussian distribution with mean $\boldsymbol{p}_{s,a}^*$ and degenerate covariance matrix $I(\boldsymbol{p}_{s,a}^*)/N$. Therefore, the 1-step returns $\tilde{\boldsymbol{p}}_{s,a}^\mathsf{T} \boldsymbol{w}_{s,a}$ is asymptotically a univariate Gaussian random variable, i.e., $\lim_{N \to \infty} \sqrt{N}(\tilde{\boldsymbol{p}}_{s,a}^\mathsf{T} \boldsymbol{w}_{s,a} - \boldsymbol{p}_{s,a}^{*\mathsf{T}} \boldsymbol{w}_{s,a}) \sim \mathcal{N}\left(0, \boldsymbol{w}_{s,a}^\mathsf{T} I^{-1}(\boldsymbol{p}_{s,a}^*) \boldsymbol{w}_{s,a}\right)$. Then, we can write

$$\lim_{N \to \infty} \sqrt{N}\left(f_{s,a}(\boldsymbol{v}) - \boldsymbol{p}_{s,a}^{*\mathsf{T}} \boldsymbol{w}_{s,a}\right) = -\Phi^{-1}(1-\alpha)\|\boldsymbol{w}_{s,a}\|_{I(\boldsymbol{p}_{s,a}^*)} \ , \tag{35}$$

where $\Phi^{-1}$ represents the CDF inverse of a standard normal distribution. Equation (35) follows from the analytical form of VaR of a Gaussian random variable, i.e., for any Gaussian random variable $\tilde{Y} \sim \mathcal{N}(\mu, \sigma^2)$ with mean $\mu$, variance $\sigma^2$, and confidence level $\alpha$, $\text{VaR}_\alpha[\tilde{Y}] = \mu - \Phi^{-1}(1-\alpha)\sigma$.

Since $f_{s,a}$ is convex in $\boldsymbol{v}$ when $\tilde{\boldsymbol{p}}_{s,a}$ is Multivariate Gaussian distributed, we can use the definition of support function of a closed convex set [9] to retrieve the unique ambiguity set $\mathcal{P}_{s,a}^{\text{VaR}}$.

Note that, in this case the support function is $f_{s,a}$ and $\mathcal{P}_{s,a}^{\text{VaR}_\alpha}$ is the unknown closed convex set

$$\forall \boldsymbol{v} \in \mathbb{R}^S : \min_{\boldsymbol{p}_{s,a} \in \mathcal{P}_{s,a}^{\text{VaR}_\alpha}} \boldsymbol{p}_{s,a}^\mathsf{T}(\boldsymbol{r}_{s,a} + \gamma \boldsymbol{v}) = f_{s,a}(\boldsymbol{v})$$

$$\forall \boldsymbol{v} \in \mathbb{R}^S : \min_{\boldsymbol{p}_{s,a} \in \mathcal{P}_{s,a}^{\text{VaR}_\alpha}} \sqrt{N} \boldsymbol{p}_{s,a}^\mathsf{T} \boldsymbol{w}_{s,a} = \sqrt{N} f_{s,a}(\boldsymbol{v})$$

$$\forall \boldsymbol{v} \in \mathbb{R}^S : \min_{\boldsymbol{p}_{s,a} \in \mathcal{P}_{s,a}^{\text{VaR}_\alpha}} \sqrt{N}(\boldsymbol{p}_{s,a}^\mathsf{T} \boldsymbol{w}_{s,a} - \boldsymbol{p}_{s,a}^{*\mathsf{T}} \boldsymbol{w}_{s,a}) = \sqrt{N}(f_{s,a}(\boldsymbol{v}) - \boldsymbol{p}_{s,a}^{*\mathsf{T}} \boldsymbol{w}_{s,a})$$

$$\forall \boldsymbol{v} \in \mathbb{R}^S, \boldsymbol{z}_{s,a} \in \left(\sqrt{N}(\mathcal{P}_{s,a}^{\text{VaR}_\alpha} - \boldsymbol{p}_{s,a}^*)\right) : \boldsymbol{z}_{s,a}^\mathsf{T} \boldsymbol{w}_{s,a} \geq -\Phi^{-1}(1-\alpha)\|\boldsymbol{w}_{s,a}\|_{I(\boldsymbol{p}_{s,a}^*)} \ ,$$

where the last statement follows from the positive homogeneity and translation property of support functions.

Rearranging the terms in the above and setting $\boldsymbol{z}_{s,a} = \boldsymbol{p}_{s,a} - \boldsymbol{p}^*_{s,a}$, we get

$$\lim_{N\to\infty} \sqrt{N}(\mathcal{P}^{\mathrm{VaR}\alpha}_{s,a} - \boldsymbol{p}^*_{s,a}) = \left\{ \boldsymbol{p}_{s,a} - \boldsymbol{p}^*_{s,a} \Big| \forall \boldsymbol{v} \in \mathbb{R}^S,\, \boldsymbol{p}_{s,a} \in \Delta^S,\, \boldsymbol{p}^\mathsf{T}_{s,a}\boldsymbol{w}_{s,a} \geq \boldsymbol{p}^*_{s,a}{}^\mathsf{T}\boldsymbol{w}_{s,a} - \Phi^{-1}(1-\alpha)\|\boldsymbol{w}_{s,a}\|_{I(\boldsymbol{p}^*_{s,a})} \right\} \ . \tag{36}$$

Using basic algebraic manipulations as shown in Proposition B.1, we can write the asymptotic form of ambiguity set $\mathcal{P}^{\mathrm{VaR}\alpha}_{s,a}$ as an ellipsoid

$$\lim_{N\to\infty} \sqrt{N}(\mathcal{P}^{\mathrm{VaR}\alpha}_{s,a} - \boldsymbol{p}^*_{s,a}) = \left\{ \boldsymbol{p}_{s,a} \in \Delta^S \Big| \|\boldsymbol{p}^*_{s,a} - \boldsymbol{p}_{s,a}\|_{I'(\boldsymbol{p}^*_{s,a})^{-1}} \leq \Phi^{-1}(1-\alpha) \right\} - \boldsymbol{p}^*_{s,a} \ . \tag{37}$$

**Proposition B.1.** *Consider the two ambiguity sets given in equations* (36) *and* (37). *These two representations are equivalent.*

*Proof.* For any state $s$ and action $a$, as $N \to \infty$, $(\boldsymbol{p}_{s,a} - \boldsymbol{p}^*_{s,a}) \in \sqrt{N}(\mathcal{P}^{\mathrm{VaR}\alpha}_{s,a} - \boldsymbol{p}^*_{s,a})$ satisfies

$$\boldsymbol{p}^\mathsf{T}_{s,a}\boldsymbol{w}_{s,a} \geq \boldsymbol{p}^*_{s,a}{}^\mathsf{T}\boldsymbol{w}_{s,a} - \Phi^{-1}(1-\alpha)\|\boldsymbol{w}_{s,a}\|_{I(\boldsymbol{p}^*_{s,a})}$$

$$\boldsymbol{p}^\mathsf{T}_{s,a}\boldsymbol{w}_{s,a} - \boldsymbol{p}^*_{s,a}{}^\mathsf{T}\boldsymbol{w}_{s,a} \geq -\Phi^{-1}(1-\alpha)\|\boldsymbol{w}_{s,a}\|_{I(\boldsymbol{p}^*_{s,a})} \tag{38}$$

$$\boldsymbol{w}^\mathsf{T}_{s,a}(\boldsymbol{p}_{s,a} - \boldsymbol{p}^*_{s,a})(\boldsymbol{p}_{s,a} - \boldsymbol{p}^*_{s,a})^\mathsf{T}\boldsymbol{w}_{s,a} \leq (\Phi^{-1}(1-\alpha))^2 \boldsymbol{w}^\mathsf{T}_{s,a}I(\boldsymbol{p}^*_{s,a})^{-1}\boldsymbol{w}_{s,a} \ .$$

The first and second equations follow from the definition of $\mathcal{P}^{\mathrm{VaR}}_{s,a}$ and simple algebraic manipulations. The third equation follows by multiplying by $-1$ on both sides and squaring both sides.

Recall that $\tilde{\boldsymbol{p}}_{s,a}$ for any state $s$ and action $a$ is a categorical random variable. We assumed that the prior is a Dirichlet distribution and therefore, by the property of conjugate prior, the posterior distribution of $\tilde{\boldsymbol{p}}_{s,a}$ is also a Dirichlet distribution. We assumed in Section 2 that as $N \to \infty$, $\tilde{\boldsymbol{p}}_{s,a}$ is Gaussian-distributed with mean $\boldsymbol{p}^*_{s,a}$ and degenerate covariance matrix $I(\boldsymbol{p}^*_{s,a})^{-1}/N$, i.e., covariance matrix $I(\boldsymbol{p}^*_{s,a})^{-1}/N$ has $S-1$ independent rows and $S-1$ independent columns.

For any state $s$ and action $a$ and transition probabilities $\tilde{\boldsymbol{p}}_{s,a}$, let $\tilde{\boldsymbol{q}}_{s,a} \in \mathbb{R}^{S-1}$ represent the first $S-1$ elements of the random variable $\tilde{\boldsymbol{p}}_{s,a}$, i.e., $\tilde{\boldsymbol{q}}_{s,a} = \{(\tilde{\boldsymbol{p}}_{s,a})_i\}_{i=1}^{S-1}$. Then, $\tilde{\boldsymbol{q}}_{s,a}$ is Gaussian-distributed with the mean $\boldsymbol{q}^*_{s,a} = \{(\boldsymbol{p}^*_{s,a})_i\}_{i=1}^{S-1}$ and invertible covariance matrix $I(\boldsymbol{q}^*_{s,a})^{-1}/N = I(\{(\boldsymbol{p}^*_{s,a})_i\}_{i=1}^{S-1})^{-1})^{-1}/N$.

Thus, if we sample $\boldsymbol{q}_{s,a}$ from $\mathcal{N}\left(\boldsymbol{q}^*_{s,a}, I(\boldsymbol{q}^*_{s,a})^{-1}/N\right)$ and $\boldsymbol{q}_{s,a} \succeq \boldsymbol{0}$ and $\sum_{i=1}^{S-1}(\boldsymbol{q}_{s,a})_i \leq 1$, then we can easily compute the corresponding transition model as

$$\{(\boldsymbol{q}_{s,a})_i\}_{i=1}^{S-1} = \{(\boldsymbol{p}_{s,a})_i\}_{i=1}^{S-1}$$

$$(\boldsymbol{q}_{s,a})_S = 1 - \sum_{i=1}^{S-1}(\boldsymbol{p}_{s,a})_i \ . \tag{39}$$

Equipped with the above notation, we can proceed to prove the result.

Using the definition of positive semi-definite matrices in Equation (38), we can write

$$\left((\Phi^{-1}(1-\alpha))^2 I(\boldsymbol{p}^*_{s,a})^{-1} - (\boldsymbol{p}_{s,a} - \boldsymbol{p}^*_{s,a})(\boldsymbol{p}_{s,a} - \boldsymbol{p}^*_{s,a})^\mathsf{T}\right) \succeq 0 \ . \tag{40}$$

Sub-matrices of positive-semidefinite matrices are also positive-semidefinite. Thus, we can write,

$$\left((\Phi^{-1}(1-\alpha))^2 I(\boldsymbol{q}^*_{s,a})^{-1} - (\boldsymbol{q}_{s,a} - \boldsymbol{q}^*_{s,a})(\boldsymbol{q}_{s,a} - \boldsymbol{q}^*_{s,a})^\mathsf{T}\right) \overset{(a)}{\succeq} 0$$

$$\left(I(\boldsymbol{q}^*_{s,a})\right)\left((\Phi^{-1}(1-\alpha))^2 I(\boldsymbol{q}^*_{s,a})^{-1} - (\boldsymbol{q}_{s,a} - \boldsymbol{q}^*_{s,a})(\boldsymbol{q}_{s,a} - \boldsymbol{q}^*_{s,a})^\mathsf{T}\right)\left(I(\boldsymbol{q}^*_{s,a})\right)^\mathsf{T} \overset{(b)}{\succeq} 0$$

$$\left((\Phi^{-1}(1-\alpha))^2 I(\boldsymbol{q}^*_{s,a})I(\boldsymbol{q}^*_{s,a})^{-1} - (\boldsymbol{q}_{s,a} - \boldsymbol{q}^*_{s,a})(\boldsymbol{q}_{s,a} - \boldsymbol{q}^*_{s,a})^\mathsf{T}\right)I(\boldsymbol{q}^*_{s,a})^\mathsf{T} \overset{(c)}{\succeq} 0$$

$$\left((\boldsymbol{q}_{s,a} - \boldsymbol{q}^*_{s,a})^\mathsf{T}I(\boldsymbol{q}^*_{s,a})(\boldsymbol{q}_{s,a} - \boldsymbol{q}^*_{s,a})\right)\left((\Phi^{-1}(1-\alpha))^2 I \right.$$

$$\left. - (\boldsymbol{q}_{s,a} - \boldsymbol{q}^*_{s,a})(\boldsymbol{q}_{s,a} - \boldsymbol{q}^*_{s,a})^\mathsf{T}\right)I(\boldsymbol{q}^*_{s,a})^\mathsf{T}\left((\boldsymbol{q}_{s,a} - \boldsymbol{q}^*_{s,a})^\mathsf{T}I(\boldsymbol{q}^*_{s,a})((\boldsymbol{q}_{s,a} - \boldsymbol{q}^*_{s,a}))\right)^\mathsf{T} \overset{(d)}{\succeq} 0$$

$$(\Phi^{-1}(1-\alpha))^2(\boldsymbol{q}_{s,a} - \boldsymbol{q}^*_{s,a})^\mathsf{T}I(\boldsymbol{q}^*_{s,a})(\boldsymbol{q}_{s,a} - \boldsymbol{q}^*_{s,a}) - \left((\boldsymbol{q}_{s,a} - \boldsymbol{q}^*_{s,a})^\mathsf{T}I(\boldsymbol{q}^*_{s,a})(\boldsymbol{q}_{s,a} - \boldsymbol{q}^*_{s,a})\right)^2 \overset{(e)}{\succeq} 0 \ .$$

($a$) holds because for any $\boldsymbol{A} \in \mathbb{R}^{n \times n}$ if $\boldsymbol{x}^\mathsf{T} \boldsymbol{A} \boldsymbol{x} \succeq 0 \, \forall \boldsymbol{x} \in \mathbb{R}^n$, then $\boldsymbol{A} \succeq 0$, ($b$) follows from $\boldsymbol{U}^\mathsf{T} \boldsymbol{M} \boldsymbol{U} \succeq 0 \, \forall \boldsymbol{U}, \boldsymbol{M} \succeq 0$, ($c$) follows from simple algebraic manipulations, ($d$) follows from $(\boldsymbol{q}_{s,a} - \boldsymbol{q}_{s,a}^*)^\mathsf{T} I(\boldsymbol{q}_{s,a}^*)(\boldsymbol{q}_{s,a} - \boldsymbol{q}_{s,a}^*) \succeq 0$ because $I(\boldsymbol{q}_{s,a}^*)^{-1} \succeq 0$, and ($e$) follows from simply rearranging all the terms in the above equation.

Therefore, we get

$$(\boldsymbol{q}_{s,a} - \boldsymbol{q}_{s,a}^*)^\mathsf{T} I(\boldsymbol{p}_{s,a}^*)(\boldsymbol{q}_{s,a} - \boldsymbol{q}_{s,a}^*) \leq (\Phi^{-1}(1-\alpha))^2 \implies \|\boldsymbol{q}_{s,a} - \boldsymbol{q}_{s,a}^*\|_{I(\boldsymbol{q}_{s,a}^*)^{-1}} \leq \Phi^{-1}(1-\alpha) \ .$$

Thus, it follows from construction that $\|\boldsymbol{p}_{s,a} - \boldsymbol{p}_{s,a}^*\|_{I'(\boldsymbol{p}_{s,a}^*)^{-1}} \leq \Phi^{-1}(1-\alpha)$.

The proof for the other direction is simply the reverse of this proof and therefore we omit it. $\qquad\square$

### B.11  Proof of Theorem 4.3

**Theorem 4.3** (Asymptotic Radius of Bayesian Credible Regions). *For any state $s$ and action $a$, let $\mathcal{P}_{s,a}^{BCR_\alpha}$ represent any Bayesian credible region. Let $\xi < \sqrt{\chi_{S-1,1-\alpha}^2}/\Phi^{-1}(1-\alpha)$. Suppose that the normality assumptions on the posterior distribution of $\tilde{\boldsymbol{P}}$ in Section 2 are satisfied. Then,*

$$\forall s \in \mathcal{S}, a \in \mathcal{A} : \lim_{N\to\infty} \sqrt{N}(\mathcal{P}_{s,a}^{BCR_\alpha} - \boldsymbol{p}_{s,a}^*) \not\subseteq \lim_{N\to\infty} \sqrt{N}\xi(\mathcal{P}_{s,a}^{\mathrm{VaR}_\alpha} - \boldsymbol{p}_{s,a}^*) \ . \tag{9}$$

*Proof.* For any state $s$ and action $a$ and transition probabilities $\tilde{\boldsymbol{p}}_{s,a}$, let $\tilde{\boldsymbol{q}}_{s,a} \in \mathbb{R}^{S-1}$ represent the first $S-1$ elements of the random variable $\tilde{\boldsymbol{p}}_{s,a}$, i.e., $\tilde{\boldsymbol{q}}_{s,a} = \{(\tilde{\boldsymbol{p}}_{s,a})_i\}_{i=1}^{S-1}$. Then, $\tilde{\boldsymbol{q}}_{s,a}$ is Gaussian-distributed with the mean $\boldsymbol{q}_{s,a}^* = \{(\boldsymbol{p}_{s,a}^*)_i\}_{i=1}^{S-1}$ and invertible covariance matrix $I(\boldsymbol{q}_{s,a}^*)^{-1}/N = I(\{(\boldsymbol{p}^*_{s,a})_i\}_{i=1}^{S-1})^{-1})^{-1}/N$.

Thus, if we sample $\boldsymbol{q}_{s,a}$ from $\mathcal{N}\left(\boldsymbol{q}_{s,a}^*, I(\boldsymbol{q}_{s,a}^*)^{-1}/N\right)$ and $\boldsymbol{q}_{s,a} \succeq \boldsymbol{0}$ and $\sum_{i=1}^{S-1}(\boldsymbol{q}_{s,a})_i \leq 1$, then we can easily compute the corresponding transition model as

$$\begin{aligned} \{(\boldsymbol{q}_{s,a})_i\}_{i=1}^{S-1} &= \{(\boldsymbol{p}_{s,a})_i\}_{i=1}^{S-1} \\ (\boldsymbol{q}_{s,a})_S &= 1 - \sum_{i=1}^{S-1}(\boldsymbol{p}_{s,a})_i \ . \end{aligned} \tag{41}$$

Equipped with the above notation, we will now prove this theorem by contradiction.

For any state $s$ and action $a$, suppose that $\lim_{N\to\infty} \sqrt{N}(\mathcal{P}_{s,a}^{BCR_\alpha} - \boldsymbol{p}_{s,a}^*) \subseteq \lim_{N\to\infty} \sqrt{N}\xi(\mathcal{P}_{s,a}^{\mathrm{VaR}_\alpha} - \boldsymbol{p}_{s,a}^*)$.

Since $\mathcal{P}_{s,a}^{BCR_\alpha}$ is a credible region, it satisfies that

$$\begin{aligned} 1 - \alpha &\overset{(a)}{\leq} \Pr\left[(\tilde{\boldsymbol{p}}_{s,a} - \boldsymbol{p}_{s,a}^*) \in \mathcal{P}_{s,a}^{BCR_\alpha} - \boldsymbol{p}_{s,a}^*\right] \\ &\overset{(b)}{=} \Pr\left[\lim_{N\to\infty} \sqrt{N}(\tilde{\boldsymbol{p}}_{s,a} - \boldsymbol{p}_{s,a}^*) \in \lim_{N\to\infty} \sqrt{N}(\mathcal{P}_{s,a}^{BCR_\alpha} - \boldsymbol{p}_{s,a}^*)\right] \\ &\overset{(c)}{=} \Pr\left[\lim_{N\to\infty} \sqrt{N}(\tilde{\boldsymbol{p}}_{s,a} - \boldsymbol{p}_{s,a}^*) \in \lim_{N\to\infty} \sqrt{N}\xi(\mathcal{P}_{s,a}^{\mathrm{VaR}_\alpha} - \boldsymbol{p}_{s,a}^*)\right] \\ &\overset{(d)}{=} \Pr\left[\lim_{N\to\infty} \sqrt{N}(\tilde{\boldsymbol{p}}_{s,a} - \boldsymbol{p}_{s,a}^*)^\mathsf{T} I'(\boldsymbol{p}_{s,a}^*)\sqrt{I'(\boldsymbol{p}_{s,a}^*)^{-1}N}(\tilde{\boldsymbol{p}}_{s,a} - \boldsymbol{p}_{s,a}^*) \leq \xi^2(\Phi^{-1}(1-\alpha))^2\right] \ . \end{aligned}$$

($a$) follows from the definition of Bayesian credible regions, ($b$) follows from multiplying by $\sqrt{N}$ on both sides and taking limit $N \to \infty$, ($c$) follows from the assumption $\lim_{N\to\infty} \sqrt{N}(\mathcal{P}_{s,a}^{BCR_\alpha} - \boldsymbol{p}_{s,a}^*) \subseteq \lim_{N\to\infty} \sqrt{N}\xi(\mathcal{P}_{s,a}^{\mathrm{VaR}_\alpha} - \boldsymbol{p}_{s,a}^*)$, and ($d$) follows from applying results in (37) and squaring both sides.

Since $\xi < \frac{\sqrt{\chi^2_{S-1,1-\alpha}}}{\Phi^{-1}(1-\alpha)}$, there exists $\beta > 0$, such that $\xi^2(\Phi^{-1}(1-\alpha))^2 \leq \chi^2_{S-1,1-\beta-\alpha} - \beta$. Fix such $\beta$. Then we can write

$$
\begin{aligned}
1 - \alpha &\leq \Pr\left[\lim_{N\to\infty} \sqrt{N}(\tilde{\boldsymbol{p}}_{s,a} - \boldsymbol{p}^*_{s,a})^\mathsf{T} I'(\boldsymbol{p}^*_{s,a})\sqrt{N}(\tilde{\boldsymbol{p}}_{s,a} - \boldsymbol{p}^*_{s,a}) \leq \xi^2(\Phi^{-1}(1-\alpha))^2\right] \\
&\overset{(a)}{\leq} \Pr\left[\lim_{N\to\infty} \sqrt{N}(\tilde{\boldsymbol{p}}_{s,a} - \boldsymbol{p}^*_{s,a})^\mathsf{T} I'(\boldsymbol{p}^*_{s,a})\sqrt{N}(\tilde{\boldsymbol{p}}_{s,a} - \boldsymbol{p}^*_{s,a}) \leq (\chi^2_{S-1,1-\beta-\alpha} - \beta)\right] \\
&\overset{(b)}{\leq} \Pr\left[\lim_{N\to\infty} \|\sqrt{N}(\tilde{\boldsymbol{p}}_{s,a} - \boldsymbol{p}^*_{s,a})\|^2_{I'(\boldsymbol{p}^*_{s,a})^{-1}} \leq (\chi^2_{S-1,1-\beta-\alpha} - \beta)\right] \\
&\overset{(c)}{\leq} \Pr\left[\lim_{N\to\infty} \|\sqrt{N}(\tilde{\boldsymbol{p}}_{s,a} - \boldsymbol{p}^*_{s,a})\|^2_{I'(\boldsymbol{p}^*_{s,a})^{-1}} \leq \chi^2_{S-1,1-\beta-\alpha}\right] \\
&\overset{(d)}{\leq} 1 - \beta - \alpha < 1 - \alpha \quad \text{Contradiction!} .
\end{aligned}
$$

$(a)$ follows from choosing $\beta > 0$ such that $\xi^2(\Phi^{-1}(1-\alpha))^2 \leq \chi^2_{S-1,1-\beta-\alpha} - \beta$, $(b)$ follows from simple algebraic manipulations, $c$ follows because of the monotonic property of CDF, and $(d)$ follows because $\lim_{N\to\infty} \sqrt{N}(\tilde{\boldsymbol{p}}_{s,a} - \boldsymbol{p}^*_{s,a}) \sim \mathcal{N}(\boldsymbol{0}, I(\boldsymbol{p}^*_{s,a}))$, from construction, $\lim_{N\to\infty} \|\sqrt{N}(\tilde{\boldsymbol{p}}_{s,a} - \boldsymbol{p}^*_{s,a})\|^2_{I'(\boldsymbol{p}^*_{s,a})^{-1}} = \lim_{N\to\infty} \|\sqrt{N}(\tilde{\boldsymbol{q}}_{s,a} - \boldsymbol{q}^*_{s,a})\|^2_{I(\boldsymbol{q}^*_{s,a})^{-1}}$ and, $\lim_{N\to\infty} \|\sqrt{N}(\tilde{\boldsymbol{q}}_{s,a} - \boldsymbol{q}^*_{s,a})\|^2_{I(\boldsymbol{q}^*_{s,a})^{-1}}$ is a sum of squares of $S-1$ normal random variables, which in turn is a standard Chi-squared random variable with degrees of freedom $S-1$. Therefore, the probability in $(d)$ can be at most $1 - \beta - \alpha$.

Therefore, $\xi \geq \frac{\sqrt{\chi^2_{S-1,1-\alpha}}}{\Phi^{-1}(1-\alpha)}$.

$\square$

## C Experiments

---

**Algorithm C.1:** VaR Value Iteration Algorithm for Gaussian case

---

**Input:** Confidence $\alpha \in (0, 1/2)$, Posterior distribution $f$, Value approximation error $\varepsilon \geq 0$
**Output:** Robust policy $\pi_k$, Lower bound $\boldsymbol{u}_k$

1   Initialize robust value-function $\boldsymbol{u}_0 \in \mathbb{R}^S$, $k = 0$
2   Sample $M$ models $\tilde{\boldsymbol{P}}(\omega_1), \tilde{\boldsymbol{P}}(\omega_2), \ldots, \tilde{\boldsymbol{P}}(\omega_M)$ from posterior $f$
3   **repeat**
4      **for** $s \leftarrow 1$ **to** $S$ **do**
5          **for** $a \leftarrow 1$ **to** $A$ **do**
6              $\boldsymbol{w}_{s,a} \leftarrow \boldsymbol{r}_{s,a} + \gamma \boldsymbol{u}_k$                      // 1-step return from $(s, a)$
7              $q_a \leftarrow \bar{\boldsymbol{p}}^\mathsf{T}_{s,a} \boldsymbol{w}_{s,a} - \Phi^{-1}(1-\alpha)\sqrt{\boldsymbol{w}^\mathsf{T}_{s,a} \boldsymbol{\Sigma}_{s,a} \boldsymbol{w}_{s,a}}$    // $\bar{\boldsymbol{p}}_{s,a}, \boldsymbol{\Sigma}_{s,a}$ are mean and covariance of $f$ at $(s,a)$
8          **end**
9      **end**
10      $k \leftarrow k + 1$
11      $\boldsymbol{u}_k(s) \leftarrow \max_{a\in\mathcal{A}}(\boldsymbol{q}_a), \quad \pi_k(s) \leftarrow \arg\max_{a\in\mathcal{A}}(\boldsymbol{q}_a)$            // $\text{VaR}_\alpha$ Bellman optimality update
12   **until** $\|\boldsymbol{u}_k - \boldsymbol{u}_{k-1}\|_\infty \leq \varepsilon(1-\gamma)/\gamma$
13   **return** $\pi_k, \boldsymbol{u}_k$

---

## D Scalability of the VaR Framework

We define the $\text{VaR}_\alpha$ q-value function (Q-function) for any policy $\pi \in \Pi_D$ as $\boldsymbol{q}^\pi : S \times A \to \mathbb{R}$ such that for any state $s$ and action $a$, $\boldsymbol{q}^\pi(s, a) = \text{VaR}_\alpha[\boldsymbol{p}^\mathsf{T}_{s,a}(\boldsymbol{r}_{s,a} + \gamma \boldsymbol{v}^\pi)]$, where $\boldsymbol{v}^\pi = \mathcal{T}^\pi_{\text{VaR}_\alpha} \boldsymbol{v}^\pi$ is the fixed point of the $\text{VaR}_\alpha$ Bellman evaluation operator for policy $\pi$. We will use $\hat{\boldsymbol{q}}$ to denote the q-value function corresponding to the optimal $\text{VaR}_\alpha$ policy $\hat{\pi}$, i.e., $\hat{\pi}(s) = \arg\max_{a\in\mathcal{A}} \hat{\boldsymbol{q}}(s, a)$.

Let $\hat{\boldsymbol{q}}_{\boldsymbol{\theta}}$ and $\hat{\boldsymbol{q}}_{\bar{\boldsymbol{\theta}}}$ denote the parameterized Q-value and corresponding target Q-value networks with parameters $\boldsymbol{\theta} \in \Theta$ and $\bar{\boldsymbol{\theta}} \in \Theta$ respectively. Here $\hat{\boldsymbol{q}}_{\boldsymbol{\theta}}$ and $\hat{\boldsymbol{q}}_{\bar{\boldsymbol{\theta}}}$ may represent any function approximators with parameter space $\Theta$.

| Methods | Riverswim | Inventory | Population-Small | Population |
|---|---|---|---|---|
| VaR | $83.51 \pm 11.76$ | $474.98 \pm 0.7$ | $\mathbf{-1877.66 \pm 104.64}$ | $\mathbf{-3338.26 \pm 213.44}$ |
| VaRN | $92.32 \pm 11.76$ | $472.49 \pm 2.24$ | $-2271.13 \pm 165.98$ | $-3140.68 \pm 122.72$ |
| BCR $l_1$ | $82.04 \pm 8.82$ | $377.73 \pm 0.0$ | $-3731.29 \pm 122.4$ | $-4363.46 \pm 240.62$ |
| BCR $l_\infty$ | $80.57 \pm 0.0$ | $199.82 \pm 39.5$ | $-6668.47 \pm 43.1$ | $-8118.68 \pm 327.74$ |
| WBCR $l_1$ | $82.04 \pm 8.82$ | $470.39 \pm 5.82$ | $-3286.26 \pm 1115.22$ | $-4257.33 \pm 194.4$ |
| WBCR $l_\infty$ | $80.57 \pm 0.0$ | $199.82 \pm 39.5$ | $-6395.77 \pm 88.1$ | $-7547.52 \pm 160.76$ |
| Soft-Robust | $115.33 \pm 1.16$ | $477.81 \pm 0.0$ | $-2052.83 \pm 78.2$ | $-3869.46 \pm 124.72$ |
| Naive Hoeffding | $52.29 \pm 9.18$ | $-0.0 \pm 0.0$ | $-7778.21 \pm 89.36$ | $-8496.47 \pm 205.54$ |
| Opt Hoeffding | $51.15 \pm 6.88$ | $-0.0 \pm 0.0$ | $-7723.6 \pm 4.82$ | $-8583.83 \pm 16.06$ |

Table 2: shows the 95% confidence interval of the robust (percentile) returns achieved by *VaR*, *VaRN*, *BCR $\ell_1$*, *BCR $\ell_\infty$*, *WBCR $\ell_1$*, *WBCR*, *Soft Robust*, *Naive Hoeffding* and *Opt Hoeffding* agents at $\delta = 0.15$ in Riverswim, Inventory, Population-Small, and Population domain.

| Methods | Riverswim | Inventory | Population-Small | Population |
|---|---|---|---|---|
| VaR | $100.27 \pm 17.24$ | $483.08 \pm 0.4$ | $\mathbf{-1117.57 \pm 240.02}$ | $-1850.26 \pm 191.08$ |
| BCR $l_1$ | $108.9 \pm 21.12$ | $391.39 \pm 34.28$ | $-2578.25 \pm 104.04$ | $-3058.77 \pm 972.58$ |
| BCR $l_\infty$ | $95.96 \pm 0.0$ | $254.43 \pm 44.82$ | $-5437.67 \pm 46.26$ | $-6428.03 \pm 58.36$ |
| WBCR $l_1$ | $108.9 \pm 21.12$ | $481.07 \pm 4.58$ | $-2251.96 \pm 685.08$ | $-2492.59 \pm 228.48$ |
| WBCR $l_\infty$ | $95.96 \pm 0.0$ | $239.76 \pm 0.0$ | $-5133.85 \pm 85.44$ | $-5944.77 \pm 192.38$ |
| VaRN | $123.78 \pm 15.34$ | $482.92 \pm 1.18$ | $-1514.44 \pm 24.62$ | $-1785.27 \pm 125.52$ |
| Soft-Robust | $165.15 \pm 0.54$ | $484.53 \pm 0.0$ | $-692.08 \pm 55.04$ | $-1565.09 \pm 76.24$ |
| Naive Hoeffding | $53.31 \pm 13.26$ | $-0.0 \pm 0.0$ | $-7326.64 \pm 77.52$ | $-7560.94 \pm 295.96$ |
| Opt Hoeffding | $51.66 \pm 9.94$ | $-0.0 \pm 0.0$ | $-7020.42 \pm 4.74$ | $-7457.76 \pm 12.44$ |

Table 3: shows the 95% confidence interval of the robust (percentile) returns achieved by *VaR*, *VaRN*, *BCR $\ell_1$*, *BCR $\ell_\infty$*, *WBCR $\ell_1$*, *WBCR*, *Soft Robust*, *Naive Hoeffding* and *Opt Hoeffding* agents at $\delta = 0.30$ in Riverswim, Inventory, Population-Small, and Population domain. Bolded text indicates the instances in which the VaR framework outperforms the other baselines in terms of the mean robust performance.

The optimal Q-value network ($\hat{q}_{\boldsymbol{\theta}^*}$) can be learned by simply minimizing the empirical $\text{VaR}_\alpha$ Bellman residual error $J_{\text{VaR}_\alpha}(\boldsymbol{\theta})$, i.e.,

$$
\begin{aligned}
\hat{q}_{\boldsymbol{\theta}^*} &= \arg\min_{\theta \in \Theta} J_{\text{VaR}_\alpha}(\boldsymbol{\theta}) \\
&= \arg\min_{\boldsymbol{\theta} \in \Theta} \mathbb{E}_{(s,a) \sim \mathcal{D}} \left[ \left( q_{\boldsymbol{\theta}}(s,a) - \max_{a' \in \mathcal{A}} \widehat{\text{VaR}_\alpha}[\mathbb{E}_{s' \sim \tilde{\boldsymbol{p}}_{s,a}} \left[ \boldsymbol{r}_{s,a} + \gamma \bar{\boldsymbol{q}}_{\boldsymbol{\theta}}(s', a') \right]] \right)^2 \right] ,
\end{aligned}
\tag{42}
$$

where data $\mathcal{D}$ is a list state-action (s, a) tuples that is collected using the current policy on the mean model $\bar{\boldsymbol{P}}$. In this case, the target Q-value network parameters $\bar{\boldsymbol{\theta}}$ is periodically updated by overwriting the target q-value network parameter with the Q-value network parameters ($\boldsymbol{\theta}$).

We can also using any of the existing Actor-Critic methods [22, 42] to learn the optimal $\text{VaR}_\alpha$ policy. In this case, instead of optimizing the policy to minimize the Bellman residual error, the algorithm will have to optimize the policy to minimize the $\text{VaR}_\alpha$ Bellman residual error $J_{\text{VaR}_\alpha}(\boldsymbol{\theta})$.

# E   Implementation Details

| Hyperparameters for Riverswim Domain | |
|---|---|
| Number of train models per dataset (M) | 80 |
| Number of test models (K) | 700 |
| Number of train datasets (L) | 10 |

| Hyperparameters for Inventory Domain | |
|---|---|
| Number of train models per dataset (M) | 80 |
| Number of test models (K) | 200 |
| Number of train datasets (L) | 10 |

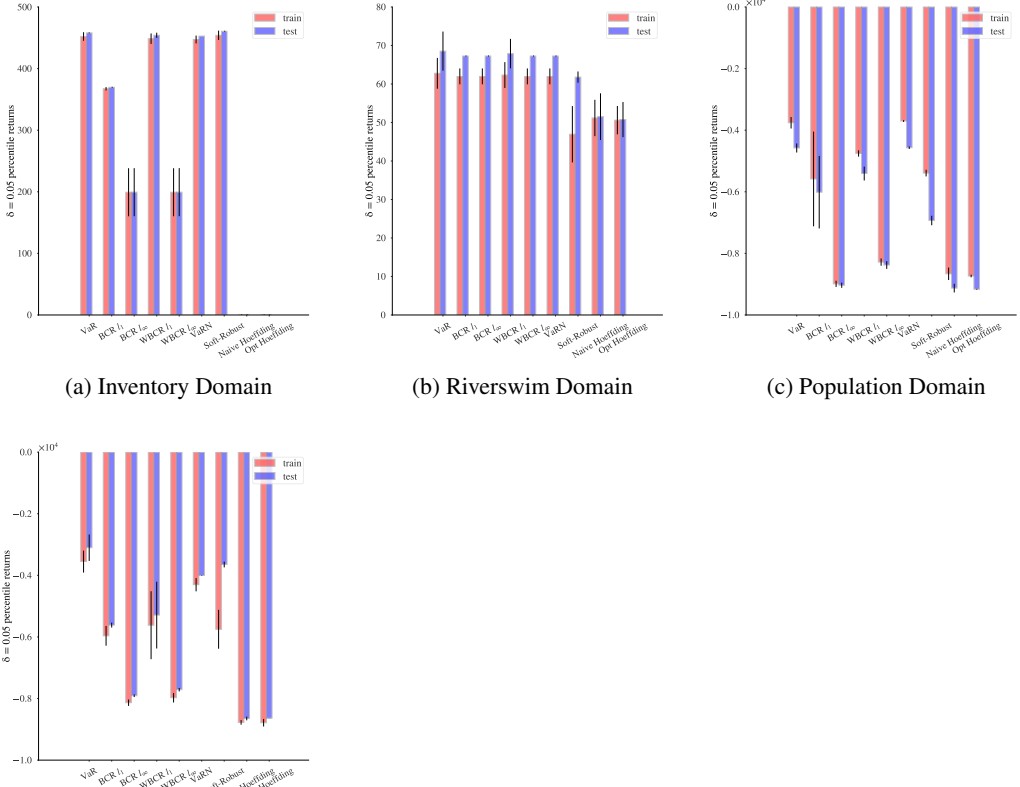

(a) Inventory Domain  (b) Riverswim Domain  (c) Population Domain

(d) Population-Small Domain

Figure 2: Comparison of test and train robust returns achieved by *VaR*, *VaRN*, *BCR* $\ell_1$, *BCR* $\ell_\infty$, *WBCR* $\ell_1$, *WBCR*, *Soft Robust*, *Naive Hoeffding* and *Opt Hoeffding* agents at confidence level $\delta = 0.05$ in Riverswim, Inventory, Population-Small and Population domain. VaR framework achieves the highest mean robust returns in most of the domains on test and train datasets.

| Hyperparameters for Population-Small Domain | |
| --- | --- |
| Number of train models per dataset (M) | 80 |
| Number of test models (K) | 100 |
| Number of train datasets (L) | 10 |

| Hyperparameters for Population Domain | |
| --- | --- |
| Number of train models per dataset (M) | 800 |
| Number of test models (K) | 1000 |
| Number of train datasets (L) | 9 |

## E.1 Code

The code and datasets are made available at https://github.com/elitalobo/VaRFramework.git.

## E.2 Machine Specifications

We concurrently ran all experiments using 120 threads on a CPU swarm cluster with 2GB memory per thread. The total computational time was ∼3 hours.

