# OpenReview forum: "Percentile Criterion Optimization in Offline Reinforcement Learning"
_NeurIPS.cc/2023/Conference — NeurIPS 2023 poster_

### Official Review · Reviewer_CS3B · 2023-07-07

**Soundness:** 3 good
**Presentation:** 3 good
**Contribution:** 3 good
**Rating:** 6
**Confidence:** 3

**Summary:**

This paper proposes a novel dynamic programming algorithm for optimizing the percentile criterion in offline Reinforcement Learning (RL) without explicitly constructing any uncertainty sets. The authors introduce the Value-at-Risk (VaR) Bellman operator for optimizing the percentile criterion, and show that it is a valid contraction mapping that optimizes a tighter lower bound on the percentile criterion compared to Robust Markov Decision Processes (RMDPs) with Bayesian credible region (BCR) ambiguity sets. The paper also provides theoretical analysis and bounds on the performance loss of the proposed framework and compares the asymptotic sizes of BCR and VaR ambiguity sets. Empirical results demonstrate the efficacy of the VaR framework in several domains.

**Strengths:**

As a non-expert in this field, I find the paper interesting and addressing an important problem in offline reinforcement learning by proposing a novel dynamic programming algorithm.
The authors provide a thorough analysis of the proposed VaR framework, including performance guarantees and comparisons with BCR ambiguity sets.
Empirical results show the effectiveness of the VaR framework in various domains, outperforming baselines in terms of robust performance.

**Weaknesses:**

The clarity of this paper could indeed be improved in several aspects. For instance, the authors use "w" to represent the index in line 89 and also use "w_{s,a}" in line 139 to represent the values, which might be confusing for readers.

Additionally, the authors attempt to answer a key question in this paper: "Are Bayesian credible regions the optimal ambiguity sets for optimizing the percentile criterion?" However, this problem is not clearly defined, and the criterion for optimality is not explicitly explained.


**Questions:**

In Table 1 of the paper, what is the criterion for highlighting a cell in each row corresponding to different domains (Riverswim, Population, Population-Small, and Inventory)? It seems that all of the item in the first row should be highlighted.

What will the number of samples of the posterior samples of $\tilde{P}$ affect the experimental results?

**Limitations:**

The paper does not discuss the application of the VaR framework to continuous state-action spaces in RL domains, limiting the understanding of its applicability to a wider range of problems. The time complexity of a single iteration in the algorithm, as mentioned, is O(SAN), which may make it challenging to apply the method to MDPs with large or continuous state and action spaces.

---

> ### Author Rebuttal · Authors · 2023-08-09
>
>
> Thank you for your review and helpful questions and suggestions.
>
> *The clarity of this paper could indeed be improved in several aspects. For instance, the authors use "w" to represent the index in line 89 and also use "w\_{s, a}" in line 139 to represent the values, which might be confusing for readers.*
>
> Thank you so much for this suggestion. We will improve this in the next version of the paper.
>
> *Additionally, the authors attempt to answer a key question in this paper: "Are Bayesian credible regions the optimal ambiguity sets for optimizing the percentile criterion?" However, this problem is not clearly defined, and the criterion for optimality is not explicitly explained.*
>
> Thank you for pointing this out. We will explicitly specify the criterion for optimality in the main paper.
> The criterion for optimality is the size of the ambiguity sets for any given confidence level $\alpha$. It is well known [1,2,3] that larger ambiguity sets result in unnecessarily conservative policies and poor robust returns.
> The goal of the paper was to show that the Bayesian credible regions can be unnecessarily large (Theorem  4.1) and are always larger than the VaR ambiguity sets for the same confidence level (Theorem 4.4). This implies that the Bayesian credible regions are not in-fact optimal for optimizing the percentile criterion and that VaR ambiguity sets are better suited for optimizing the percentile criterion.
>
> [3] A Bayesian Approach to Robust Reinforcement Learning, UAI 2019
>
> *In Table 1 of the paper, what is the criterion for highlighting a cell in each row corresponding to different domains (Riverswim, Population, Population-Small, and Inventory)? It seems that all of the items in the first row should be highlighted.*
>
> Thank you for pointing it out. We will fix it in the next version of the paper.
>
> *What will the number of samples of the posterior samples  affect the experimental results?*
>  Theorem 3.4 computes the asymptotic performance loss of the VaR framework relative to the performance of the optimal policy that optimizes the percentile criterion. As expected, the performance loss goes to zero, as the number of posterior samples goes to infinity , i.e., $N\to\infty$.
>  We will add an ablation study to the main paper that shows how the number of samples empirically affects the experimental results.
>
> *The paper does not discuss the application of the VaR framework to continuous state-action spaces in RL domains, limiting the understanding of its applicability to a wider range of problems. The time complexity of a single iteration in the algorithm, as mentioned, is O(SAN), which may make it challenging to apply the method to MDPs with large or continuous state and action spaces.*
>
> We note that this complexity is only for tabular MDPs. In the case of MDPs with continuous states, we can use the Sample Average Approximation method (SAA) [1] to obtain an empirical estimate of the $VaR$ update via sampling. In this case, the time complexity would depend on the number of samples of next-states sampled using the transition models sampled from the posterior, the number of these (posterior) transition models and the number of actions.
>  We have provided a discussion on how to scale our algorithm to MDPs with continuous state space in the global response.
>
>  [1] Sample Average Approximation,  Encyclopedia of Operations Research and Management Science 2016

---

> > ### Comment · Reviewer_CS3B · 2023-08-21
> >
> > Thank you for the rebuttal. I will maintain my positive score.

---

### Official Review · Reviewer_sypg · 2023-07-07

**Soundness:** 3 good
**Presentation:** 4 excellent
**Contribution:** 3 good
**Rating:** 5
**Confidence:** 3

**Summary:**

This paper proposes a novel offline RL algorithm, which is a dynamic programming algorithm to optimize a tight lower-bound approximation on the percentile criterion based on Value-at-Risk.
A solid theoretical analysis is made to demonstrate the proposed VaR value iteration algorithm implicitly constructs tight uncertainty sets that are smaller in size than any optimized Bayesian credible region (BCR).
Furthermore, this work provides both finite-sample and asymptotic bounds on the loss within the VaR framework.
Empirical results show that VaR outperforms BCR Robust MDPs on several tasks.

**Strengths:**

1. The proposed VaR algorithm is devised to optimize a lower bound on the percentile criterion without explicitly constructing ambiguity sets compared to BCR.
2. The paper provides comprehensive theoretical analysis on various aspects, including the Lower Bound Percentile Criterion, finite-sample and asymptotic bounds on the loss within the VaR framework, as well as the Equivalence of VaR Bellman optimality operator and a unique robust MDP with SA-rectangular ambiguity sets. These analyses offer reliable guarantees and contribute to the robustness of the proposed approach.

**Weaknesses:**

1. The VaR algorithm heavily relies on a precisely estimated transition model $\tilde{p} _{s,a}$. However, in offline RL, the availability of a limited dataset poses a constraint on the quality of this estimation. This limitation could potentially pose challenges for the VaR algorithm, particularly in complex tasks where the accurate modeling of the transition dynamics becomes more challenging.
2. The proposed VaR, as originally designed, focuses on MDPs with discrete state and action spaces.
Extending the algorithm to complex environments with continuous state and action space can be really hard.

**Questions:**

1. As stated in *Proposition 3.1*, the confidence level $\alpha$ is set as $\delta / S$. However, when the state space is large, will the calculation of $\text{VaR} _{\alpha}[\tilde{p} _{s,a} ^ \top \omega _{s,a}]$ be challenging due to the high confidence?
2. Additionally, the small value of $\alpha$ in *Proposition 3.2* results in a large value in the second term of the RHS. This raises concerns about the tightness of the lower bound and its suitability for optimizing the LHS.
3. I don't think the results of VaR and VaRN on Riverswim should be bolded in Table 1 since they are the same as others.
Furthermore, the results for Riverswim do not differentiate between the algorithms.
It would be worth considering increasing the number of states in Riverswim to enhance the difficulty level and provide a more comprehensive evaluation for the algorithms.

**Limitations:**

**Limitations** is discussed in *Section 6*.

---

> ### Author Rebuttal · Authors · 2023-08-09
>
> We thank you for your review and helpful questions and suggestions.
>
> *The VaR algorithm heavily relies on a precisely estimated transition model*
>
> **We would like to clarify that this is not true. We use Bayesian models to represent the uncertainty in the transition model estimate. We have mentioned in the preliminaries of the main paper that we derive a posterior distribution over the transition models using the offline data $\mathcal{D}$. We are proposing the percentile criterion method to achieve robustness against the uncertainty in the estimate transition model.**
>
> *The proposed VaR, as originally designed, focuses on MDPs with discrete state and action spaces. Extending the algorithm to complex environments with continuous state and action space can be really hard.*
>
> We would like to clarify that this claim is not true as well. It is possible to scale this algorithm to MDPs with continuous state space. We have addressed this concern in the global response.
>
> *As stated in Proposition 3.1, the confidence level ...is set as. However, when the state space is large, will the calculation of
>  be challenging due to the high confidence?*
>
> For large state spaces, the confidence level would have to be treated as a hyper-parameter and selected using cross-validation method based on the desired empirical robust performance on the validation dataset. Please see the global response for more details.
>
> It is also important to note that several prior works establish guarantees for tabular MDPs [1,2,3,4] as establishing these guarantees for MDPs with continuous state space is difficult. In these prior works, the confidence level depends on both, the number of states |S| as well as the number of actions |A|. Our work improves upon these works by obtaining a tighter bound on the confidence level $\alpha$ that is only dependent on |S| instead of |S| and |A| (Proposition 3.1).
>
> [1] Beyond Confidence Regions: Tight Bayesian Ambiguity Sets for Robust MDPs, Neurips 2018
> [2] Optimizing Percentile Criterion using Robust MDPs, AISTATS 2021
> [3] Robust Markov Decision Processes, Mathematics of Operations Research, 2014
> [4] Robust Dynamic Programming, Mathematics of Operations Research, 2014
>
> *Additionally, the small value of Proposition 3.2 results in a large value in the second term of the RHS. This raises concerns about the tightness of the lower bound and its suitability for optimizing the LHS.*
>
>  As mentioned above, our work improves upon the bounds established by prior work.
>  We agree that these bounds may not very tight but they are better than the state-of-the-art percentile-optimization frameworks proposed in [1,2].
>
>
> *I don't think the results of VaR and VaRN on Riverswim should be bolded in Table 1 since they are the same as others. Furthermore, the results for Riverswim do not differentiate between the algorithms. It would be worth considering increasing the number of states in Riverswim to enhance the difficulty level and provide a more comprehensive evaluation of the algorithms.*
>
> Thank you for this suggestion. We will use a larger and more complex Riverswim environment in the final version of the paper as well as fix the bolding in Table 1.

---

### Official Review · Reviewer_8VhD · 2023-07-07

**Soundness:** 2 fair
**Presentation:** 3 good
**Contribution:** 2 fair
**Rating:** 5
**Confidence:** 2

**Summary:**

This paper considers robust policy optimization problem, which optimizes the percentile criterion to get a policy. It constructs an uncertainty set that contains the true model with high probability, and optimizes for the worst model in the set. Previous works use Bayesian credible regions as uncertainty sets, but it was usually considered to be overly conservative. This paper proposes Value-at-Risk based dynamic programming approach that does not explicitly construct uncertainty sets, but learns with a modified backup operator. The paper shows that theoretically and empirically the proposed approach constructs much less conservative robust policy compared to Bayesian credible regions.

**Strengths:**

- First paper to consider VaR framework and VaR Bellman operators, which are quite novel.
- Concrete theoretical results for proposed framework and algorithm
- Explicitly comparing the size of ambiguity sets theoretically seems to be a novel approach
- The algorithm itself is quite simple and practical

**Weaknesses:**

- mainly only compares with BCR theoretically and empirically, while there exists a number of other different approaches in solving Robust MDP, as far as I know.
- I does not think showing the returns alone in the experiments are enough to prove the efficiency of proposed methods, since any RL algorithm that is independent to given $\delta$ will give the best return. In other words, we need to be sure that the algorithms satisfies the percentile criterion. In that sense, the empirical experiments do not seem complete.
- While this paper only focuses on methods to solve for percentile criterion and robust MDPs, it would be interesting to also compare against other safe RL or offline RL methods to see whether the algorithm works practically compared to those methods.

**Questions:**

- How is this approach "practically" differs from CVAR based approaches? Can you provide empirical results of CVAR algorithms as well?
- Are the experiment results statistically significant?
- Even if BCR method assumes a lot larger region resulting in overly conservative polices, isn't it OK practically if we get good trade-off between the return and the robustness? Since we will usually optimize the robustness hyper-parameter to make the algorithm work in desired way, if BCR with $\delta=0.5$ gives same policy to VaR with $\delta=0.05$, it seems just fine for me. If we give $\delta=1$, the compared algorithms will all act like standard RL algorithms, and there will exist $\delta'$ for BCR that corresponds to the performance of $VaR$ with $\delta$.

**Limitations:**

The authors addressed the limitations.

---

> ### Author Rebuttal · Authors · 2023-08-09
>
> Thank you for your review and helpful questions and suggestions.
>
> *mainly only compares with BCR theoretically and empirically, while there exists a number of other different approaches in solving Robust MDP, as far as I know.*
> We have addressed this concern in the global response to the authors.
>
> *I do not think showing the returns alone in the experiments is enough to prove the efficiency of the proposed methods since any RL algorithm that is independent of the given will give the best return. In other words, we need to be sure that the algorithms satisfy the percentile criterion. In that sense, the empirical experiments do not seem complete.*
>
> We would like to clarify that we compare the percentile returns of all the algorithms on the test dataset. The percentile returns are computed as the expected returns corresponding to the worst $\delta$-percentile model in the test set (for 1-$\delta confidence guarantees).
>
> *While this paper only focuses on methods to solve for percentile criterion and robust MDPs, it would be interesting to also compare against other safe RL or offline RL methods to see whether the algorithm works practically compared to those methods.*
>
>
> Thank you for this suggestion. Our paper focuses on developing and theoretically analyzing a percentile-criterion optimization framework that improves upon existing percentile-optimization frameworks in Robust Robust Reinforcement Learning. We are more concerned with handling model uncertainty (epistemic uncertainty) and not system uncertainty (aleatoric uncertainty) which is what safe RL methods do and therefore, we do not think that these methods can be compared with safe RL methods. Additionally, it is unclear how to modify the safe RL and general offline RL methods to work with Bayesian uncertainty.
>
> *How is this approach "practically" differs from CVAR-based approaches? Can you provide empirical results of CVAR algorithms as well?*
>
> All the existing work [1,2,3,4,5,6] based on CVAR approaches only handles aleatoric (system) uncertainty and not model uncertainty. We have provided additional experimental results with a CVaR Bellman framework. However, we note that we are not aware of any published work that proposes that analyzes the CVaR measure in the context of model uncertainty.
>
> [1] Risk-Constrained Reinforcement Learning
>
> with Percentile Risk Criteria, JMLR 2018
>
> [2] Towards Safe Reinforcement Learning via Constraining Conditional Value at Risk, ICML 2021
>
> [3]Algorithms for CVaR Optimization in MDPs, Neurips 2014
>
> [4] Risk-Averse Offline Reinforcement Learning
>
> [5] Risk-Averse Offline Reinforcement Learning, Arxiv 2021
>
> [6] Being Optimistic to Be Conservative: Quickly Learning a CVaR Policy, 2019
>
>
>
> *Are the experiment results statistically significant?*
>
>
> In the main paper, we have used a single train dataset of models for training the agent and a single test dataset of models for computing the percentile returns. The experiments in the paper were meant to be case studies and so we did not ensure that they are statistically significant. However, we have now computed statistically significant results and added them in the attached pdf below.
> We will add these results to the final version of the paper.
>
> *Even if BCR method assumes a lot larger region resulting in overly conservative policies, isn't it OK practically if we get a good trade-off between the return and the robustness?*
>
> When we say that the robust policies are overly conservative, it means that they tend to take unnecessarily sub-optimal actions in the worst-case scenario which results in the agent achieving poor poor robust performance. The goal of Robust RL frameworks is to maximize the worst-case returns and therefore, there is a need for learning policies that have good robust performance for the same confidence guarantees.
> Prior work has shown [1,2,3] that large uncertainty sets result in poor robust performance and therefore, much work [1,2] has attempted to optimize the size of these ambiguity sets using various heuristics. However, many of these works use Bayesian credible regions as ambiguity sets which can be very large as we show in our paper (Theorem 4.4). Therefore, these works still suffer from conservative policies. Our work evades this problem by using the VaR framework that constructs ambiguity sets that are tighter than any Bayesian credible regions.
>
> [1] Beyond Confidence Regions: Tight Bayesian Ambiguity Sets for Robust MDPs, Neurips 2018
>
> [2] Optimizing Percentile Criterion using Robust MDPs, AISTATS 2021
>
> [3] A Bayesian Approach to Robust Reinforcement Learning, UAI 2019
> .

---

### Official Review · Reviewer_JNMN · 2023-07-11

**Soundness:** 3 good
**Presentation:** 2 fair
**Contribution:** 3 good
**Rating:** 6
**Confidence:** 2

**Summary:**

This paper studies a type of robust offline reinforcement learning methods, percentile criterion optimization. Percentile criterion optimization optimizes the policy under the worst \alpha-percent of the models in an uncertainty set of models, based on the data. In this paper, the authors proposed a type of value iteration algorithms, replacing the standard Bellman backup values with value at risk (VaR) with respect to the posterior distribution of underlying models. The work is built on top of the previous work on Bayesian ambiguity sets and has a less conservative value estimates compared with previous work BCR.

**Strengths:**

1. This work proposed a VaR based dynamic programming method to optimize the percentile criterion, and avoid explicitly construct the uncertainty set of models. The proposed dynamic programming method is more tractable computationally compared with some methods constructing the uncertainty set explicitly.
2. The theoretical analysis in this paper shows that the proposed method can always yield a value estimates that dominate the value function produced by BCR methods, and be a lower confidence bound in the meantime.


**Weaknesses:**

1. The notations and proofs details about the results to section 4 is not very clear.
 - a) It is not clear what is the definition of $v^\pi$. Which dynamics/models the value function is with respect to.
 - b) The top paragraph in Page 22, in the proof of Proposition 4.1, is very unclear. Please cite and state which theorem it uses, and what's the condition and results, or derive the complete proof for this particular results.
 - c) What does the notation of a set minus a vector (or multiplied by a scalar) means in Eq (10)?
 - b) How does Theorem 4.4 shows "the asymptotic radius of the BCR ambiguity sets grows with the number of states" and is "larger" (claimed in the last paragraph of introduction), without showing the asymptotic shape of the BCR ambiguity sets. Theorem 4.4 only shows there exists certain elements in $\mathcal{P}^{BCR}_{s,a}$ that can be from a distance (depending on $|S|$) with $\bar{p}$.

2. The proposed methods and analysis are limited to the finite state-action space.

3. The empirical study only includes different variants of BCR as baselines.

4. This paper lacks some discussion about related work.
 - a) It lacks of discussion on more practical, robust MDP inspired methods such as [1].
 - b) The discussion of related work is limited to the robust MDP literature and especially Bayesian robust MDP. It lacks of discussion on methods based on frequentists' lower confidence bounds, which is similar to Proposition 3.2, for example pessimistic approaches with tabular settings or linear function approximation ([2] and many similar work).

[1] Model-Based Offline Reinforcement Learning with Pessimism-Modulated Dynamics Belief. NeurIPS 2022.

[2] Is Pessimism Provably Efficient for Offline RL. ICML 2021.

[3] Robust Dynamic Programming. Garud N. Iyengar

**Questions:**

1. Please address the questions about soundness of section 4 in weakness #1.
2. Why it is important to consider Percentile Criterion for offline RL, given it is often very conservative?
3. Can authors explain the relationship between the proposed methods with robust dynamic programming [3] and the optimal ambiguity set in [paper's ref 14 and 28]. My understanding is that the proposed method is an instantiation of robust dynamic programming algorithm [3] with the uncertainty set to be the optimal ambiguity set in [paper's ref 14 and 28].

**Limitations:**

The author discussed the limitations of their work.

---

> ### Author Rebuttal · Authors · 2023-08-09
>
> We thank you for your review and helpful questions and suggestions.
>
> *The notations and proof details about the results in section 4 are not very clear.*
> Thank you for pointing this out. We have added the correct notation in the attached pdf. We will add this fix to the main paper.
>
> *The top paragraph on Page 22, in the proof of Proposition 4.1, is very unclear.*
>
> Thank you for your feedback. The proof follows along lines similar to Theorem 2.1 in [1] which shows that a robust Bellman operator with S-A rectangular ambiguity sets (e.g. The VaR ambiguity sets in Equation 3) optimizes the policy to maximize the returns of the worst model in the ambiguity set (Equation 4). Due to lack of space, we cannot provide the entire proof in the rebuttal but we will add it in the next version of the paper.
>
> [1] Robust Dynamic Programming, Mathematics of Operations Research, 2014
>
> *What does the notation of a set minus a vector mean?*
>
> This notation indicates that we are simply subtracting the given vector from each vector in the set. Both, VaR and BCR ambiguity sets are ellipsoids centered at $\bar{p}_{s,a}$. We are simply zero-centering these ambiguity sets. This notation is common in Operations research papers [2].
>
> [2] Near-Optimal Bayesian Ambiguity Sets for Distributionally Robust Optimization
> We will clarify this notation in the main paper.
>
>
> *How does Theorem 4.4 show "the asymptotic radius of the BCR ambiguity... is "larger"? *
>
> The theorem shows that even we scale the $VaR$ ambiguity set by $\xi <\frac{\sqrt{\chi^2_{S,1-\alpha}}}{\Phi^{-1}(1-\alpha)}$, it will not contain any $BCR$ ambiguity set. In other words, the result is independent of the particular shape of the BCR ambiguity set. Here, the factor $\frac{\sqrt{\chi^2_{S,1-\alpha}}}{\Phi^{-1}(1-\alpha)}$ is proportional to the square root of  $1-\alpha$ percentile of a Chi-squared distribution with $|S|$ degrees of freedom. The value of $1-\alpha$ percentile of a Chi-squared distribution grows with the number of degrees of freedom and hence we state that the size of $\mathcal{P}_{s,a}^{BCR}$ will grow with $|S|$. The result uses Theorem 10 in [2], which proves it by contradiction.
>
> *The empirical study only includes different variants of BCR as baselines.*
>
> The goal of our paper is to improve upon the existing algorithms that optimize the percentile criterion. The Robust MDPs with $L_1$ and $L_{\infty}$ weighted ambiguity sets (WBCR-$l_{1}$, WBCR-$l_{\infty}$) proposed by [3] are the state-of-the-art frameworks that optimize the percentile criterion and hence, we compare against these methods. However, we have included additional experiments that compare against other baselines in the attached pdf.
>
> *This paper lacks of discussion on methods based on frequentist bounds. ...more practical, robust MDP*
> We will add a discussion about this paper in our related works section.
>
> * It lacks a discussion on methods based on frequentists' lower confidence bounds*
>
> Thank you for pointing this out. We will add a discussion on the frequentist methods and compare confidence bounds with them in the next version of our paper. We note that prior work [3,5] has empirically shown that Robust MDPs with Bayesian credible regions give better robustness guarantees against model uncertainty than Robust MDPs with frequentist ambiguity sets. Our work improves upon the confidence guarantees of [3,4] by a factor of |A|.
>
> [3] Optimizing Percentile Criterion using Robust MDPs, AISTATS 2021
> [4] Beyond Confidence Regions: Tight Bayesian Ambiguity Sets for Robust MDPs, Neurips 2018
>
> *Why it is important to consider Percentile Criterion, given it is often very conservative*
>
> The percentile criterion is well-studied in the Robust RL [3,5,6,7] literature due to its probabilistic robustness guarantees under model uncertainty. In model-based offline RL, transition probabilities must be estimated from data. Errors in estimates of transition probabilities can accumulate and results in learning policies that fail catastrophically when deployed. Hence, we need the percentile criterion to account for uncertainty in the transition models.
> The percentile criterion is less conservative than worst-case Robust MDPs [4] that are popularly adopted in Robust RL literature. This is because the latter optimize against the worst model in the uncertainty set whereas the percentile criterion allows users to determine the percentage of models against which the policy should be robust.
>
> [6] Percentile Optimization for Markov Decision Processes with Parameter Uncertainty
>
> [7] Value-at-Risk vs. Conditional Value-at-Risk in Risk Management and Optimization, Operations Research 2014
>
> *Can authors explain the methods in [3] and in [paper's ref 14 and 28]?*
>
> The robust dynamic programming work in [3] optimizes Robust MDPs with frequentist ambiguity sets and provides frequentist guarantees whereas the VaR framework internally constructs Bayesian ambiguity sets and provides Bayesian robustness guarantees against model uncertainty.
> [14] constructs Wasserstein ambiguity sets but does not optimize the percentile criterion and hence, does not provide any probabilistic guarantees on the true returns of the optimal policy like the VaR framework.
> [28] optimizes the percentile criterion using Robust MDPs with Bayesian credible regions and frequentist ambiguity sets and shows that Robust MDPs with Bayesian ambiguity regions perform better than Robust MDPs with frequentist ambiguity sets.
> Our work improves upon the confidence guarantees provided in [28] by a factor of $|A|$, (see Proposition 3.1) i.e. The confidence level $\alpha$ used in the construction of ambiguity sets for $1-\delta$ confidence on the true return is required to be $\frac{\delta}{SA}$ in [28] whereas in our work only requires $\alpha$ to be $\frac{\delta}{S}$ for the same confidence guarantees. We also show that the VaR framework will outperform Robust MDPs with any type of Bayesian credible region with high probability (Theorem 4.4).

---

> > ### Comment · Reviewer_JNMN · 2023-08-18
> > **Thank you for your response**
> >
> > Thanks to the authors for their detailed response.
> >
> > > clarity on the theory
> >
> > I am glad the hear the response from the author. I don't have any issue about it now, under the assumptions that they will fix the proofs in the future version of this paper, and clarify the notation to make it clear for the CS/ML audience.
> >
> > > How does Theorem 4.4 show "the asymptotic radius of the BCR ambiguity... is "larger"?
> >
> > I appreciate the author's explanation and please forgive my follow up question if I missed the point. I understand that Theorem shows the size of BCR ambiguity set will grow with $|S|$. My point is one set does not contain the other seems not indicate either of them is larger. For example, a ball with radius 1 does not contain a thin spike with length $|S|$, but the ball can still has a larger volume. My question might be particularly related to the words "larger" you used in the paper.
> >
> > Please kindly provide references if this type of comparison (of the radius) and claim (of the size) is common in the Bayesian uncertainty set literature, as I am not very familiar to these related work.
> >
> > > empirical study
> > > However, we have included additional experiments that compare against other baselines in the attached pdf.
> >
> > I appreciate the authors' effort on addressing this question. It enhance the evaluation of proposed method. However, offline RL is not some very new area, but has lots of algorithms (BCQ, BEAR, CQL, MOPO, MoRel, IQL, TD3+BC, COMBO, ATAC, ARMOR) and some commonly used datasets (D4RL). I would say comparing with optimzing CVaR and some variants of Hoeffding's on tabular MDP is very weak in terms of empirical evaluation. Even for confidence bounds based methods (like Hoeffding's), there is approach from 2015 that should be better than naive Hoeffding's.
> >
> > [1] P. S. Thomas, G. Theocharous, and M. Ghavamzadeh. High Confidence Policy Improvement. In Proceedings of the Thirty-Second International Conference on Machine Learning (ICML), 2015.

---

> > > ### Author Response · Authors · 2023-08-20
> > > **Thank you so much for your response to our rebuttal!**
> > >
> > > We thank the reviewer for reading our rebuttal and responding to it. We really appreciate the time and effort the reviewer put into this.
> > >
> > > **Regarding the comparison of radii of BCR and VaR ambiguity sets**
> > >
> > >
> > > Firstly, we apologize for any confusion that we may have caused in the rebuttal.
> > >
> > > For references of asymptotic radii comparison, please see [2], [3]. [2] compares the size of Bayesian credible regions and Frequentist confidence regions based ambiguity sets with VaR ambiguity sets by comparing their asymptotic radii.
> > >
> > > The authors of [2] mention in the paper (Section 4.3 and 5.1) that asymptotically, the Bayesian credible regions (BCR) and frequentist confidence regions are atleast $\Omega(\sqrt{r})$ larger than VaR ambiguity sets where $r$ is the number of uncertain parameters (Theorem 10 and Theorem 14). Note that we extend these results to RL setting and in our case, $r=|\mathcal{S}|$.
> > >  However,
> > > when they state that Bayesian credible regions (BCR) and frequentist confidence regions are $\Omega(\sqrt{r})$ times larger than VaR ambiguity sets, they specify that this means that there exist directions in which these sets are at least $\Omega(\sqrt{r})$ times larger than the asymptotic VaR ambiguity sets (e.g., Line 6 on page 6, last line of section 4.3). They are not comparing volumes while comparing the size of ambiguity sets but their asymptotic radii.  We use and extend precisely these results in our VaR framework. We will add this clarification about the size/radii comparison in the camera-ready version of the paper.
> > > We thank the reviewer for pointing out this issue.
> > >
> > > [2] Near-Optimal Bayesian Ambiguity Sets for Distributionally Robust Optimization, https://faculty.marshall.usc.edu/Vishal-Gupta/Papers/Bayes_DRO_v5_final.pdf
> > >
> > > [3] Data-driven ambiguity sets with probabilistic
> > > guarantees for dynamic processes, https://arxiv.org/pdf/1909.11194.pdf
> > >
> > >
> > > Please see the following response for other clarifications.

---

> > > ### Author Response · Authors · 2023-08-20
> > > ****Regarding comparisons with other Offline RL works****
> > >
> > > We thank the reviewer for their comments.
> > >
> > > #### **Regarding lack of baselines after 2016 work**
> > > We actually have two baselines [4,5] based on confidence bounds-based methods that were published after 2016.
> > >
> > > [4] Optimizing Percentile Criterion Using Robust MDPs, AISTATS 2021
> > >
> > > [5] Beyond Confidence Regions: Tight Bayesian Ambiguity Sets for Robust MDPs, Neurips 2018
> > >
> > > #### **Comparison with High Confidence Policy Improvement work**
> > > The work in High Confidence Policy Improvement paper [1] provides probabilistic guarantees against system/aleatoric uncertainty (randomness in the **returns** due to the stochasticity of the transition model).
> > > Please note that we provide probabilistic guarantees against epistemic uncertainty (randomness in the **expected returns** due to uncertainty in the parameters of the transition model). These guarantees are different; therefore, this work [1] is not comparable to our method.
> > >
> > > #### **Regarding comparisons with other Offline RL works**
> > > We would like to clarify that our work aims to improve the existing percentile optimization algorithms that handle epistemic uncertainty in model-based RL. Hence, we specifically compare with prior work that optimizes the percentile criterion objective. Therefore, none of these model-based and model-free work (BCQ, BEAR, CQL, MOPO, MoRel, IQL, TD3+BC, COMBO, ATAC, ARMOR)  are directly comparable with our work.
> > >
> > > We are happy to add experiments that compare with percentile optimization algorithms in RL that handle epistemic uncertainty, but we are unaware of any work other than our current baselines.
> > >
> > > We will add experiments comparing against some of the above listed model-based RL methods if it is absolutely necessary.
> > > However, we note that the main contributions of our work are theoretical, and extensive empirical studies are out of the scope of this paper.
> > >
> > > #### **Our main contributions are detailed below:**
> > >
> > > 1. A theoretically grounded VaR Bellman operator that optimizes a lower bound on the VaR of the expected returns (Percentile criterion) objective in Reinforcement Learning. This is a non-trivial extension of the VaR optimization in a supervised learning setting and it also does not make any assumptions on the priors of the posterior distributions of the transition model. (Proposition 3.1)
> > >
> > > 2. Proofs showing that the VaR Bellman operator is a contraction mapping and that it optimizes a tighter lower bound on the percentile criterion. (Proposition A.4)
> > >
> > > 3. Derivation of a simplification of the VaR Bellman operator for Sub-Gaussian and Gaussian case. (Proposition 3.2, Proposition A.5)
> > >
> > > 4. An in-depth theoretical comparison between VaR framework and the Bayesian Robust MDPs that is currently the SOTA optimization method for percentile criterion in RL (Proposition 4.1, Proposition 4.2, Theorem 4.3, Theorem 4.4, Lemma A.7, Lemma A.8).
> > >
> > > 5. Finite and asymptotic error bounds on the performance of the VaR framework (Theorem 3.3, Theorem 3.4).
> > >
> > > 6. Error bounds on the VaR value iteration algorithm (Theorem 3.6, Proposition A.6)
> > >
> > > 7. The VaR Value iteration algorithm (Algorithm 3.1) and 4 empirical case studies where we compare the VaR framework with **5** baseline methods that either optimize the percentile criterion or provide robustness against model (epistemic ) uncertainty (Attached pdf, Section 5 and Appendix Section C).

---

### Official Review · Reviewer_CU8M · 2023-07-26

**Soundness:** 3 good
**Presentation:** 3 good
**Contribution:** 2 fair
**Rating:** 5
**Confidence:** 4

**Summary:**

This paper proposes using percentile criterion optimization for offline and robust reinforcement learning. Instead of using the Bellman optimality operator for value iteration, this paper suggests an algorithm using the Value-at-Risk (VaR) operator to account for pessimism. The VaR operator takes the given percentile of the return among ambiguity sets rather than the maximum. The authors show theoretically and empirically that such an operator leads to tighter value estimation with a given confidence level compared to prior works using Bayesian credible regions. Experiments on several tabular MDPs show that the VaR framework performs better than prior works using Bayesian credible regions.


**Strengths:**

1. The paper investigates an important problem of robust RL: Bayesian credible regions can be unnecessarily large, leading to overly conservative policies. The proposed VaR operator provably improves prior methods using the Bayesian credible regions.

2. The paper provides a statistical complexity analysis and a computation complexity bound for the proposed empirical VaR operator.

3. The paper is generally well-written and easy to read.

**Weaknesses:**

1. The proposed method degrades to frequentist pessimism with confidence sets (e.g., [1,2]) for Gaussian cases, which means the method is essentially the same as prior frequentist methods up to constant factors when the posterior is normally distributed, or the approximation in line 8 of Algorithm 3.1 is used (i.e., *VaRN* in the experiments). From the experiments, it seems that *VaRN* performs similarly to *VaR*. It is unclear how much improvement the proposed algorithm has over prior frequentist methods, especially when the posterior is not normally distributed.

2. The experiments are conducted on small tabular MDPs, and it is unclear how to scale up the proposed method to real-world settings. While the focus of the paper is on the theory side, real-world scale experiments would greatly strengthen the paper.


[1] Jin, Chi, et al. "Provably efficient reinforcement learning with linear function approximation." Conference on Learning Theory. PMLR, 2020.

[2] Jin, Ying, Zhuoran Yang, and Zhaoran Wang. "Is pessimism provably efficient for offline rl?." International Conference on Machine Learning. PMLR, 2021.

**Questions:**

1. How does the proposed method compare to frequentist methods, especially when the posterior is not normally distributed?

2. How can we scale up the proposed method to large-scale RL problems?

**Limitations:**

Limitations are well discussed in the paper.

---

> ### Author Rebuttal · Authors · 2023-08-09
>
> We thank you for your review and helpful questions and suggestions.
>
> *The proposed method degrades to frequentist pessimism with confidence sets*
> It is not true that the method degrades to frequentist pessimism. The method remains Bayesian, no matter what is the number of samples. We agree that the solution techniques are similar to methods that aim to provide frequentist guarantees, but the pessimistic coefficients involved in the algorithms are computed very differently in the Bayesian setting from the frequentist setting [3]. As a result, the guarantees in our objective do not translate to similar guarantees in frequentist settings (and vice versa).
>
> Also note that the normal posterior result (Bernstein-Von Mises) represents a frequentist analysis of a Bayesian setting. Even in this case, the guarantees that we compute are Bayesian, regardless of the number of samples.
>
>
> *The proposed method degrades to frequentist pessimism ... as prior frequentist methods up to constant factors when the posterior is normally distributed, or the approximation in line 8 of Algorithm 3.1 is used (i.e., VaRN in the experiments).*
>
> We note that the Gaussian VaR Bellman operator (Proposition A.2) and the Sub-Gaussian VaR Bellman operator (Proposition 3.2) are only special cases of our VaR framework. The main contributions of this paper are  a)
> a generalized VaR Bellman operator (Equation 5) for optimizing the percentile criterion and b) An in-depth theoretical analysis of the properties of this framework showing that the VaR Bellman operator optimizes a tighter lower bound on the percentile criterion than the state-of-the-art robust MDPs with BCR ambiguity sets. The generalized VaR Bellman operator does not make any assumptions on the distribution of the transition models.
>
> Furthermore, the methods proposed in [1,2] are not comparable with our framework for two reasons. a) They do not specifically account for Bayesian uncertainty in the transition models and hence do not provide any robustness guarantees against the worst $\alpha$ percentile model, which is our main objective. b) It is not clear how to adapt these algorithms to handle Bayesian uncertainty in the models.
> Finally, [1,2] do not theoretically or empirically compare their results with any Bayesian Robust RL methods.
>
> *From the experiments, it seems that VaRN performs similarly to VaR. It is unclear how much improvement the proposed algorithm has over prior frequentist methods, especially when the posterior is not normally distributed.*
>
> Prior work [3,4] has compared frequentist and Bayesian approaches in the context of robust MDPs. This work already shows that Robust MDPs with Bayesian credible regions outperform Robust MDPs with frequentist ambiguity sets (See figure 4 in [3], Table 1 and 2 in [4]) for any given confidence level.
> Our work improves over the state-of-the-art Robust MDPs with Bayesian credible regions and therefore are guaranteed to perform better than Robust MDPs with frequentist ambiguity sets. Per the reviewer's request, we have added additional experiments that compare with the above-mentioned frequentist methods.
>
> *The experiments are conducted on small tabular MDPs, and it is unclear how to scale up the proposed method to real-world settings. While the focus of the paper is on the theory side, real-world-scale experiments would greatly strengthen the paper.*
>
> Thank you for this suggestion. As you rightly pointed out, the focus of this paper is to theoretically show the limitations of existing state-of-the-art algorithms that optimize the percentile criterion framework and propose a new framework with better theoretical guarantees.
>  However, the VaR algorithm can be scaled to MDPs with continuous state space. Please check the global response for more details.
>
> [1] Jin, Chi, et al. "Provably efficient reinforcement learning with linear function approximation." Conference on Learning Theory. PMLR, 2020.
>
> [2] Jin, Ying, Zhuoran Yang, and Zhaoran Wang. "Is pessimism provably efficient for offline rl?." International Conference on Machine Learning. PMLR, 2021.
>
> [3] Beyond Confidence Regions: Tight Bayesian Ambiguity Sets for Robust MDPs, Neurips 2018
>
> [4] Optimizing Percentile Criterion using Robust MDPs, AISTATS 2021

---

> > ### Comment · Reviewer_CU8M · 2023-08-18
> >
> > Thanks for the reply. However, many of my concerns remain unsolved.
> >
> > *The method remains Bayesian, no matter what the number of samples.*
> >
> > To clarify, I mean the method degrades to the frequentist one when using **Gaussian priors** rather than changing the number of samples. Such equivalence appears in many places (e.g., Bayesian linear regression [5]).
> >
> > *Prior work has compared frequentist and Bayesian approaches in the context of robust MDPs.*
> >
> > Prior works compare their methods only on small MDPs (e.g., RiverSwim has  6 states and 2 actions) or with special priors (e.g., Dirichlet priors). The proposed VaR method degrades to the frequentist one with Gaussian priors, which forms a loop (i.e., freq method $\approx$ proposed VaR method > prior work > freq method) and makes me doubt if the Bayesian method outperforms the frequentist one universally. I believe a thorough comparison with frequentist methods is needed.
> >
> > *The methods proposed in [1,2] are not comparable with our framework ... They do not provide any robustness guarantees against the worst $\alpha$ percentile model, which is our main objective*
> >
> > Prior works like [1,2] have high probability guarantees, similar to the $\alpha$-percentile guarantee, and there are also many prior works [6,7,8] using the VaR objective explicitly from a frequentist point of view.
> >
> > *Per the reviewer's request, we have added additional experiments that compare with the above-mentioned frequentist methods.*
> >
> > I appreciate the authors' effort in conducting more experiments to compare with frequentist methods. However, the new experiment seems insufficient. Especially, there is only one task for the frequentist method, and they achieve exactly zero performance, and "the number of samples for some state-action pairs is 0" should not be a barrier since we can just use $\max(1,n)$.
> >
> > To summarize, while I appreciate the authors' effort to theoretically analyze the Bayesian VaR framework, the following concerns remain:
> >
> > 1. The proposed method is similar to the frequentist one (especially for 1. Gaussian priors or sub-Gaussian noises and 2. practical implementations like *VaRN*), and analyzing VaR from a frequentist point of view appears in many prior works [6,7,8].
> >
> > 2. The experiments seems insufficient. A thorough comparison with frequentist methods would greatly strengthen the paper.
> >
> > [5] https://www.cs.toronto.edu/~rgrosse/courses/csc411_f18/slides/lec19-slides.pdf
> >
> > [6] Chow, Yinlam, et al. "Risk-sensitive and robust decision-making: a cvar optimization approach." Advances in neural information processing systems 28 (2015).
> >
> > [7] Nguyen, Quoc Phong, et al. "Value-at-risk optimization with Gaussian processes." International Conference on Machine Learning. PMLR, 2021.
> >
> > [8] Wang, Kaiwen, Nathan Kallus, and Wen Sun. "Near-Minimax-Optimal Risk-Sensitive Reinforcement Learning with CVaR." arXiv preprint arXiv:2302.03201 (2023).

---

> > > ### Author Response · Authors · 2023-08-20
> > > **Thank you for the response to our rebuttal**
> > >
> > > We thank the reviewer for their suggestions and response to our rebuttal.
> > >
> > > Per their request, we will add more experiments that compare with the frequentist methods in our camera-ready version.
> > >
> > >  ### **Regarding our method degrading to frequentist one when using Gaussian priors**:
> > > Consider the VaR of any univariate random variable $X$ that is Gaussian distributed with mean $\mu $ and variance $\sigma $ , i.e., $\mathcal{N}(\mu, \sigma)$ . This VaR value can be expressed as $(\mu - \Phi^{-1}(\alpha) \sigma)$ for any confidence level $\alpha$. Note that the estimated parameter values $(\mu$ and $\sigma$) of the Gaussian distribution will change depending on whether it is a Bayesian or frequentist setting when the number of samples is finite. Please see the posterior hyper-parameters for Gaussian/Normal prior in [1]. These values are not the same as the Maximum Likelihood Estimates (MLEs) of these parameters computed by frequentist methods [3], as they depend on the parameters of the Gaussian priors.
> > > This means that the parameters of the Gaussian posterior distribution over transition probabilities **do not coincide** with the Frequentist parameter estimates of the Gaussian distribution over transition probabilities when the **number of samples is finite**. Hence, we believe that the reviewer's claim that 'the method degrades to the frequentist one when using Gaussian priors rather than changing the number of samples' is not true in a finite-sample setting.
> > >
> > > Additionally, we would like to point out that VaRN is only analyzed as a special case of our framework. Algorithm 3.1 contains the update rule for generalized VaR as well. We have also analyzed the generalized VaR (VaR) in our experiments.
> > >
> > >
> > > ### **Regarding comparison with other frequentist methods**
> > >  We would also like to clarify that **the results of Theorem 4.3** and **Theorem 4.4 that establish the sub-optimality of BCR ambiguity sets are also applicable to frequentist confidence regions based ambiguity sets.** i.e., the lower-bound on the ratio of the asymptotic radii of BCR to VaR ambiguity sets is the same as the lower-bound on the ratio of the asymptotic radii of frequentist confidence regions-based ambiguity sets to VaR ambiguity sets. Hence, even the frequentist confidence regions based ambiguity sets are sub-optimal for optimizing the percentile criterion.
> > > This simply follows from the Bernstein von Mises theorem [2] that states that the posterior distribution centered at the MLE is asymptotically equivalent to the sampling distribution of the MLE.
> > > (i.e., As $N\to\infty$, the posterior distribution of the transition probabilities becomes a Gaussian distribution centered at the MLE of the transition probabilities and is independent of the priors).
> > > **This proves that our VaR framework is theoretically guaranteed to be better than any Robust MDPs with Bayesian credible regions and Frequentist confidence regions-based ambiguity sets.**
> > > We will clarify this result and add it as a theorem in the main paper. We are also happy to add more empirical evidence of this theoretical result (other than the results provided in the global response) if required. Please seed the global response for updated experiments.
> > >
> > > ### **Regarding comparison with papers that provide probabilistic guarantees and similarity between our work and [6,8]**
> > > Not all papers that provide probabilistic guarantees are comparable with our work. It is essential to understand the random variables of interest for which these probabilistic guarantees are established and against what kind of uncertainty are these works providing robustness. Please note that we provide probabilistic guarantees against epistemic uncertainty (randomness in the **expected returns** due to uncertainty in the parameters of the transition model), and these guarantees are established on the expected returns of the optimal policy. [6,8] provide probabilistic guarantees against system/aleatoric uncertainty (randomness in the **returns** due to the stochastic transition model). These guarantees are not the same, and therefore, these frameworks are not comparable.
> > >
> > > **References**
> > >
> > > [1] https://en.wikipedia.org/wiki/Conjugate\_prior
> > >
> > > [2] https://en.wikipedia.org/wiki/Bernstein-von\_Mises\_theorem
> > >
> > > [3] https://www.cs.ubc.ca/~murphyk/Teaching/CS340-Fall07/reading/paramEst.pdf
> > >
> > > Please see the following response for other clarifications.

---

> > > > ### Author Response · Authors · 2023-08-20
> > > > **Regarding similarity between our work and 'Value-at-risk optimization with Gaussian processes.' paper, and our main contributions**
> > > >
> > > > ### **Regarding similarity with Value-at-risk optimization with Gaussian processes**
> > > > The VaR robust measure/objective is popular in the RL, ML and Operations research literature [4,5,6,7,8,9,10] due to its probabilistic robustness guarantees. However, the context in which it is used drastically changes the difficulty of optimizing it. The authors of [7] describe how to optimize the VaR of a black box objective function under Gaussian priors in a **supervised learning** setting which is trivial compared to optimizing this objective for a **multi-step sequential decision problem** that involves value functions. When we optimize the VaR of expected returns with the transition model as the random variable of interest, we get the percentile criterion (Equation 2 in main paper). Prior works [11,12,8] have shown that since VaR is **non-convex**, it is **NP-Hard** to optimize the VaR of expected discounted returns (percentile criterion), and therefore, these work
> > > > approximately optimizes it using Robust MDPs with Frequentist or Bayesian ambiguity sets.
> > > > Our work also optimizes a (tighter) lower bound on the VaR of expected returns. We achieve this by optimizing the policy to maximize the VaR of the value function at each step via the VaR Bellman operator (**Proposition 3.1**). We also theoretically prove that a) the  VaR Bellman operator is a valid contraction mapping (**Proposition A.4**), and b) it has a unique fixed point which is the value of the optimal VaR policy that maximizes a lower bound on VaR of expected returns (**Proposition A.4**).
> > > > Note that the above extension of the VaR framework to RL setting for handling epistemic uncertainty  is non-trivial.
> > > >
> > > > Finally, we would like to reiterate that the main objective of our work is to understand the shortcomings of existing percentile criterion optimization algorithms that handle model uncertainty in RL and improve upon these algorithms.
> > > >
> > > > #### **Our main contributions are detailed below:**
> > > >
> > > > 1. A theoretically grounded VaR Bellman operator that optimizes a lower bound on the VaR of the expected returns (Percentile criterion) objective in Reinforcement Learning. This is a non-trivial extension of the VaR optimization in a supervised learning setting and it also does not make any assumptions on the priors of the posterior distributions of the transition model. (Proposition 3.1)
> > > >
> > > > 2. Proofs showing that the VaR Bellman operator is a contraction mapping and that it optimizes a tighter lower bound on the percentile criterion. (Proposition A.4)
> > > >
> > > > 3. Derivation of a simplification of the VaR Bellman operator for Sub-Gaussian and Gaussian case. (Proposition 3.2, Proposition A.5)
> > > >
> > > > 4. An in-depth theoretical comparison between VaR framework and the Bayesian Robust MDPs that is currently the SOTA optimization method for percentile criterion in RL (Proposition 4.1, Proposition 4.2, Theorem 4.3, Theorem 4.4, Lemma A.7, Lemma A.8).
> > > >
> > > > 5. Finite and asymptotic error bounds on the performance of the VaR framework (Theorem 3.3, Theorem 3.4).
> > > >
> > > > 6. Error bounds on the VaR value iteration algorithm (Theorem 3.6, Proposition A.6)
> > > >
> > > > 7. The VaR Value iteration algorithm (Algorithm 3.1) and 4 empirical case studies where we compare the VaR framework with **5** baseline methods that either optimize the percentile criterion or provide similar robustness against model (epistemic ) uncertainty (Section 5 and Appendix Section C).
> > > >
> > > > **References**
> > > >
> > > > [4]Percentile Optimization for Markov Decision Processes with Parameter Uncertainty, Operations Research, 2008
> > > > [5] Value-at-Risk vs. Conditional Value-at-Risk in Risk Management and Optimization, Operations Research, 2014
> > > >
> > > > [6] Value-at-Risk Based Risk Management: Optimal Policies and Asset Prices, The Review of Financial Studies, 2008
> > > >
> > > > [7] Computing near-optimal Value-at-Risk portfolios using Integer Programming techniques, European Journal of Operations Research, 2021
> > > >
> > > > [8] Encoded Value-at-Risk: A machine learning approach for portfolio risk measurement, 2022
> > > >
> > > > [9] DeepVaR: a framework for portfolio risk assessment leveraging probabilistic deep neural networks, Arxiv, 2023
> > > >
> > > > [10] Value-at-risk optimization with Gaussian processes." International Conference on Machine Learning. PMLR, 2021.
> > > >
> > > > [11] Optimizing Percentile Criterion Using Robust MDPs, AISTATS, 2021
> > > >
> > > > [12] Beyond Confidence Regions: Tight Bayesian Ambiguity Sets for Robust MDPs, Neurips, 2018

---

> > > > > ### Comment · Reviewer_CU8M · 2023-08-21
> > > > >
> > > > > Thanks for the detailed reply. However, my concern remains. In Gaussian cases, while they are not identical, they are the same up to constants (i.e., different $\alpha$ in $\mu-\alpha \sigma$). This makes the improvement incremental and the practical application limited.
> > > > > It is nice to be able to handle epistemic uncertainty. However, it is a natural result of employing a Bayesian approach, and such an advantage is not demonstrated empirically like prior works[1]. I believe a thorough comparison and discussion with prior works on non-Bayesian VaR methods would greatly strengthen the paper.
> > > > >
> > > > >
> > > > > [1] Chua, Kurtland, et al. "Deep reinforcement learning in a handful of trials using probabilistic dynamics models." Advances in neural information processing systems 31 (2018).

---

> > > > > > ### Author Response · Authors · 2023-08-21
> > > > > > **Regarding Gaussian case and Empirical Analysis**
> > > > > >
> > > > > > Thank you for your response!
> > > > > >
> > > > > > *In Gaussian cases, while they are not identical, they are the same up to constants.*
> > > > > >
> > > > > > Can you please provide references to papers that prove this claim?
> > > > > > It is also essential to consider that in RL, we recursively apply the VaR Bellman operator, and so, even in the Gaussian case, these small changes in the VaR values can accumulate and result in a significantly different RL policy.
> > > > > > To the best of our knowledge, no papers show that the optimal VaR RL policy is the same in frequentist and Bayesian settings.
> > > > > >
> > > > > > *I believe a thorough comparison and discussion with prior works on non-Bayesian VaR methods would greatly strengthen the paper.*
> > > > > >
> > > > > > We are not aware of any other RL papers that optimize the frequentist VaR objective to provide guarantees against the epistemic uncertainty in RL other than the baselines we have used or the works we have cited. If you can kindly point them out, we will add them to the camera-ready version of the paper. Furthermore, we have provided (above) the sketch of the proof that our method is theoretically guaranteed to be better than Robust MDPs with frequentist ambiguity sets. We have also added experiments to the global response that demonstrate these theoretical results in Population-Small, Population, and Inventory domains.
> > > > > >
> > > > > > *Chua, Kurtland, et al. "Deep reinforcement learning in a handful of trials using probabilistic dynamics models." Advances in neural information processing systems 31 (2018)* -> Unfortunately, this work assumes access to the true transition model for sampling a few trajectories at a time to improve the learned probabilistic model and the optimal policy. Furthermore, this work mentions in the last paragraph of the introduction that "Isolating epistemic uncertainty is especially useful for directing exploration [Thrun, 1992], although we leave this for future work." and does not provide any theoretical guarantees against epistemic uncertainty or comparison with Robust RL methods. Hence, this work as well is not comparable to ours.
> > > > > >
> > > > > >
> > > > > > *In Gaussian cases, while they are not identical, they are the same up to constants. This makes the improvement incremental and the practical application limited.*
> > > > > >
> > > > > > This comment also suggests that **any** prior work that uses Bayesian methods are incremental and have limited practical application since they ***may*** be identical to frequentist methods (up to a constant) when Gaussian priors are used. We disagree with this comment.
> > > > > >
> > > > > >
> > > > > > Finally, we believe that it is unfair to ignore all our contributions because a special case (Gaussian case) of our framework ***may*** be identical to frequentist methods. The special case analysis of our framework is a very small part of our main contributions.
> > > > > > We also argue that our theoretical contributions are as non-trivial as the extensive empirical analysis of empirical RL papers.

---

### Author Rebuttal · Authors · 2023-08-09

**Scalability of VaR Framework**

The main goal of this paper is to theoretically show the limitations of existing state-of-the-art algorithms that optimize the percentile criterion framework and propose a new framework with better theoretical guarantees and therefore, we focus less on the empirical analysis.
***However, it is possible to computationally scale our framework to MDPs with continuous state spaces.***

Recall that the VaR Bellman operator optimizes a lower bound on the percentile criterion. Thus, we can use the VaR Bellman operator with any deep Q-learning or Actor-Critic methods and minimize the VaR Bellman residual error ($J_{VaR}(q^{\pi}_{\theta})$ defined in attached pdf) to obtain the optimal VaR policy that optimizes a lower bound on the percentile-criterion.

In this case, we would treat the confidence level $\alpha$ as the hyperparameter value and choose $\alpha$ using the cross-validation method based on the desired empirical robustness performance on the validation model dataset. Note that this approach is not uncommon. Several scalable prior work on Robust MDPs that use KL Divergence based ambiguity sets, Relative Entropy based ambiguity sets, Wasserstein distance based ambiguity sets or  $\Phi$-Divergence based ambiguity sets [1,2,10,11] treat the size of these ambiguity sets as a hyperparameter.

We can also directly use scalable algorithms for Robust MDPs when we assume that $\tilde{P}$ is Gaussian distributed. Recall that when $\tilde{P}$ is Gaussian distributed, the VaR Bellman operator optimizes a Robust MDP with a special ambiguity set provided in Eq 9 in the main paper.

Thus, we can use scalable algorithms in [8,9] to directly optimize this Robust MDP.

We note that our current theoretical probabilistic guarantees only apply to tabular settings. However, it is important to realize that several prior [1,2,3,4,5,6,7] work on Robust MDPs and percentile optimization only provide theoretical guarantees for tabular settings. This does not indicate that it is not possible to get any guarantees for continuous state space but that it is difficult and unclear how to derive them.

[1] Robust Markov Decision Processes, Mathematics of Operations Research, 2014

[2] Robust Dynamic Programming, Mathematics of Operations Research, 2014

[3] Distributionally Robust Markov Decision Processes, Neurips 2010

[4] Robust Markov Decision Processes: Beyond Rectangularity, 2018

[5] Reinforcement Learning in Robust Markov Decision Processes, Neurips 2013

[6] Beyond Confidence Regions: Tight Bayesian Ambiguity Sets for Robust MDPs

[7] Optimizing Percentile Criterion using Robust MDPs, AISTATS 2021

[8] Scaling Up Robust MDPs by Reinforcement Learning, JMLR 2014

[9] Robust Reinforcement Learning using Least Squares Policy Iteration with Provable Performance Guarantees, ICML 2021

[10] Wasserstein Robust Reinforcement Learning, 2019
[11] Robust ϕ-Divergence MDPs

**Additional Experiments**

We have compared our VaR framework against the following baselines in the additional experiments provided in the pdf.

 [1*] WBCR $l_1$, WBCR $l_{\infty}$ -> Optimizing Percentile Criterion using Robust MDPs, AISTATS 2021

[2*] BCR $l_1$, BCR $l_{\infty}$, Optimized Hoeffding -> Beyond Confidence Regions: Tight Bayesian Ambiguity Sets for Robust MDPs, Neurips 2018, Deviation of the Empirical Distribution, Weissman et al, 2003

[3*] Worst RMDP -> Robust Reinforcement Learning under model misspecification, ICLR 2020

[4*] Soft-robust -> Soft-Robust Actor-Critic Policy-Gradient, UAI 2018

[5*] Naive Hoeffding -> Safe Policy Improvement by Minimizing Robust Baseline Regret, Petrik et al 2016, Inequalities for the L1


Experiment details: We use the Bootstrap sampling approach to compute the mean and standard deviation statistic on the robust performance ($\delta$ percentile of expected returns) of the robust RL methods.  We generate 10 datasets of train model by randomly sampling 80% models from the original train dataset. We use each train dataset to train one RL agent per method. We then compute the mean and standard deviation of the robust performance ($\delta$ percentile of expected returns) of RL agents of each method on the test dataset.

Table 1 and Table 2 show the mean and standard deviation of the robust (percentile) returns at confidence level $\delta=0.05$ and $\delta=0.30$ achieved by *{VaR}*, *VaRN*, *BCR* $\ell_1$, *BCR* $\ell_\infty$, *WBCR* $\ell_1$ and *WBCR* $\ell_\infty$, *CVaR*, *Soft-Robust*, *Worst (-case) RMDP*, *Naive Hoeffding* and *Optimized Hoeffding* in Riverswim, Inventory , Population Small and Population domains.

We note that frequentist methods , i.e., Naive Hoeffding and Optimized Hoeffding construct frequentist ambiguity sets (Theorem 4.1 and Appendix C.1 in [2*]) whose size depends on the number of samples observed per state-action pair in the dataset . In Riverswim, Population small and Population dataset, the number of samples for some state-action pairs is 0 and hence, we couldn't evaluate frequentist methods on these datasets.


Our experiments confirm that VaR Framework and Bayesian Robust MDPs achieve superior robust performance compared to frequentist methods (Naive Hoeffding and Optimized Hoeffding).

We also note that in many cases, we observe that VaR has similar performance to Soft-robust and Worst-case RMDP. We conjecture that this is due to small variance in the transition models as a result of which the expected returns distribution does not have a fat tail. Furthermore, we also believe that the environments may be too simple for evaluating the true difference in performance of these robust measures. However, note that prior work have used these environments for robustness evaluation [6,7].   We will work towards adding more complex environments in the camera ready version of the paper.

---

> ### Author Response · Authors · 2023-08-21
> **Additional experimental results showing comparison with Robust MDPs with frequentist ambiguity sets**
>
> *I appreciate the authors' effort in conducting more experiments to compare with frequentist methods. However, the new experiment seems insufficient. Especially, there is only one task for the frequentist method, and they achieve exactly zero performance, and "the number of samples for some state-action pairs is 0" should not be a barrier since we can just use max(1,n)*
>
> As per the above request of the CU8M, we provide additional preliminary results for Population-Small, Inventory and Population domain by setting max #nsa (number of samples for some state-action pairs) =100 in population-small and population domains (2 seeds).
>
> $\delta=0.3$
>
> |  Method |  Inventory | Population-Small  |Population   |
> |---|---|---|---|
> VaR  | 483.08 $\pm$ 0.2 | -1117.57 $\pm$ 120.01 | -1856.06 $\pm$ 83.74 |
> VaRN  | 482.92 $\pm$ 0.59 | -1514.44 $\pm$ 12.31 | -1806.12 $\pm$ 1.77 |
> Naive Hoeffding [5]  | -3.10237e-14 $\pm$ 1.2621774483e-29 | -4485.15 $\pm$ 63.58 | -4985.78 $\pm$ 0.0|
> Optimized Hoeffding [2]  | -3.1023e-14 $\pm$ 1.26217e-29 | -1142.17 $\pm$ 83.84 | -2032.66 $\pm$ 0.0 |
>
>
> $\delta=0.05$
>
> |  Method |  Inventory | Population-Small  |Population   |
> |---|---|---|---|
> VaR | 457.95 $\pm$ 0.37 | -3102.48 $\pm$ 214.85 | -4578.84 $\pm$ 69.76 |
> VaRN  | 452.78 $\pm$ 0.01 | -4005.53 $\pm$ 4.38 | -4576.65 $\pm$ 0.0 |
> Naive Hoeffding [5]  | -6.53070e-14 $\pm$ 1.262177448e-29 | -7146.76 $\pm$ 0.0 | -7651.01 $\pm$ 0.0 |
> Optimized Hoeffding [2]  | -6.53070e-14 $\pm$ 1.26217744e-29 | -4355.4 $\pm$ 42.36 | -5998.41 $\pm$ 0.0 |
>
> We will add this result and additional ablation study with datasets containing different number of samples in the camera-ready version of the paper paper.

---

### Author Response · Authors · 2023-08-21
**Regarding the lack of extensive experiments**

The main contributions of our work are primarily theoretical, and extensive empirical studies are out of the scope of this paper.
However, as per the request of the reviewers, we will add a larger Riverswim domain and, if required, more experiments showing comparison with frequentist methods (including those provided in the rebuttal) in the camera-ready version of the paper.

Our main contributions are:

1. A theoretically grounded VaR Bellman operator that optimizes a lower bound on the VaR of the expected returns (Percentile criterion) objective in Reinforcement Learning. This is a non-trivial extension of the VaR optimization in a supervised learning setting, and it also does not make any assumptions on the priors of the posterior distributions of the transition model. (Proposition 3.1)

2. Proofs showing that the VaR Bellman operator is a contraction mapping and that it optimizes a tighter lower bound on the percentile criterion. (Proposition A.4)

3. Derivation of a simplification of the VaR Bellman operator for Sub-Gaussian and Gaussian cases. (Proposition 3.2, Proposition A.5)

4. An in-depth theoretical comparison between VaR framework and the Bayesian Robust MDPs that is currently the SOTA optimization method for percentile criterion in RL (Proposition 4.1, Proposition 4.2, Theorem 4.3, Theorem 4.4, Lemma A.7, Lemma A.8).

5. Finite and asymptotic error bounds on the performance of the VaR framework (Theorem 3.3, Theorem 3.4).

6. The Generalized VaR Value iteration algorithm (Algorithm 3.1) and the corresponding error bounds. (Theorem 3.6, Proposition A.6)

7. Empirical evaluation of the VaR framework on 4 domains where we compare the VaR framework with **5** baseline methods that either optimize the percentile criterion or provide similar robustness guarantees against model (epistemic ) uncertainty (Section 5, Appendix Section C, and attached document).

---

### Decision · Program_Chairs · 2023-09-21

**Decision:**

Accept (poster)

**Comment:**

This paper considers a novel notion of ambiguity sets based on Bayesian inference to execute risk-sensitive optimization in the offline setting, with technically innovative theoretical analysis substantiating the proposed approach. While experimental evidence is limited, theoretical development and analysis is novel enough to warrant acceptance, which is mostly the perspective of the reviewers.